

# Osteology, relationships and functional morphology of *Weigeltisaurus jaekeli* (Diapsida, Weigeltisauridae) based on a complete skeleton from the Upper Permian Kupferschiefer of Germany

Adam C. Pritchard[1,2], Hans-Dieter Sues[2], Diane Scott[3] and Robert R. Reisz[3,4]

[1] Department of Paleontology, Virginia Museum of Natural History, Martinsville, Virginia, United States
[2] Department of Paleobiology, National Museum of Natural History, Smithsonian Institution, Washington, District of Columbia, United States
[3] Department of Biology, University of Toronto Mississauga, Mississauga, Ontario, Canada
[4] Dinosaur Evolution Research Centre and International Centre of Future Science, Jilin University, Changchun, China

Corresponding author
Adam C. Pritchard,
adamcarlpritchard@gmail.com

## ABSTRACT

**Background:** Weigeltisauridae is a clade of small-bodied diapsids characterized by a horned cranial frill, slender trunk and limbs, and a patagium supported by elongated bony rods. Partial skeletons and fragments are definitively known only from upper Permian (Lopingian) rocks in England, Germany, Madagascar and Russia. Despite these discoveries, there have been few detailed descriptions of weigeltisaurid skeletons, and the homologies of many skeletal elements—especially the rods supporting the patagium—remain the subject of controversy.

**Materials & Methods:** Here, we provide a detailed description of a nearly complete skeleton of *Weigeltisaurus jaekeli* from the upper Permian (Lopingian: Wuchiapingian) Kupferschiefer of Lower Saxony, Germany. Briefly addressed by past authors, the skeleton preserves a nearly complete skull, postcranial axial skeleton, appendicular skeleton, and patagial supports. Through comparisons with extant and fossil diapsids, we examine the hypotheses for the homologies of the patagial rods. To examine the phylogenetic position of Weigeltisauridae and characterize the morphology of the clade, we integrate the material and other weigeltisaurids into a parsimony-based phylogenetic analysis focused on Permo-Triassic non-saurian Diapsida and early Sauria (61 taxa, 339 characters).

**Results:** We recognize a number of intriguing anatomical features in the weigeltisaurid skeleton described here, including hollow horns on the post-temporal arch, lanceolate teeth in the posterior portion of the maxilla, the absence of a bony arch connecting the postorbital and squamosal bones, elongate and slender phalanges that resemble those of extant arboreal squamates, and patagial rods that are positioned superficial to the lateral one third of the gastral basket.
Our phylogenetic study recovers a monophyletic Weigeltisauridae including *Coelurosauravus elivensis*, *Weigeltisaurus jaekeli*, and *Rautiania* spp. The clade is

recovered as the sister taxon to Drepanosauromorpha outside of Sauria (=Lepidosauria + Archosauria).

**Conclusions:** Our anatomical observations and phylogenetic analysis show variety of plesiomorphic diapsid characters and apomorphies of Weigeltisauridae in the specimen described here. We corroborate the hypothesis that the patagial ossifications are dermal bones unrelated to the axial skeleton. The gliding apparatus of weigeltisaurids was constructed from dermal elements unknown in other known gliding diapsids. SMNK-PAL 2882 and other weigeltisaurid specimens highlight the high morphological disparity of Paleozoic diapsids already prior to their radiation in the early Mesozoic.

# INTRODUCTION

The transition between the Paleozoic and Mesozoic eras involved a radical reversal of fortunes for the Diapsida, the clade including all extant reptiles and birds. Diapsid fossils are known from Permian strata around the globe, but they are less speciose than and vastly outnumbered by nonmammalian synapsids and parareptiles. The subsequent Triassic Period saw the emergence of many new diapsid clades and their rapid rise to abundance over the parareptiles and nonmammalian synapsids that survived the end-Permian extinction (*Ezcurra, Scheyer & Butler, 2014*; *Sues, 2020a*).

Despite their relative rarity and limited species diversity, diapsids achieved a surprising morphological disparity long before they rose to prominence in the Mesozoic. Permian examples of the group include lizard-like terrestrial forms (e.g., *Gow, 1975*; *Carroll & Thompson, 1982*; *Smith & Evans, 1996*; *Gottmann-Quesada & Sander, 2009*), aquatic reptiles with sculling tails (e.g., *Carroll, 1981*; *Currie, 1981a*, *1982*; *De Buffrénil & Mazin, 1989*), and—by the very end of the period—large-bodied carnivores (e.g., *Sennikov, 1988*; *Ezcurra, 2016*). Among the most highly specialized of all known Permian diapsids are the Weigeltisauridae, a clade of diapsids with frilled crania and specialized gliding adaptations. Herein, we describe a nearly complete skeleton of a weigeltisaurid diapsid from the upper Permian (Lopingian: ?Wuchiapingian) of Germany. The completeness of the specimen provides an opportunity to describe the osteology of weigeltisaurid reptiles, contextualize the unique gliding apparatus in the clade, and revise the phylogenetic placement of the group.

## Geological context

The specimen described here was found in the Kupferschiefer of Ellrich in the Mansfeld mining district in Saxony-Anhalt of Germany (see Fig. 7 of *Paul, 2006*). The Kupferschiefer was deposited in a large marine basin extending from northern England (where it is known as the Marl Slate) far into Poland during the late Permian (Lopingian). The basin was located at a paleolatitude of about 15 to 20 degrees North and was bordered to the South by

the Variscian mountains. The Kupferschiefer is a finely laminated marly shale that usually attains a thickness of less than one meter. When freshly exposed it is dark gray to black in color due to a high content of degraded organic carbon (*Wedepohl, 1994*). The Kupferschiefer contains significant amounts of copper (locally up to 3% in Germany), lead, silver, and zinc and was mined from possibly Bronze Age times well into the last century.

The Kupferschiefer has yielded a diverse fauna of marine invertebrates and fishes along with plant and insect remains and occasional tetrapod skeletons introduced from adjoining land regions (*Haubold & Schaumberg, 1985*; *Brandt, 1997*; *Sues, 2020b*). It preserves few remains of benthic organisms, which, along with an absence of bioturbation and a relatively high amount of organic carbon, indicates deposition of the shale in anoxic deeper waters.

The Kupferschiefer Formation (T1) is the stratigraphically lowest unit of the Zechstein Group. The age of this formation continues to be contentious. It is usually dated as early Wuchiapingian based primarily on the presence of the conodont taxa *Merrillina divergens* and *Mesogondolella britannica* in both the Kupferschiefer and the overlying Zechsteinkalk (Z1; *Szurlies, 2013*). However, *Hounslow & Balabanov (2018)* noted that *Merrillina divergens* has a much more extensive stratigraphic range (including strata dated as Changhsingian by other means) and cautioned against its use as an index fossil. *Denison & Peryt (2009)* placed the entire Zechstein Group in the Changhsingian based on strontium radioisotopic data. Furthermore, *Hounslow & Balabanov (2018)* argued that the base of the Zechstein Group is in the oldest portion of the Permian magnetochron LP2n.3n and thus the Kupferschiefer would be Changhsingian in age.

## History of research

*Weigelt (1930a)* described much of the skeleton of an unusual reptile from the Kupferschiefer of Eisleben in the Mansfeld region of Saxony-Anhalt (Germany). The famous German paleontologist Otto Jaekel had purchased this specimen (now cataloged as SSWG 113/7 in the geological collections of the University of Greifswald), identified in a note as a "flying reptile," along with some other fossils by from a dealer in 1913 (Fig. 1). Preoccupied with many other projects, he never published on this fossil but prepared it himself. When Weigelt succeeded Jaekel as professor of geology and paleontology at the University of Greifswald he found the specimen in the collections. He described it as a new taxon, *Palaeochamaeleo jaekeli* and assigned it to Rhynchocephalia. Weigelt emphasized the distinctive casque formed by the posterior portion of the cranium, which bore a striking resemblance to those in chamaeleonid lizards. Associated with the skeleton were bundles of rod-like bones. Jaekel considered these rods fin-rays of an overlying caudal fin of the actinistian fish *Coelacanthus granulatus* and removed many of them during preparation to expose the reptilian skeleton.

*Huene (1930)* first drew attention to various similarities between *Palaeochamaeleo jaekeli* and *Coelurosauravus elivensis*, which was briefly described by *Piveteau (1926)* from late Permian strata of the Lower Sakamena Formation in Madagascar. He considered both taxa closely related and interpreted them as very slender-limbed reptiles capable of

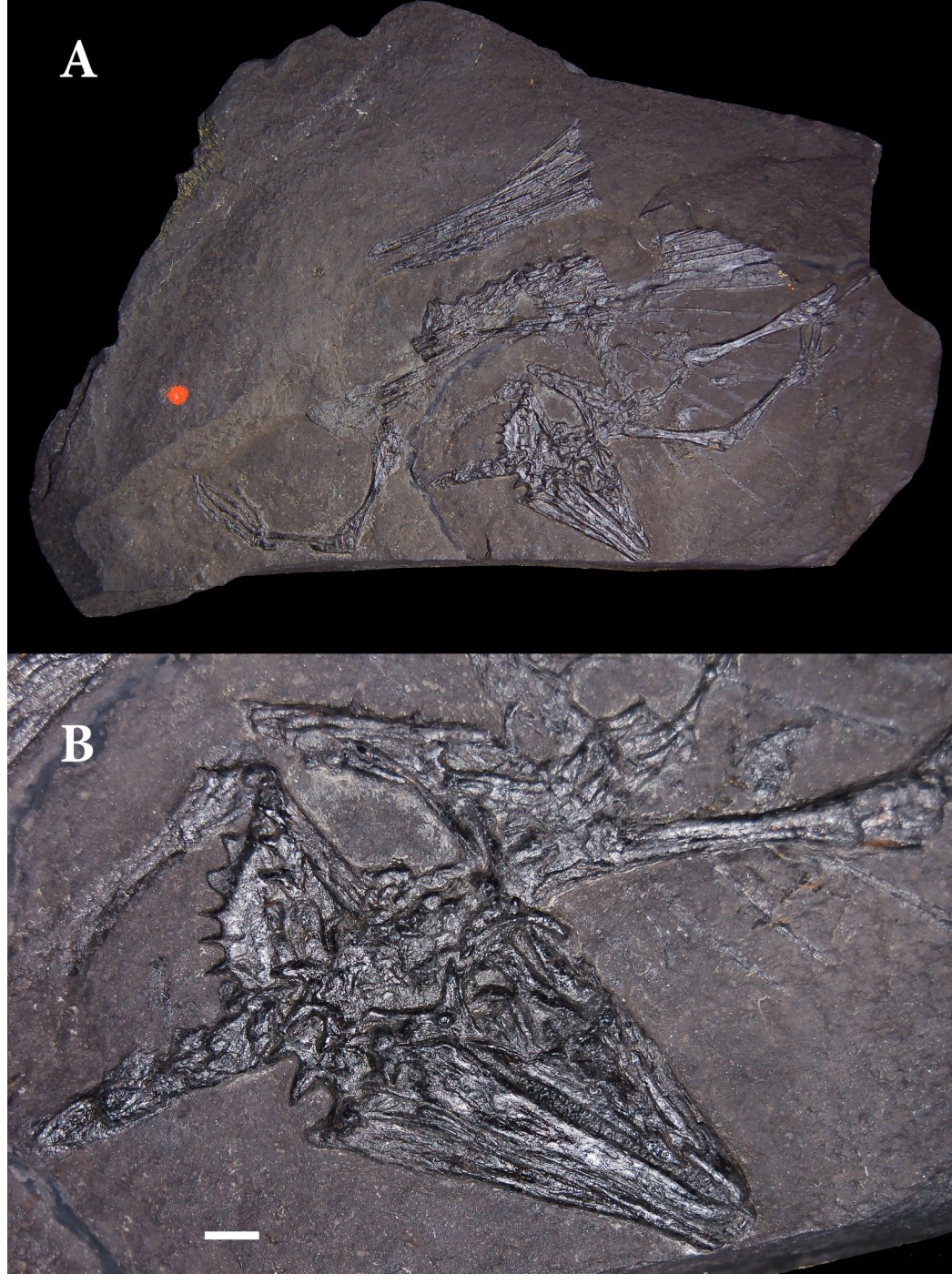

**Figure 1 Holotype skeleton of *Weigeltisaurus jaekeli* (SSWG 113/7) from the Kupferschiefer of Germany.** (A) Photograph of complete specimen as preserved in 2010. (B) Close-up photograph of articulated skull in right lateral view. Photographs taken in 2010, following removal of most patagial elements by O. Jaekel. Scale bar equals 5 mm.

climbing. He quoted a letter from the British paleontologist D. M. S. Watson who had examined Piveteau's specimens in 1927 and concluded that denticulated bones identified by Piveteau as lower jaws were more likely parts of a chameleon-like cranial frill.

*Kuhn (1939)* noted that the generic nomen *Palaeochamaeleo* was preoccupied by *De Stefano (1903)* and proposed the replacement name *Weigeltisaurus*. He also reinterpreted the temporal region of the cranium as just having a single large temporal opening on either side rather than the common diapsid condition with two openings suggested by *Weigelt (1930a)*.

After several decades, interest in *Weigeltisaurus* and *Coelurosauravus* was reawakened by the discovery of additional specimens of the former in Germany (*Schaumberg, 1976*, *1986*) and England (*Pettigrew, 1979*). Schaumberg noted the presence of numerous bony rods in the new skeletons reported by him and argued that they were parts of the reptile's skeleton rather than of overlying coelacanth fins. Independently, *Carroll (1978)* re-examined the material of *Coelurosauravus elivensis* reported by *Piveteau (1926)*. He segregated one nearly complete skeleton as a new taxon, *Daedalosaurus madagascariensis*. Carroll interpreted its long rod-like bones as greatly elongated ribs and the new reptile as a glider.

*Evans (1982)* provided a detailed description of the well-preserved partial skeleton referred to *Weigeltisaurus jaekeli* by *Pettigrew (1979)* from the Marl Slate of the Eppleton quarry near Hetton-le-Hole in northeastern England. She also reviewed the then-known specimens of *Weigeltisaurus*, *Coelurosauravus*, and *Daedalosaurus*. Correcting some of Carroll's interpretations of skeletal features, Evans concluded that *Daedalosaurus* was a subjective junior synonym of *Coelurosauravus*. She placed both *Coelurosauravus* and *Weigeltisaurus* in a family Coelurosauravidae. Evans considered the trunk ribs bipartite, each composed of a short proximal and a greatly elongated distal portion.

Following re-examination of the holotype of *Weigeltisaurus jaekeli* and identification of another specimen, *Evans & Haubold (1987)* argued that this taxon was congeneric with *Coelurosauravus elivensis*. This reallocation was subsequently widely accepted in the literature. The authors also synonymized *Gracilisaurus ottoi*, based on disarticulated postcranial remains of a small reptile from the Kupferschiefer of the Otto-Schacht (mine) in the Mansfeld region (*Weigelt, 1930b*), with *Coelurosauravus jaekeli*.

The exceptionally complete skeleton that forms the basis of this paper was found near Ellrich in the Mansfeld region of Saxony-Anhalt (Germany) in 1992. It is preserved as part and counterpart. A private collector acquired one of the slabs and later the Staatliches Museum für Naturkunde Karlsruhe purchased the other. *Frey, Sues & Munk (1997)* briefly discussed the nature of the rod-like bones that supported a gliding membrane and explicitly noted that these structures were neomorphs rather than parts of ribs. *Schaumberg (1976)* had hinted at this possibility but hesitated to state it explicitly. *Schaumberg, Unwin & Brandt (2007)* provided additional information on the Ellrich specimen and discussed the structure of the gliding apparatus.

In 2005, a Russian expedition prospecting late Permian lacustrine strata near the village of Kul'chumovo in the Saraktashkii district of the Orenburg region (Russia) discovered a deposit of numerous mostly isolated but well-preserved bones referable to *Weigeltisaurus*-like reptiles. *Bulanov & Sennikov (2006)* identified a new genus *Rautiania* and divided it into two species based on differences among the cranial bones. *Bulanov & Sennikov (2010)* provided a detailed anatomical description of additional material

referable to *Rautiania*. Subsequently, these authors re-examined *Coelurosauravus elivensis* (*Bulanov & Sennikov, 2015a*) and *Weigeltisaurus jaekeli* (*Bulanov & Sennikov, 2015b*). They argued for maintaining a generic separation of these two taxa and noted that Weigeltisauridae *Kuhn, 1939* has clear priority over Coelurosauravidae *Evans, 1982*. Like *Kuhn (1939)*, *Bulanov & Sennikov (2015a, 2015b)* reconstructed the temporal region of the cranium as having a single large temporal opening on either side. Finally, *Bulanov & Sennikov (2015c)* recognized an additional weigeltisaurid taxon, *Glaurung schneideri*, based on much of a skeleton including the skull found in the vicinity of Mansfeld and now housed in a private collection. *Schaumberg, Unwin & Brandt (2007)* had previously briefly discussed this find, which they identified as *Coelurosauravus* sp. *Glaurung schneideri* differs from *Weigeltisaurus jaekeli* in various cranial features, especially the proportionately much broader parietals and squamosals.

# MATERIALS & METHODS

## Systematic Paleontology

Diapsida Osborn, 1903
Weigeltisauridae *Kuhn, 1939*
*Weigeltisaurus* *Kuhn, 1939*

Type species: *Weigeltisaurus jaekeli* (*Weigelt, 1930a*)

SYNONYMY
*Palaeochamaeleo jaekeli* *Weigelt, 1930a*.
*Gracilisaurus ottoi* *Weigelt, 1930b*.
*Coelurosauravus jaekeli* (*Weigelt, 1930a*).

**Holotype:** SSWG 113/7, complete skull, nearly complete forelimbs, partial hindlimbs, articulated dorsal vertebrae, scattered patagials (many prepared away).

**Referred specimen**: SMNK-PAL 2882, nearly complete skull and skeleton preserved on a slab (part only).

**Stratigraphic occurrence**: Kupferschiefer of Germany (Lopingian, Permian) and correlative Marl Slate of England.

**Differential diagnosis**: We herein list a number of characters that allow referral of SMNK-PAL 2882 to *Weigeltisaurus jaekeli* and that differentiate the specimen from other named weigeltisaurid taxa. In all listed characters, SMNK-PAL 2882 compares favorably with SSWG 113/7, the holotype of *Weigeltisaurus jaekeli*, and the diagnosis offered for the species by *Bulanov & Sennikov (2015b)*. A full revision of the taxonomy of European weigeltisaurids is beyond the scope of this paper,

SMNK-PAL 2882 differs from *Coelurosauravus elivensis* in (1) presence of prominent horns on dorsolateral surface of parietal (absent in *C. elivensis*), (2) jugal with anteroposteriorly short, rapidly tapering anterior process (proportionally longer in *C. elivensis*), and, tentatively, (3) ventral bases of anterior patagial ossifications with

little-to-no inter-element spacing (widely spaced in *C. elivensis* based on MNHN.F. MAP327).

SMNK-PAL 2882 differs from the material referred to *Rautiania* sp. in (1) premaxilla with space for eight to nine teeth (more than ten teeth present in referred premaxillae of *Rautiania* sp.), (2) lateral horn on quadratojugal proportionally shorter than all but dorsalmost squamosal horn (lateral horn on quadratojugal proportionally longer in *Rautiania* sp.)

SMNK-PAL 2882 differs from *Rautiania alexandri* in (1) possessing dorsoventrally shallow and gradually tapering jugal facet on posterodorsal face of maxilla (proportionally broader and more rapidly tapering in *R. alexandri*) and (2) bearing slender, tapering horns separated by concave spaces on lateral surface of parietal (horns are dorsoventrally broader without spaces in *R. alexandri*).

SMNK-PAL 2882 differs from *Rautiania minichi* in (1) possessing dorsoventrally shallow and gradually tapering jugal facet on posterodorsal face of maxilla (proportionally broader and more rapidly tapering in *R. minichi*).

SMNK-PAL 2882 differs from *Glaurung schneideri* in (1) presence of prominent, tapering horns on lateral surface of parietal (rugose margin present in *G. schneideri*), (2) presence of slender lateral horns on lateral surface of squamosal, separated by distinct gaps (broader horns without gaps in *G. schneideri*), and (3) presence of lateral horns on dorsalmost portion of squamosal (dorsal horns absent in *G. schneideri*).

## Overview and Preservation

SMNK-PAL 2882 is a nearly complete skeleton of *Weigeltisaurus jaekeli* (Fig. 2). *Frey, Sues & Munk (1997)* and *Schaumberg, Unwin & Brandt (2007)* provided preliminary anatomical details of the specimen, focusing on the anatomy of the skull and the homology of the patagial spars. The specimen is preserved as part and counterpart on two separate slabs. Only the part has been accessioned in the collections of the Staatliches Museum für Naturkunde Karlsruhe (as SMNK-PAL 2882). The counterpart is held in a private collection and is inaccessible to scientific study.

The skull is preserved in ventral view (Fig. 3). Identifiable elements of the palate and braincase are absent, such that the ventral surfaces of the rostrum and skull roof are exposed. Much of the left mandibular ramus is also absent, although a tiny dentigerous portion of the dentary and the postdentary complex are preserved around the level of the orbit. Much of the skull is slightly disarticulated, and many bones remain three-dimensional and uncrushed. Our observations of the skull in SMNK-PAL 2882 corroborate the reconstruction of the *Weigeltisaurus jaekeli* skull in lateral view presented by *Bulanov & Sennikov (2015b*: Fig. 2).

Substantial segments of the vertebral column are missing in SMNK-PAL 2882 (Fig. 2). Some cervical vertebrae are absent. Only a small number of dorsal vertebrae are preserved in lateral view. Two partial dorsal vertebrae are preserved in articulation posterior to the pectoral girdle. A large gap separates these from the next segment, beginning at roughly 1/3 the length of the trunk region. These consist of a posterior half of a dorsal vertebra articulated to a series of seven transversely compressed dorsal vertebrae. The second and

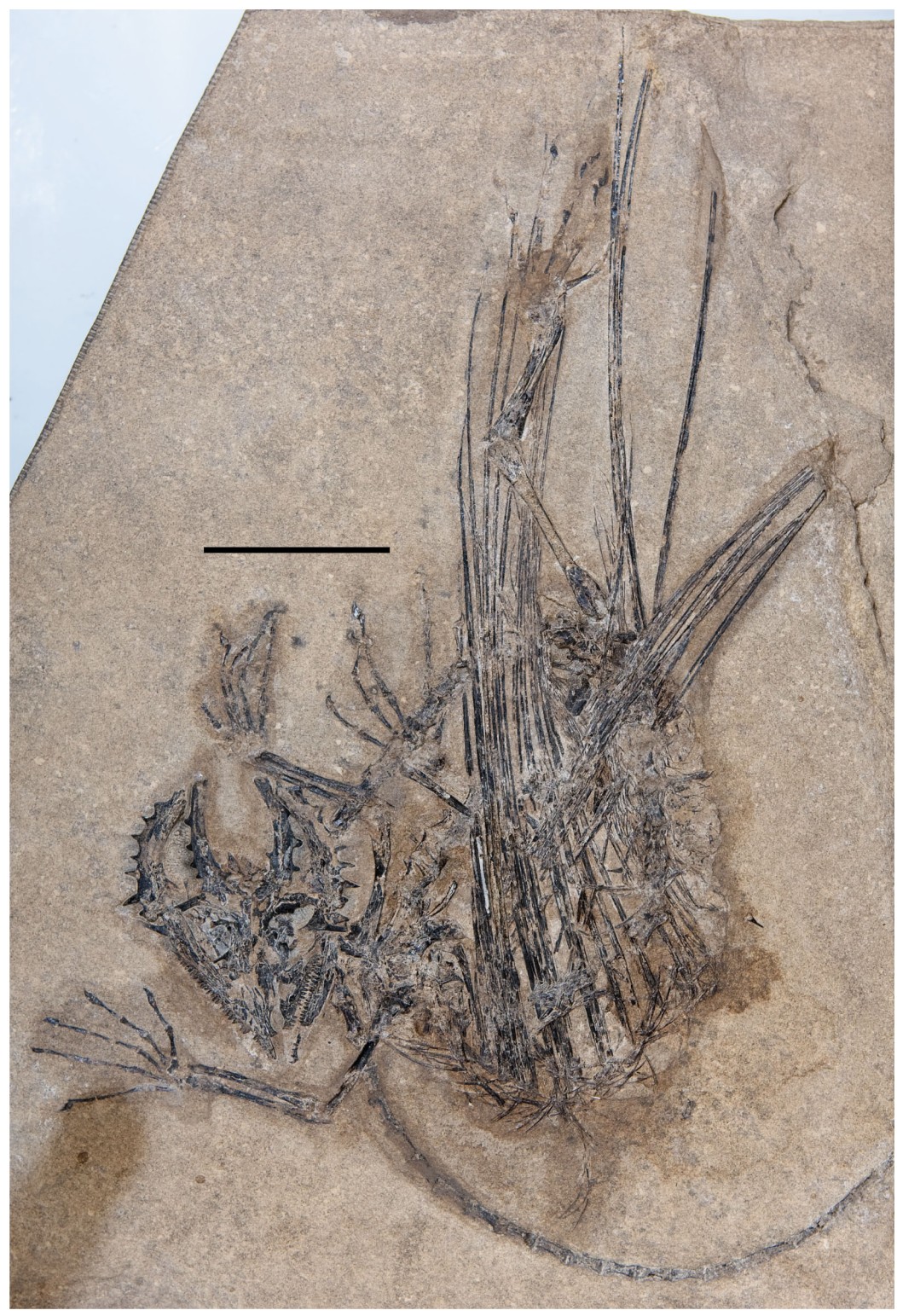

**Figure 2** **The skeleton of SMNK-PAL 2882 (*Weigeltisaurus jaekeli*).** Scale bar equals 5 cm.

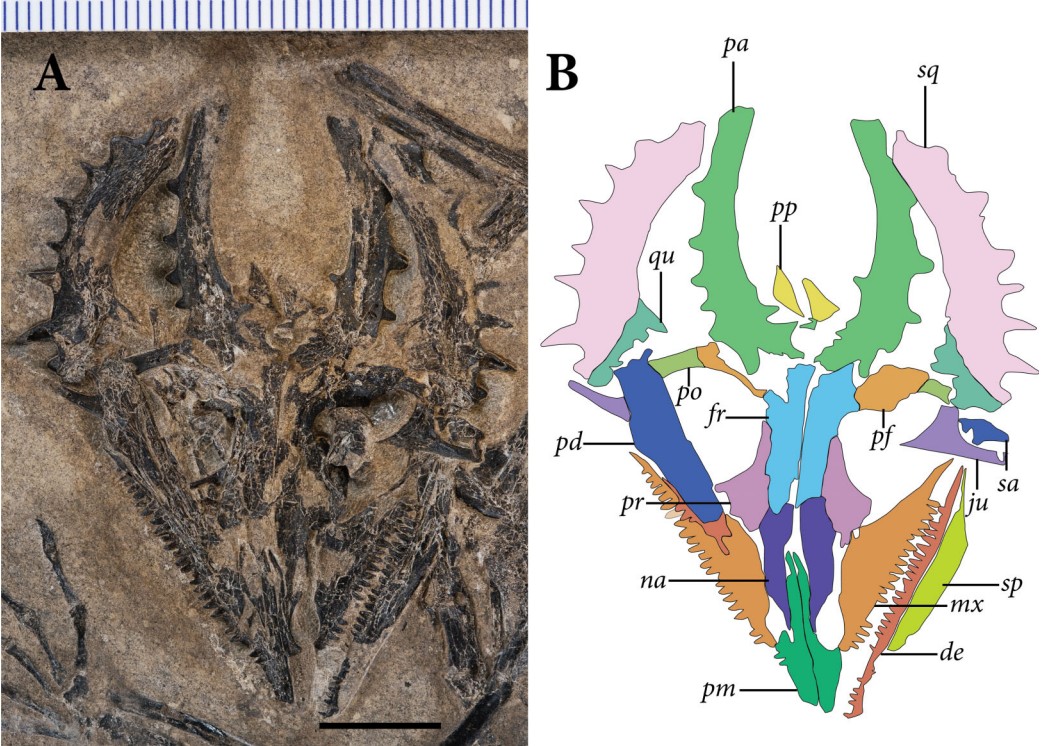

**Figure 3 The skull of SMNK-PAL 2882 (*Weigeltisaurus jaekeli*).** (A) Photograph of specimen. (B) Tracing of skull elements with identification callouts. Abbreviations: **de**, dentary; **fr**, frontal; **mx**, maxilla; **pd**, postdentary bones (unable to delineate sutures); **pf**, postfrontal; **pm**, premaxilla; **po**, postorbital; **pp**, postparietal; **pr**, prefrontal; **qu**, quadrate; **sa**, surangular (right horn); **sp**, splenial; **sq**, squamosal and quadratojugal (unable to delineate sutures between bones). Scale bar equals 1 cm.

seventh in the series are only partially preserved, surrounded by an impression of the complete element.

The most complete portion of the vertebral column is the caudal series. The anteriormost portion of the tail is covered by a sheet of appressed right patagial ossifications, followed by a series of seven articulated caudal vertebrae. The mid-portion of this series is superficial to the right tarsus. The subsequent two caudal vertebrae are preserved deep to proximal portion of the right forearm, followed by a substantial segment of the tail preserved positioned deep to the skull. The preserved tail terminates as a series of 23 posterior caudal vertebrae, which curves in parallel to the curvature of the trunk region. A few probable chevrons are preserved as well.

The appendicular skeleton is largely complete in SMNK-PAL 2882, although the pectoral and pelvic girdles are badly crushed. In our interpretation, the left scapulocoracoid is nearly complete and exposed in lateral view. The anterior edge of the scapulocoracoid is framed by two anteroposteriorly slender, curved rods of bone, which we interpret as the clavicle and cleithrum, respectively. Both forelimbs are complete except for the carpal elements—of which only fragments are preserved—and the first digit of the left manus.

The pelvic region is exposed in dorsal view between two bundles of patagial spars. It is not clear how much of the pelvic girdle is preserved, although both femora appear to remain in articulation with their respective acetabula. The left hindlimb is complete except for all of the tarsal elements and segments of each pedal digit. The proximal end of the right femur is exposed, but the remainder is buried deep under a bundle of patagial spars. The right fibula is not preserved, and only the distal end of the right tibia can be seen, positioned deep under the patagial spars and an articulated series of caudal vertebrae. The right tarsal bones are not preserved, but the remainder of the right foot is nearly complete.

A large number of gastralia are preserved, framing the ventral portion of the trunk posterior to the pectoral girdle and anterior to the pelvis. Some are preserved only as impressions, and an accurate reconstruction of an individual segment is not feasible.

The most remarkable and noticeable bones present in SMNK-PAL 2882 are the patagial ossifications: straight, slender bones that extend from the ventrolateral portion of the trunk. Some of these represent the longest bones in the skeleton, being over four times the length of the left femur. The left patagial ossifications are very incomplete. The distal ends of a bundle from the anterior trunk region are preserved overlapping the posterior trunk vertebrae and the proximal ends of the right patagial ossifications near the pelvis. The right patagial ossifications are nearly complete, positioned deep to the preserved dorsal vertebral column. The posteriormost patagial ossifications are tightly bundled together and anteroposteriorly short, making an accurate count of the ossifications difficult.

It is plausible that the counterpart of SMNK-PAL 2882 preserves many of the missing bones. Considering the phylogenetic and functional importance of the palate, braincase, carpus, and tarsus, we encourage future paleontologists to pursue the specimen and bring it into a publicly accessible museum collection for study.

## Phylogenetic Definitions

For the clade Neodiapsida *Benton, 1985*, we employ the stem-based definition of *Reisz, Modesto & Scott (2011*: 3733*)* as "*Youngina capensis* Broom, 1914 [12] and all species more closely related to it than to *Petrolacosaurus kansensis* Lane, 1945 [18]," in contrast to the node-based definition of *Laurin (1991)* which used Younginiformes as a reference taxon. As noted by *Bickelmann, Müller & Reisz (2009)* and *Reisz, Modesto & Scott (2011)*, Younginiformes is likely a non-monophyletic grouping of non-saurian diapsids. We note that this definition of Neodiapsida would incorporate a wide range of non-traditional taxa such as Parareptilia and Varanopidae under the phylogenetic hypotheses of *Laurin & Piñeiro (2017)* and *Ford & Benson (2019b)*. Future revisions to the definition of Neodiapsida are encouraged.

The clade Avicephala *Senter, 2004* was defined by *Senter (2004*: 261) as "all taxa more closely related to *Coelurosauravus* and *Megalancosaurus* than to Neodiapsida." As our definition of Neodiapsida would incorporate weigeltisaurids and drepanosauromorphs, a revised definition is required. We herein redefine Avicephala as a stem-based taxon including all taxa more closely related to *Weigeltisaurus jaekeli* Weigelt 1930 and

*Drepanosaurus unguicaudatus* Pinna 1979 than to *Petrolacosaurus kansensis* Lane 1945, *Orovenator mayorum* Reisz, Modesto & Scott, 2011, *Claudiosaurus germaini* Carroll, 1978, *Youngina capensis* Broom 1914, or Sauria Macartney 1802 (sensu Gauthier, Estes & De Queiroz (1988)). If future analyses strongly support the non-monophyly of Avicephala relative to the other reference taxa listed, we recommend the abandonment of the taxon.

Although Weigeltisauridae was first named by Kuhn (1939), no modern phylogenetic definition for the clade has been offered. Kuhn (1969) defined the order Weigeltisauria and family Weigeltisauridae based on a series of anatomical characters including the single temporal fenestra, ornamented post-temporal arches, and an elongated internarial process of the premaxilla. To coincide with the intent of this definition we define Weigeltisauridae as a stem-based taxon including *Weigeltisaurus jaekeli* Weigelt 1930 and all taxa more closely related to it than *Petrolacosaurus kansensis* Lane 1945, *Orovenator mayorum* Reisz, Modesto & Scott, 2011, *Drepanosaurus unguicaudatus* Pinna 1979, *Claudiosaurus germaini* Carroll, 1978, *Youngina capensis* Broom 1914, and Sauria Macartney 1802 (sensu Gauthier, Estes & de Queiroz (1988)).

## Phylogenetic Methods

To explore the phylogenetic affinities of *Weigeltisaurus jaekeli* and Weigeltisauridae among diapsid reptiles, we integrated new codings from SMNK-PAL 2882 into a modified phylogenetic matrix based on the diapsid analysis from Pritchard & Sues (2019). This analysis represents a downstream modification of earlier analyses presented in Pritchard & Nesbitt (2017) and Pritchard et al. (2018). Changes to codings based on observations of new material and new literature are noted in Appendix 1. Our modified matrix does not take into account modifications suggested by Scheyer et al. (2020), which focused on the affinities of the saurian *Colobops noviportensis*, as that is outside the scope of this work. Outside of additional codings for weigeltisaurid taxa, the most substantial changes occur in the codings of *Orovenator mayorum* based on the description of Ford & Benson (2019a).

Our analysis employs the araeoscelid diapsid *Petrolacosaurus kansensis* as an outgroup, as in many prior studies of diapsid interrelationships (e.g., Dilkes, 1998; Ezcurra, 2016). We note that numerous recent phylogenetic analyses recover some non-diapsid taxa as more closely related to Neodiapsida than Araeoscelida. Laurin & Piñeiro (2017) presented a modification of the Laurin & Reisz (1995) analysis in which Araeoscelida were recovered outside of a clade including *Paleothyris*, 'Younginiformes,' and Parareptilia. This analysis was heavily criticized by MacDougall et al. (2018) who argued that Laurin & Piñeiro (2017) had used an outdated data matrix and did not account for the substantial variation in temporal fenestration in Palaeozoic amniotes (but see Laurin & Piñeiro, 2018). Ford & Benson (2019b) recovered Araeoscelida as the outgroup of the clade Varanopidae + (Parareptilia + Neodiapsida). Although we do not address these phylogenetic possibilities with the taxon sample in our analysis, none of these results are incongruent with the use of araeoscelids as an outgroup to a clade including the diapsid sample employed within.

Two new characters were added to describe possible apomorphies for Weigeltisauridae:
338) Premaxilla, anterodorsal process, contribution to anteroposterior length of rostrum: (0) contribution to anteroposterior length of rostrum subequal to that of alveolar

process of premaxilla; (1) contribution to anteroposterior length of rostrum twice that of alveolar process of premaxilla.

This character describes the considerable elongation of the anterodorsal (= internarial) process of the premaxilla in known weigeltisaurids. This feature is evident in *Weigeltisaurus jaekeli* (SMNK-PAL 2882) and *Rautiania* spp. (*Bulanov & Sennikov, 2010*). In nearly all other diapsid reptiles, the contribution of the anterodorsal process to the length of the rostrum by the anterodorsal process is subequal to that of the tooth-bearing portion of the premaxilla. Unfortunately, this character could not be coded for *Coelurosauravus elivensis* as no premaxillae have been identified in the available material. Taxa coded as state "1" for Character 5, which describes the presence or absence of an anterodorsal process, are coded as "-" for this character.

339) Postorbital, posterior process for articulation with squamosal: (0) present, contacting squamosal posteriorly; (1) absent, no contact between posterior portion of postorbital and anterior portion of squamosal.

This character describes one of the most remarkable anatomical traits of known Weigeltisauridae: the apparent absence of the upper temporal bar formed by the postorbital and squamosal bones [here recognized in SMNK-PAL 2882 and by *Bulanov & Sennikov (2010, 2015a, 2015b)*]. As such, weigeltisaurids have a continuous, large temporal opening, extending from the lower temporal bar ventrally to the medial elements of the skull roof dorsally. In all other diapsids in this analysis for which the temporal region is completely known, the postorbital extends posteriorly to contact the squamosal (e.g., *Reisz, 1981*; *Gow, 1975*; *Modesto & Sues, 2004*; *Simões et al., 2018*). Taxa coded as "1" for this character are coded as "-" for Character 51, which describes the relative dorsoventral position of the upper temporal bar.

We also completely redefined Character 204 from the analysis by *Pritchard & Sues (2019)*, which initially described the presence or absence of bipartite dorsal ribs. This character was used to describe the patagial ossifications of weigeltisaurids following the hypothesis by *Evans (1982)*. However, in light of the work of *Schaumberg (1986)*, *Frey, Sues & Munk (1997)*, and the anatomical observations described below, we consider these structures to be dermal ossifications rather than part of the axial skeleton. The codings have not changed from prior iterations of this analysis. *Weigeltisaurus jaekeli* and *Coelurosauravus elivensis* are coded as "present" for this character. *Rautiania* spp. is coded as "?", as patagials have not been reported in the Russian fossils. The new version of Character 204 reads as follows:

204) Patagial ossifications (elongate bony spars positioned lateral/superficial to dorsal ribs: (0) absent, (1) present.

We analyzed the matrix in TNT v 1.5 (*Goloboff & Catalano, 2016*). We used the traditional search option with 10,000 replicates of Wagner trees followed by tree bisection and reconnection (TBR), holding 10 trees per replicate. The best trees found were subjected to a final round of TBR branch swapping. We used branch collapsing Rule 1 of *Coddington & Scharff (1994)*, collapsing all branches with a minimum length of 0 in any most-parsimonious tree. *Petrolacosaurus kansensis* was designated as the outgroup for the analysis. We ran the STATS.RUN script file to obtain the consistency and retention

indices and the BREMER.RUN script to obtain decay indices for the branches. This matrix is available on Morphobank (www.morphobank.org) as Project 3656. Jackknife values were obtained in TNT by 10,000 replicates with a 20% character-removal probability per replicate; the values are presented for each branch as frequency difference values.

# RESULTS

## Cranium

The **premaxillae** each consist of an alveolar process and an anterodorsal process. The left premaxilla is complete and preserved in ventromedial view, whereas the right premaxilla only preserves the anterodorsal process. The alveolar process is triangular and anteriorly acuminate, contributing to a sharply pointed rostrum. The preserved portion of the left alveolar process preserves space for at least eight premaxillary teeth, although only the penultimate two teeth are preserved in position. This compares favorably with the skull of the holotype of *Weigeltisaurus jaekeli* (SSWG 113/7; *Bulanov & Sennikov, 2015b*) and is lower than the tooth count of 12 or 13 in *Rautiania* spp. (PIN 5130/43; *Bulanov & Sennikov, 2010*). This indicates variability in the premaxillary tooth count of Weigeltisauridae, although they compare well with the upper end of the range of counts in early Neodiapsida. Low counts of three to five teeth occur in *Petrolacosaurus kansensis* (*Reisz, 1981*), *Orovenator mayorum* (*Ford & Benson, 2019a*), and *Protorosaurus speneri* (*Gottmann-Quesada & Sander, 2009*). Higher counts occur in *Gephyrosaurus bridensis* (eight to 10 teeth; *Evans, 1980*) and *Claudiosaurus germaini* (up to 13 teeth in SAM-PK-8263; and *Carroll, 1981*).

The posterior margin of the alveolar process is concave where it would articulate with the anterior process of the maxilla. Dorsal to the posterior part of the alveolar process, the premaxilla contributes an anteroposteriorly elongate, anteriorly tapered margin to the external naris. There is no posterodorsal process of the premaxilla, such that the maxilla forms much of the posterior margin of the external naris. This narial conformation compares well with the holotype of *Weigeltisaurus jaekeli* (SSWG 113/7; *Bulanov & Sennikov, 2015b*), *Rautiania* sp. (PIN 5130/44; *Bulanov & Sennikov, 2010*), and non-archosauromorph diapsids such as *Petrolacosaurus kansensis* (*Reisz, 1981*) and the *Tropidostoma* Zone younginiform (SAM/PK 7710).

The anterodorsal process is twice the anteroposterior length of the alveolar process, a feature common in weigeltisaurids (e.g., SSWG 113/7, PIN 5130/44). It extends posterodorsally from the anterior tip of the premaxilla. At the level of the posterior edge of the alveolar process, the anterodorsal process of the premaxilla contacts the medial edge of the nasal. This contact extends for the remainder of the anteroposterior length of the anterodorsal process. The anterodorsal process exhibits a uniform transverse width throughout most of its anteroposterior length. The process begins to taper transversely at the level of the sixth maxillary alveolus, tapering to a point by the level of the ninth maxillary alveolus. Similarly elongate anterodorsal processes of the premaxillae occur in the drepanosauromorph *Megalancosaurus preonensis* (e.g., *Renesto & Dalla Vecchia, 2005*) and pterosaurs such as *Eudimorphodon ranzii* (e.g., *Wild, 1978*) and *Rhamphorhynchus* spp. (e.g., *Bonde & Leal, 2015*).

The **nasals** are partially preserved. The right element is nearly complete, whereas the left preserves only the anterior tip and posterior articulation with the left frontal. They are rhomboid in outline, with tapering anterior and posterior processes.

The anterior process is positioned at roughly the level of the first two maxillary alveoli. It abuts the lateral margin of the anterodorsal process of the premaxilla throughout its length. The lateral margin of the anterior process contributes the dorsomedial margin of the external naris. The posterior edge of the anterior process of the nasal contacts the anterior margin of the dorsal process of the maxilla, forming the posterior border of the external naris. We do not identify a tubercle on the dorsolateral surface of the anterior process of the nasal as described by *Bulanov & Sennikov (2015b)* in SSWG 113/7, but the structure may be embedded in matrix in SMNK-PAL 2882.

Posterior to the external naris, the nasal remains unchanged in transverse width throughout most of the remainder of its length. Medially, the nasal contacts the elongate anterodorsal process of the premaxilla posteriorly to the level of the 11th maxillary alveolus. Posterior to this maxillary alveolus, the two nasals contact one another along the midline for the remainder of their lengths. The ventral surface of the nasal is slightly concave.

Posterior to its contribution to the external naris, the lateral margin of the nasal contacts the dorsal process of the maxilla. This contact is sigmoid, complementing curvatures on the anterodorsal surface of the dorsal process of the maxilla. This contact ends at the level of the fifteenth maxillary alveolus, where the anterior edge of the nasal posterior process sits.

The posterior process of the nasal tapers transversely throughout its length. Laterally, it contacts the medial margin of the prefrontal. The contact between the nasal and the anterior margin of the frontal is not clear on either side, but the nasal appears to lap dorsally over the anterior tip of the frontal. The posterior margin of the nasal posterior process is straight and transversely oriented.

The **maxillae** are nearly complete on both sides. Nearly all of the dentition is preserved in situ on the left side, whereas the right maxilla has multiple gaps in its tooth row. The bone consists of an alveolar portion that bears the teeth, a distinct anterior process, and a dorsal process that forms much of the lateral surface of the snout.

Teeth are present throughout nearly the entire anteroposterior length of the maxilla. Both maxillae preserve a short, edentulous region at the posterior tip of the alveolar process. The tooth row extends posteriorly to roughly the anteroposterior midpoint of the orbit. Where it supports the bases of the teeth, the maxilla is transversely thicker than is the dorsal process of the bone. A distinct palatal process is absent.

There is a distinct anterior process of the maxilla that supports the first two maxillary teeth. This process is concave anterodorsally where it contributes to the margin of the external naris. The anterior process increases in height dorsoventrally further posteriorly, terminating where the maxilla contacts the anterolateral edge of the nasal. The dorsal process is a prominent, dorsally convex, and transversely narrow structure that forms much of the lateral surface of the snout. It increases in dorsoventral height back to the level

of the 14th maxillary alveolus. Further posteriorly, the dorsal process decreases in dorsoventral height to its posterior terminus.

The anterodorsal margin of the dorsal process of the maxilla bears a subtly sigmoid margin, corresponding to a matching curvature on the lateral margin of the nasal. This sigmoid shape occurs in other weigeltisaurid skulls (SSWG 113/7, MNHN.F.MAP327), and contrasts with the simple convexity in other diapsids such as *Youngina capensis* (AMNH FARB 5561, BP/1 3859) and *Prolacerta broomi* (BP/1 471; *Modesto & Sues, 2004*). The contact between the dorsal margin of the maxilla and the ventrolateral margin of the nasal extends from the level of the fourth through the 14th maxillary alveoli. A small notch in the dorsal margin of the maxilla is present just posterior to the nasal contact, similar to that framing the preorbital fenestra noted in SSWG 113/7 by *Bulanov & Sennikov (2015b*: Fig. 1*)*. The dorsal process of the maxilla lacks a distinct posterior concavity.

It is not clear how the maxilla contacted the lacrimal or jugal posteriorly, nor whether or not the maxilla contributed to the margin of the orbit. The right maxilla preserves a subtle posterodorsal embayment, which may mark the contact between the maxilla and the anteroventral margin of the jugal. There is an anteroposteriorly short margin at the posterodorsal edge of the dorsal process of the maxilla where it met the anterolateral edge of the dorsal process of the prefrontal.

The **prefrontals** are preserved in articulation with the frontals and nasals on both sides (Fig. 3). Both prefrontals are exposed in ventral view. The preserved portion of the bone is trapezoidal with tapering anterior and posterior processes. The posterolateral margin of the prefrontal is marked by a prominent prefrontal boss, which contributes a substantial anterodorsal corner to the orbit. The anterior and posterior processes are preserved on both sides, but the prefrontal boss is only exposed on the right side. A passage for the nasolacrimal canal is not apparent in either prefrontal of SMNK-PAL 2882.

The anterior process of the prefrontal is anteroposteriorly short. Based on the right prefrontal, it may bifurcate at its anterior tip. The right anterior process appears to be positioned near its anatomical position, situated between the posterodorsal corner of the dorsal process of the maxilla laterally and the posterolateral corner of the nasal medially. The contact with the nasal on the skull roof is anterolaterally oriented. The anterior process broadens posteriorly, reaching the maximum transverse breadth of the prefrontal at the anterior margin of the orbit. At this point, the bone is twice as broad transversely as the broadest part of the frontal.

On the right prefrontal, the prefrontal boss projects posterolaterally from the transversely widest portion of the bone. The boss itself is laterally convex and forms a prominent anterodorsal frame to the margin of the orbit. No boss is exposed on the left side, but it is likely that it remains buried in the matrix. The prominent boss compares well with the prefrontal of the holotype of *Weigeltisaurus jaekeli* (SSWG 113/7) and resembles the condition in iguanian squamates (e.g., *Evans, 2008*).

Posterior to the boss, the dorsal lamina of the prefrontal tapers posteromedially as a distinct posterior process. Medially, the prefrontal contacts the frontal for much of its anteroposterior length. The contact extends posteriorly for two-thirds the total length of

the frontal, such that the prefrontal makes up nearly the entire anterolateral border of the orbit. The prefrontal-frontal suture is straight and roughly parasagittal in orientation. At its posterior tip, the prefrontal tapers posterolaterally.

The **frontals** are anteroposteriorly elongate and slender bones with a tapering anterior margin and a transversely broadened posterior margin. Both frontals remain in articulation, and they are exposed in ventral view. Although the outer margins of each frontal are discernible, the exposed bone surfaces are heavily cracked, and much of the central portion of the right frontal has weathered away completely. The bony contribution to the canal for the olfactory bulb and tract cannot be discerned. We also cannot identify a ventral lamina of the medialmost portion of the frontal in SMNK-PAL 2882. This feature was noted by *Bulanov & Sennikov (2015a)* as a possible unique character of *Coelurosauravus elivensis*, contrasting with specimens of *Rautiania* spp. and *Weigeltisaurus jaekeli*.

Anteriorly, the frontal extends just beyond the anterior margin of the orbit. The anterior margin of the frontal is convex and makes contact with the posterior margin of the nasal. It is not clear whether or not the tapered anterior margins of the two frontals met one another in the midline.

Between the levels of the anterior and posterior margins of the orbit the frontal is roughly rectangular, retaining the same transverse breadth. For much of its length, its lateral surface contacts the medial surface of the prefrontal. At the posterolateral corner of the orbit, the prefrontal-frontal contact terminates such that the latter element has a small, laterally concave contribution to the orbit itself.

Posterior to its orbital contribution, the frontal expands transversely to twice its breadth further anteriorly. This compares well with the transverse expansion of the frontals in many squamates, such as *Iguana iguana* and *Tupinambis teguixin* (e.g., *Evans, 2008*; *Gauthier et al., 2012*). The anterolateral corner of this expanded region contacts the anteromedial edge of the postfrontal bone at a laterally concave suture. Posterior to the postfrontal-frontal suture, the frontal bears a small posterolateral embayment for receipt of an anterolateral process of the parietal.

Medially, the frontals remain in contact with one another along a sagittal suture from the level of the anterior margin of the orbit to their contact with the parietals posteriorly. Medial to the contact with the anterolateral process of the parietal noted above, the frontal contacts the central portion of the parietal at a posterolaterally concave suture. This contact forms a broad, 'W'-shaped frontoparietal contact in ventral view, with a posteriorly convex central portion. By contrast, many early diapsids have an anteriorly convex, 'U'-shaped frontoparietal contact as seen in *Clevosaurus hudsoni* (*Fraser, 1988*) and *Youngina capensis* (*Gow, 1975*).

The **parietal** is a complex bone with three primary processes: a short anterolateral process, a medial lamina in the skull roof, and a massive and elongate posterolateral process that frames the dorsomedial margin of a large temporal fenestra. Both parietals are largely complete and exposed in ventral view. The exposed ventral surface of both bones is cracked and weathered. Most of the margins are well defined, although the medial edges of the left and right laminae in the skull roof are heavily eroded.

The anterolateral process of the parietal is short and fits against the posterolateral corner of the frontal. Posterior to its contact with the frontal, the right anterolateral process contacts the right postfrontal throughout the rest of its anteroposterior length. The main portion of the roofing lamina extends medially posterior to the frontal. The right lamina is heavily cracked medially, such that its original outline is uncertain. The left lamina bears a deep medial concavity, which we identify as a margin of the parietal foramen. The roofing lamina terminates immediately posterior to the parietal foramen. Parietal foramina occur broadly in non-saurian amniotes such as *Aerosaurus wellesi* (*Langston & Reisz, 1981*), *Araeoscelis gracilis* (MCZ 4173; *Reisz, Berman & Scott, 1984*), and *Youngina capensis* (BP/1 70, 3859). They are absent in known drepanosauromorph skulls (e.g., *Megalancosaurus preonensis*, MPUM 8437; *Avicranium renestoi*, AMNH FARB 30834).

The posterolateral processes of both parietals are well preserved. They are greatly elongated—making up more than one-quarter of the total length of the skull—and convex laterally. They are strongly posteriorly oriented, their long axes extending almost parasagittally. This elongation and near posterior orientation occurs in *Coelurosauravus elivensis* (MNHN.F.MAP327; *Bulanov & Sennikov, 2015a*), *Rautiania* spp. (*Bulanov & Sennikov, 2006*), and *Glaurung schneideri* (*Bulanov & Sennikov, 2015c*). The only other diapsids in which the parietals achieve a similar proportional length and inclination are choristoderes, such as *Coeruleodraco jurassicus* (*Matsumoto et al., 2019*) and *Champsosaurus laramiensis* (*Brown, 1905*).

Prominent horns are present along nearly the entire anteroposterior length of the lateral surface of the parietal. Five lateral horns are evident on the left parietal and four are evident on the right. Two small horns are present on the anteromedial margin as well. In their preliminary description of SMNK-PAL 2882, *Schaumberg, Unwin & Brandt (2007)* illustrated the parietals with unornamented lateral margins. More recent preparation of the specimen by D. Scott revealed the presence of these horns. Horns occur in the holotype of *Weigeltisaurus jaekeli* (SSWG 113/7; *Bulanov & Sennikov, 2015b*) and *Rautiania* spp. (*Bulanov & Sennikov, 2006*). In *Coelurosauravus elivensis* (MNHN.F.MAP327; *Bulanov & Sennikov, 2015a*) and *Glaurung schneideri* (*Bulanov & Sennikov, 2015c*), the lateral margin of the parietal is marked by a roughened, rugose margin but no distinct horns.

In SMNK-PAL 2882, there are no clear embayments or crests for the attachment of adductor muscles on the lateral surfaces of the parietals. The absence of such crests is commonplace in other weigeltisaurids (e.g., SSWG 113/7, PIN 5130/1) and early diapsids such as *Araeoscelis gracilis* (e.g., MCZ 4173; *Vaughn, 1955*), *Avicranium renestoi* (AMNH FARB 30834), and *Youngina capensis* (BP/1 3859; *Pritchard et al., 2018*). Crests and embayments occur in most early Sauria, such as *Prolacerta broomi* (BP/1 5375; *Modesto & Sues, 2004*), *Protorosaurus speneri* (*Gottmann-Quesada & Sander, 2009*), *Trilophosaurus buettneri* (TMM 31025-140; *Gregory, 1945*), and *Clevosaurus hudsoni* (NHMUK R 36832; *O'Brien, Whiteside & Marshall, 2018*).

The left parietal in SMNK-PAL 2882 bears a series of four dorsolaterally projecting horns. The horns appear blunt at their tips, similar to the second-from-the-dorsalmost horn on the left squamosal. The dorsalmost horn on the left parietal is positioned just

anterior to the parietal-squamosal contact. Only three horns are present on the lateral surface of the posterolateral process of the right parietal. These are positioned symmetrically relative to the first, third, and fourth horns on the lateral edge of the left parietal. Each is blunt and similar in shape to the horns on the right side. The number of horns evident on the parietals of SMNK-PAL 2882 is lower than the number in SSWG 113/7 and those reconstructed for *Rautiania* spp. (*Bulanov & Sennikov, 2006*). However, the total number in SMNK-PAL 2882 is likely higher and obscured by the parietals being exposed in ventral view and the dorsal tips of both squamosals overlying them.

Dorsal to the dorsalmost horn on the posterolateral process of the parietal, the bone is flat where it contacted the posteromedial margin of the squamosal. Posteromedial to the contact, the posterolateral process terminates in a flat and transversely oriented surface. The two bones are slightly disarticulated on both sides. Medially, there are two small horns positioned one posterior to the other on each side. On the left posterolateral process, the anterior horn is narrow and tapering whereas the posterior horn is very short and blunt. The tip of the anterior horn on the right side is broken, such that its shape cannot be discerned. The posterior horn on the right side is small and tapered, similar to the anterior horn on the left side. Posterior to the dorsalmost horn on the medial margin, the posterolateral process of the parietal is linear and subtly concave. We cannot identify a distinct supratemporal bone nor a sutural surface on the parietal for its reception.

Both **jugals** are preserved disarticulated and exposed in medial view. Only the right jugal is fully exposed. The element is triradiate, consisting of an anterior process, a dorsal process, and a posterior process. The medial surface of the right jugal is cracked but not weathered. It appears to be flat and unornamented, without a distinct sutural surface for the ectopterygoid.

The anterior process of the jugal is relatively shorter than the other two processes. It is triangular and anteriorly tapered, comparing well with the holotype of *Weigeltisaurus jaekeli* (SSWG 113/7). It differs markedly from the narrow, gradually tapering anterior processes in *Coelurosauravus elivensis* (MNHN.F.MAP327) and in other early diapsids such as *Claudiosaurus germaini* (*Carroll, 1981*), *Acerosodontosaurus piveteaui* (*Bickelmann, Müller & Reisz, 2009*), *Youngina capensis* (BP/1 3859; *Gow, 1975*), and *Prolacerta broomi* (UCMP 37151; *Modesto & Sues, 2004*). In SMNK-PAL 2882, the dorsal margin of the anterior process slopes anteroventrally, whereas the ventral margin is horizontal and continuous with the ventral margin of the posterior process. Based on the articular surfaces on the posterior process of the maxilla and the positions of the articulations in SSWG 113/7, the entire anterior process of the jugal slotted over the posterodorsal margin of the posterior process of the maxilla.

The dorsal process of the jugal forms the ventral half of the postorbital bar. It tapers along its dorsoventral height, terminating in a dorsally positioned concavity. Based on comparisons with SSWG 113/7, this concavity received the ventral tip of the ventral process of the postorbital. A similarly broad tip of the dorsal process of the jugal occurs in *Coelurosauravus elivensis* (e.g., *Bulanov & Sennikov, 2015a*) and may represent a synapomorphy of Weigeltisauridae. In most early diapsids, the dorsal process of the jugal

tapers to a point dorsally and slots posterior to the ventral process of the postorbital as seen in *Petrolacosaurus kansensis* (*Reisz, 1981*), *Youngina capensis* (BP/1 3859; *Gow, 1975*), and *Azendohsaurus madagaskarensis* (*Flynn et al., 2010*).

The posterior process of the jugal is relatively longer than the other processes of this bone. It is completely preserved on the right side, but its posterior tip is weathered on the left. The dorsal and ventral margins of the process are subparallel throughout its anteroposterior length, such that it does not taper posteriorly. The posterior tip of the posterior process is concave, a surface that forms the suture for the quadratojugal in SSWG 113/7. All available evidence indicates that a complete lower temporal bar was present in SMNK-PAL 2882 and other specimens of Weigeltisauridae. A closed temporal bar formed by the jugal and quadratojugal also occurs in *Petrolacosaurus kansensis* (*Reisz, 1981*), *Youngina capensis* (TM 3603; *Gow, 1975*), hyperodapedontine rhynchosaurs (*Benton, 1983*), and basal archosauriforms (e.g., *Nesbitt, 2011*; *Ezcurra, 2016*).

The **postorbital** and **postfrontal** are difficult to distinguish in SMNK-PAL 2882. We present tentative sutural identifications in Fig. 3B, and we describe the bones by region below. The articulated postorbital and postfrontal consist of a ventrolaterally projecting postorbital process and an anteroposteriorly broad, subtriangular postfrontal process.

The postorbital process is roughly rectangular, with subparallel anterior and posterior margins. The ventral margin of the postorbital process is deeply concave, presumably at the facet for the dorsal process of the jugal. At its dorsal tip, the process grade smoothly into the ventrolateral edge of the triangular postfrontal process. The postfrontal process broadens medially. The posterior margin of the postfrontal is straight and transversely oriented. A similar, transversely oriented posterior margin of the postfrontal occurs in *Petrolacosaurus kansensis* (*Reisz, 1981*), *Protorosaurus speneri* (*Gottmann-Quesada & Sander, 2009*), and *Prolacerta broomi* (BP/1 5375; *Modesto & Sues, 2004*). However, this margin is in contact with a medial process of a discrete postorbital in all of these species. The posterior margin of the postfrontal is strongly posteromedially inclined in *Avicranium renestoi* (AMNH FARB 30834; *Pritchard & Nesbitt, 2017*), *Youngina capensis* (BP/1 3859, SAM-PK 7578), *Claudiosaurus germaini* (*Carroll, 1981*), *Mesosuchus browni* (SAM-PK 6536; *Dilkes, 1998*), and *Trilophosaurus buettneri* (TMM 31025-140; *Gregory, 1945*).

In SMNK-PAL 2882, the medial articular surface of the postfrontal bears a posteromedial concavity where it contacts the posterolateral edge of the frontal. It appears to have contacted the lateral surface of the anterolateral process of the parietal for a short distance.

We cannot distinguish the **quadratojugal** and **squamosal** on either side of the skull in SMNK-PAL 2882; based on comparisons with *Rautiania* spp. (e.g., PIN 5130/41), both contribute to the posttemporal arch of the cranium. Together they frame the lateral and posterodorsal margins of the quadrate. We will describe them here as a single unit.

The posttemporal arch forms a dorsoventrally tall, medially concave, and laterally convex structure that forms the posterior margin of the large temporal fenestra. Laterally, the bones bear a single row of eight laterally oriented horns that vary in apicobasal length. Herein, we number the horns 1–8 from the dorsalmost to the ventralmost.

Horn 1 is the shortest and is laterally rounded. Horn 2 is substantially taller dorsoventrally and wider transversely, but with a similarly rounded lateral margin to Horn 1. Horns 3 and 4 are similar in transverse breadth on the right and left sides, although they are substantially more weathered on the right side. Each is longer than Horn 2. The complete left horns are strongly tapered with sharp distal edges, although the weathered right horns appear more rounded. A tiny additional hornlet sits just ventral to Horn 3 on the right side.

Horns 5–7 increase sequentially in both transverse breadth and dorsoventral height. Each is similarly sharp and distally tapered. Horns 6 and 7 are the second-largest and largest horns, respectively. The posterior surfaces of the seventh horns on the right and left sides are cracked, exposing the internal surfaces of the horns. Each is clearly hollow, with the cortical bone being less than a millimeter in thickness. The eighth horn sits directly dorsolateral to the lateral quadrate condyle. It is the shortest transversely and is laterally rounded. Based on comparisons with the sutures in the posttemporal arches of *Rautiania* spp. (e.g., PIN 5130/41), the quadratojugal likely contributes only to this lowermost horn.

The nearly complete skull of the holotype of *Weigeltisaurus jaekeli* (SSWG 113/7) bears eight horns, one of which was likely attached to the quadratojugal (*Bulanov & Sennikov, 2015b*). In *Coelurosauravus elivensis*, the squamosal bears only five horns (MNHN.F. MAP317, 325, 327). However, it may not be articulated with the quadratojugal such that the lowermost horn is absent (*Bulanov & Sennikov, 2015a*). Seven spines, including a definitive lowermost quadratojugal spine, occur in *Rautiania* sp. (PIN 5130/41). In both of these species, the spines are slender and acuminate akin to those on the squamosal/quadratojugal of SMNK-PAL 2882.

Ventrally, the squamosal overlaps the quadrate along a posterolaterally sloping suture. The squamosal completely obscures the posterior surface of the dorsal tip of the quadrate in posterior view. This contact is dorsoventrally short, reaching only to the dorsoventral level of the sixth horn. The squamosal lamina lapping posterior to the quadrate compares favorably with *Rautiania* sp. (PIN 5130/41; *Bulanov & Sennikov, 2010*). This tight contact in Weigeltisauridae resembles the condition in early reptiles (e.g., *Captorhinus aguti*; *Heaton, 1979*; *Paleothyris acadiana*; *Carroll, 1969*; *Petrolacosaurus kansensis*; *Reisz, 1981*) and drepanosauromorphs (*Avicranium renestoi*; *Pritchard & Nesbitt, 2017*).

The posteroventral surface of the squamosal is heavily weathered on both sides. The left element is eroded away medially between the fifth and seventh squamosal horns, whereas the right is nearly completely weathered away between the first and third horns and laterally weathered between the third and sixth horns. The best-preserved surface is the posterodorsal surface of the left element, which is marked by subparallel dorsoventrally extending ridges.

The squamosal in SMNK-PAL 2882 bears only one clear articulation with other dermatocranial elements, that of the dorsal portion of the bone with the posterolateral process of the parietal. As in SSWG 113/7, there is no anterior process for articulation with the postorbital to form a supratemporal arch. This absence is consistent with the squamosals known for *Coelurosauravus elivensis* (MNHN.F.MAP327) and *Rautiania* spp.

(e.g., PIN 5130/41), supporting the reconstructions by *Bulanov & Sennikov (2015a,* *2015b*) showing weigeltisaurid skulls with continuous infratemporal and supratemporal fenestrae.

The configuration of the temporal region in Weigeltisauridae, with its single continuous temporal opening spanning the lower temporal bar to the parietal, is remarkable for an amniote. A continuity between the upper and lower temporal fenestrae occurs in numerous squamate species in which the postorbital and squamosal lack processes that contact one another, such as *Anniella pulchra* and *Eryx colubrinus*. However, the absence of an upper temporal bar occurs in concert with the absence of the lower temporal bar in these species, suggesting it may involve the kinetic system in Squamata (*Gauthier et al.,* *2012*).

We identify two small, dorsoventrally flattened bones positioned posterior to the roofing laminae of the parietals as the **postparietals** based on their topographic position, apparent symmetry, and absence of complex articulations for the bones of the braincase. This differs from the interpretation by *Schaumberg, Unwin & Brandt (2007)*, who identified the triangular bones as exoccipitals. The bones do not resemble the exoccipitals of any other known diapsid. They are triangular with a flattened anterior surface and a tapering posterior tip. We suggest that the flattened anterior margin was appressed to the posterior margin of the dorsal roofing lamina. It is not clear if the medial margins of the two postparietals contacted one another in life.

Postparietal ossifications are common among early eureptiles, such as *Captorhinus aguti* (*Heaton, 1979*), *Petrolacosaurus kansensis* (*Reisz, 1981*), and *Youngina capensis* (BP/1 375; *Carroll, 1981*). These vary widely in relative size and shape. The bones in captorhinids and araeoscelids are transversely broad elements of the occiput, fitting against much of the posterior margin of the parietals (*Heaton, 1979*; *Reisz, 1981*). In *Y. capensis*—the only early neodiapsid for which it can be confidently identified—the postparietal is proportionally narrower than the associated parietal. The narrow postparietal in SMNK-PAL 2882 compares most favorably with the character state in *Y. capensis*.

The **quadrate** is preserved on the left and right sides, although both quadrates are overlain posteriorly by the quadratojugal and squamosal. Only the lateral condyle of the articular end can be seen. It appears strongly convex ventrally, suggesting the presence of a double convexity similar to those in *Rautiania* spp. (e.g., *Bulanov & Sennikov, 2010*) and *Coelurosauravus elivensis* (e.g., MNHN.F.MAP327; *Bulanov & Sennikov, 2015a*). A small portion of the medially oriented pterygoid lamina is preserved on both sides, preserving the dorsal margin of the structure on the left side. The dorsal margin is positioned approximately at the level of the second-from-ventralmost squamosal horn. Both pterygoid laminae are broken at a posteromedially oriented crack, unsurprising considering the absence of any palatal bones. A similar biconvex articulation for the articular bone occurs in *Araeoscelis gracilis* (*Vaughn, 1955*), *Czatkowiella harae* (*Borsuk-Białynicka & Evans, 2009*), and *Gephyrosaurus bridensis* (*Evans, 1980*). An undivided convexity is present in various, distantly related diapsids, including *Avicranium renestoi* (*Pritchard & Nesbitt, 2017*) and *Trilophosaurus buettneri* (TMM 31025-140; *Gregory,* *1945*).

Two small, rod-like bones are present on both sides of the skull posterior to the postorbitals and medial to the quadrates. Each bears a slight expansion in the shaft at one end. They are quite small, shorter in length than the longest of the squamosal horn. These bones could represent the stapes or hyoid cornua.

## Mandible

Both mandibular rami are partially preserved in SMNK-PAL 2882. The left ramus preserves only its posteriormost tooth positions and the entire postdentary complex. The right ramus is only partially exposed. The dentigerous portion of the dentary is fully exposed, and much of the right postdentary complex remains buried in the matrix. Only a small portion of the surangular, preserving a single lateral horn, is exposed anteroventral to the right quadrate.

The **dentaries** are elongate, slender bones with tapered anterior tips. The dentigerous margin of the jaw is straight. The right dentary is the more complete and exposed in medial (=lingual) view. It is preserved in two pieces—one anterior and one posterior—but an impression in the matrix shows its original extent. The anterior piece preserves the anterodorsal tip of the dentary and six partial teeth. The medial surface of the posterior segment is covered almost entirely by the splenial. It preserves 19 teeth in varying states of completeness.

The medial surface of the dentary on the right side is covered by a black sheet of bone, which we identify as the **splenial**. It is fractured into many fragments along its dorsal margin, but it is intact ventrally. The preserved bone is smoothly textured, and its ventral margin is straight.

The left postdentary complex is complete but badly fractured into small fragments. It is exposed in dorsomedial view. The sutural boundaries between the **angular**, **coronoid**, **surangular**, **articular**, and **prearticular** cannot be discerned. The postdentary complex expanded slightly in dorsoventral height posterior to the tooth row, with a prominent, dorsally convex coronoid eminence. In this way, it resembles the rounded coronoid eminence in *Petrolacosaurus kansensis* (*Reisz, 1981*), *Araeoscelis gracilis* (MCZ 4173; *Vaughn, 1955*), and *Claudiosaurus germaini* (SAM PK-8263). The dorsal margin of the bone slopes slightly posteroventrally to its articulation with the quadrate. There is a small ventromedial bony projection posterior to the articular, which may represent a short retroarticular process. A similar, anteroposteriorly and dorsoventrally short retroarticular process occurs in the skull of the holotype of *Weigeltisaurus jaekeli* (SSWG 113/7), *Youngina capensis* (BP/1 2871; *Gow, 1975*), and *Megalancosaurus preonensis* (MFSN 1769; *Renesto, 2000*). By contrast, a retroarticular process appears absent in *Petrolacosaurus kansensis* (*Reisz, 1981*) and *Araeoscelis gracilis* (MCZ 4173; *Vaughn, 1955*). On the right side of the skull of SMNK-PAL 2882, a small laterally facing horn is exposed anteroventral to the right quadrate. Based on comparisons with the postdentary complexes of *Rautiania* spp. and the skull of the holotype of *Weigeltisaurus jaekeli*, this horn likely belongs to the right surangular (*Bulanov & Sennikov, 2010*, *2015b*).

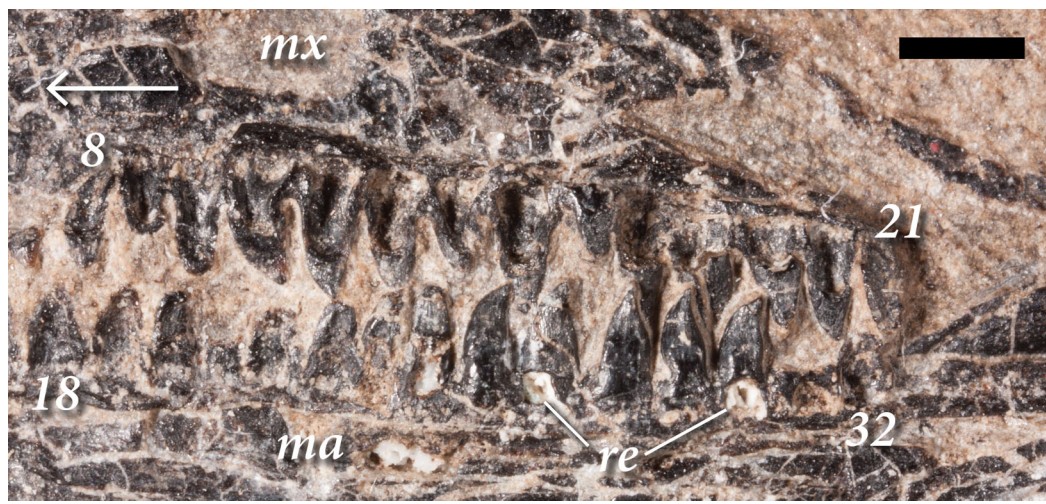

**Figure 4 The right upper and lower marginal dentition of SMNK-PAL 2882 (*Weigeltisaurus jaekeli*) in lingual view.** Abbreviations: **8**, eighth maxillary tooth position; **18**, eighteenth dentary tooth position; **21**, twenty-first maxillary tooth position; **32**, thirty-second dentary tooth position; **ma**, mandibular ramus; **mx**, maxilla. Arrow indicates anteroposterior axis of the skull. Scale bar equals 2 mm.

## Dentition

The teeth are subtly heterodont in SMNK-PAL 2882, transitioning from small and relatively simple pegs anteriorly to lanceolate, recurved teeth further posteriorly (Fig. 4). The anteriormost teeth are preserved in the left premaxilla and the anterior end of the right dentary. They are small and taper apically without being noticeably recurved. The dentary teeth are not as well preserved, with the enamel being heavily abraded. Similar conical teeth occur anteriorly in the skull of the holotype of *Weigeltisaurus jaekeli* (SSWG 113/7). In *Rautiania* spp. the premaxillary teeth possess a slight distal recurvature at the tips of the crowns but this may be a result of the exceptional preservation of those specimens (*Bulanov & Sennikov, 2010*).

The left maxilla preserves space for 22 total teeth, whereas the right preserves space for 21. This number is comparable with the 22–23 described in SSWG 113/7 (*Bulanov & Sennikov, 2015b*) and *Rautiania minichi* (*Bulanov & Sennikov, 2006*). 30 teeth are present in the maxillae of *Rautiania alexandri* (*Bulanov & Sennikov, 2006*). Within the maxilla, the anteriormost three or four teeth resemble larger versions of those in the premaxilla. However, they are relatively longer apicobasally and exhibit a very slight degree of curvature. The mesial margins of these teeth are apicodistally curved, and the distal margins are straight. The condition resembles anterior maxillary teeth in the skull of the holotype of *Weigeltisaurus jaekeli* (SSWG 113/7). By contrast, the anterior maxillary teeth of *Rautiania alexandri* are not recurved (PIN 5130/40; *Bulanov & Sennikov, 2006*).

In SMNK-PAL 2882, the next eight to 10 maxillary teeth further distally exhibit a modest mesiodistal expansion at the crown-root junction, giving them a distinctly leaf-like shape (Fig. 4). The apex of the crown is slightly recurved. The lanceolate teeth occur distal to the anterior process of the maxilla and mesial to the anteroposterior midpoint of the

dorsal process of the maxilla. The corresponding teeth in SSWG 113/7 are not particularly well preserved, but they do seem to possess both a leaf-like shape and recurvature. The lanceolate teeth of the maxilla in both species of *Rautiania* bear a subtler degree of mesiodistal expansion, but they are all similarly recurved.

The maxillary teeth distal to the lanceolate teeth are rather poorly preserved in SMNK-PAL 2882, with weathered crowns and cracked enamel on both sides. They appear to lack any mesiodistal expansion of the base of the crown, the mesial and distal margins of the teeth being relatively straight. These teeth taper rapidly very close to the apex. They resemble the poorly preserved posterior maxillary teeth of the holotype of *Weigeltisaurus jaekeli* (SSWG 113/7) and those of *Rautiania minichi* (PIN 5130/3; *Bulanov & Sennikov, 2006*).

The teeth of the right dentary differ from those in the maxilla (Fig. 4). The anteriormost teeth in the dentary are the smallest, increasing in size further posteriorly. As in the anterior part of the maxilla, these anterior teeth are relatively simple and conical. The teeth around the anteroposterior midpoint of the dentary are relatively larger, but appear simple and conical. Lanceolate teeth occur in the posterior half of the dentary, well posterior to those in the right maxilla. The right dentary contains the best-preserved of the lanceolate teeth. Each tooth bears distinct apicobasal enamel striations on the apical half of the crown. In a partial dentary referred to *Rautiania* sp. (PIN 5130/24), the anterior teeth are small and conical. The mid-posterior dentary teeth are slightly larger, subtly expanded mesiodistally, and slightly recurved.

In SMNK-PAL 2882, the posteriormost teeth in the right dentary are smaller than the mid-dentary teeth, with simple, apically rounded crowns. Only the posteriormost part of the left dentary is preserved, and all of the teeth it contains are similarly small, simple, and apically rounded.

The implantation of the teeth varies along the jaws (Figs. 3, 4). The preserved teeth of the left premaxilla show pleurodont implantation, similar to the 'iguanian mode' of pleurodonty described by *Jenkins et al. (2017)*. They sit within a groove, abutting against the medial surface of the alveolar process of the premaxilla. There is no clear medial wall to support them and the roots of individual teeth are anchored to the premaxilla by a ring of porous bone. However, the tooth roots are proportionally much shorter than those in extant *Iguana iguana* or *Gerrhosaurus validus*; in the relative shallowness of the roots, they most closely resemble the anterior pleurodont teeth of the agamid *Uromastyx acanthinura* (*Edmund, 1969*). The teeth at the anterior tip of the right dentary do not appear to have similarly sized roots; they are positioned on the apex of the bone.

The teeth in the maxillae and dentaries—excluding the posteriormost three or four teeth—also appear subtly pleurodont (Fig. 4). The short roots sit in a shallow groove, abutting against the medial surface of the alveolar process of the bone. The roots of individual teeth are mostly attached to the dentigerous elements by a ring of porous bone. There are no distinct interdental ossifications forming alveolar walls as in many archosauromorphs (*Edmund, 1969*; *Luan et al., 2009*; *LeBlanc et al., 2018*). Several teeth in both the maxillae and dentaries bear large medial resorption pits that interrupt this ring of porous bone. Small replacement teeth can be seen within some of the pits, similar to

the condition in early synapsids such as *Dimetrodon* and *Ophiacodon* (*Edmund, 1960*). However, the roots of some teeth are also cracked and broken such that the mesiodistal spacing between tooth replacement cycles in the jaws of SMNK-PAL 2882 ('z-spacing' of *DeMar & Bolt, 1981*) cannot be assessed. In the posteriormost portion of the maxilla and left dentary, the teeth do not appear to have deep roots. Instead the crowns attach at or near the dentigerous margins of the jaw elements, similar to the condition at the anteriormost tip of the right dentary.

## Vertebrae

Multiple articulated segments of the vertebral column are preserved in SMNK 2882, representing cervicals, dorsals, and caudals (Figs. 2, 5). The anteriormost cervicals spiral out from a central point, and the long tail encircles the head and trunk region.
The cervicals are not well preserved, with only vestiges of the original bone present within the impressions of the complete vertebrae. Large gaps are present in the trunk region although a series of six mid-to-posterior dorsal vertebrae is preserved just anterior to the pelvis. No sacrals are exposed on the block, but the apparent preservation of much of the pelvis suggests that they may still be buried in the matrix. A segment of seven anterior caudal vertebrae is exposed posterior to the pelvis, although the anteriormost caudals are covered by the longer patagial ossifications of the right 'wing.' The adjoining portion of the tail is covered by the skull, beyond which a segment of 22 crushed, poorly preserved mid-to-posterior caudals extends almost unobstructed.

## Cervical Vertebrae

The presumably anteriormost cervical vertebrae in SMNK-PAL 2882 are partially exposed in between the ninth and eleventh patagial spar ossifications (Fig. 5). Little can be said of their morphology, as only a small portion is exposed. A sequence of five articulated cervical vertebrae is exposed between the anterior margin of the fourth patagial ossification and the anterodorsal edge of the scapula. Preserved bone is present along the ventral surfaces of the centra; everything else is preserved as a subtle impression.

The shapes of the neural spines are obscured by poor preservation. In the second vertebra in the articulated series, there is a subtle vestige of a flat-topped spine that tapered ventrally to join with the rest of the neural arch. The spine appears dorsoventrally short, shorter than the corresponding centrum. The pedicles are also dorsoventrally short, accentuating the apparent elongation of the neural arch of SMNK-PAL 2882. Short cervical neural spines with squared-off tips and short pedicles occur in *Coelurosauravus elivensis* (MNHN.F.MAP 317, 327; *Carroll, 1978*) and *Petrolacosaurus kansensis* (*Reisz, 1981*), whereas proportionally taller spines and pedicles are present in *Hovasaurus boulei* (*Currie, 1981a*) and *Youngina capensis* (BP/1 3859; *Gow, 1975*).

Prezygapophyses are preserved as clear impressions on the second and third vertebrae in the sequence. Each is anteriorly rounded and anterodorsally inclined (Fig. 5). The processes extend well anteriorly of the anterior margin of the centrum. Postzygapophyses are not preserved in either vertebra.

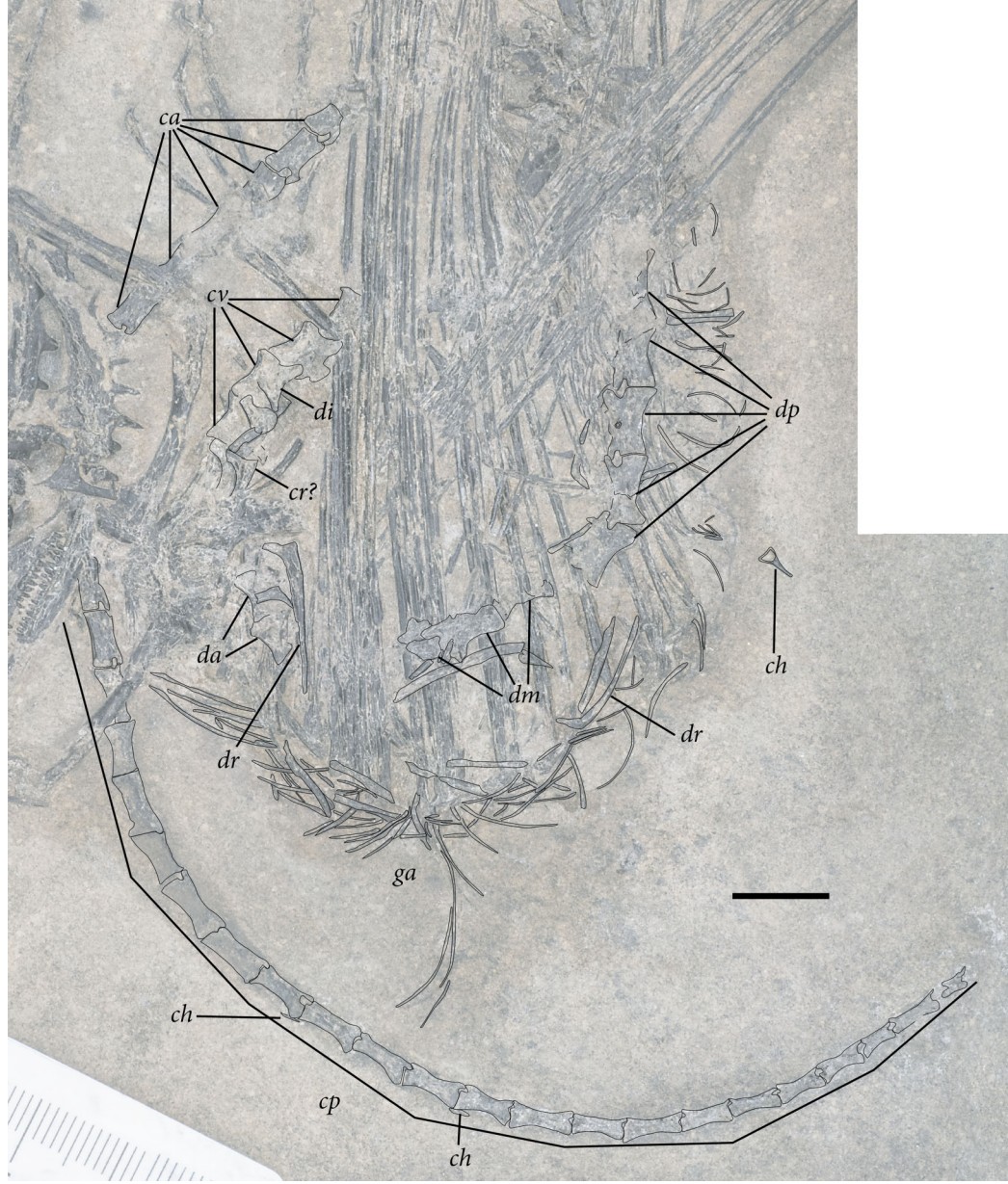

**Figure 5 Reduced opacity image of the trunk, pelvis, and tail of SMNK-PAL 2882 (*Weigeltisaurus jaekeli*), highlighted preserved segments of the vertebral column, with identification callouts.** Abbreviations: **ca**, anterior caudal vertebrae; **ch**, chevron; **cp**, posterior caudal vertebrae; **cr**, cervical rib; **cv**, cervical vertebrae; **da**, anterior dorsal vertebrae; **di**, cervical diapophysis; **dm**, mid-dorsal vertebrae; **dp**, posterior dorsal vertebrae; **dr**, dorsal rib; **ga**, gastralia. Hashmarks indicate millimeter scale. Readers may refer to Fig. 2 for an unmodified photograph of the SMNK-PAL 2882 slab. Scale bar equals 1 cm.

The second cervical vertebra in the sequence preserves a distinct diapophysis near the dorsoventral level of the prezygapophyses (Fig. 5). This structure is positioned on the anterior half of the vertebra. The third and fourth vertebrae do not preserve exposed facets. However, probable dichocephalous cervical ribs are present in the same area of the lateral surface on the third and fourth cervical vertebrae in the sequence (Fig. 5). The centra are

anteroposteriorly longer than dorsoventrally tall, although the ratio between these measures decreases from anterior to posterior. The ratio of height to length in these centra (= 2.4 in the third and most complete centrum in the series) compares well to the elongate cervical centra of *Coelurosauravus elivensis* (MNHN.F.MAP317, 327; *Carroll, 1978*), *Araeoscelis gracilis* (MCZ 4173; *Vaughn, 1955*) and *Zarcasaurus tanyderus* (CM 41704; *Brinkman, Berman & Eberth, 1984*) and contrasts with the greater height/length ratio in *Hovasaurus boulei* (*Currie, 1981a*) and *Youngina capensis* (BP/1 3859; *Gow, 1975*). Each centrum is cylindrical with a strong ventral concavity. As preserved, the anterior and posterior articular surfaces are flat. They were either amphiplatyan or amphicoelous, consistent with the morphology in most early eureptiles such as *Captorhinus aguti* (*Fox & Bowman, 1966*), *Araeoscelis gracilis* (*Vaughn, 1955*), and *Youngina capensis* (BP/1 3859; *Gow, 1975*).

In many ways, the proportionally elongate cervical centra with relatively short pedicles in weigeltisaurids and araeoscelids (following *Carroll, 1988*) resemble those of many early archosauromorphs, such as *Protorosaurus speneri* (*Gottmann-Quesada & Sander, 2009*), *Trilophosaurus buettneri* (TMM 31025-140; *Gregory, 1945*), and *Prolacerta broomi* (BP/1 2675; *Gow, 1975*). Although these archosauromorph taxa with relatively long cervical vertebrae were long considered members of a grouping variously dubbed Protorosauria or Prolacertiformes (*Wild, 1973*; *Benton, 1985*; *Evans, 1987*, *1988*), more recent analyses indicate that these reptiles represent a paraphyletic grade relative to Archosauriformes (*Dilkes, 1998*; *Pritchard et al., 2015*; *Ezcurra, 2016*; *Pritchard & Nesbitt, 2017*).

Both of these groups differ radically from the anatomy of the cervical vertebrae in non-saurian neodiapsids (e.g., *Youngina capensis*, *Thadeosaurus colcanapi*) and most lepidosauromorphs, in which the centra are much shorter anteroposteriorly and the pedicles and neural spines are proportionally taller (*Hoffstetter & Gasc, 1969*; *Gow, 1975*; *Carroll, 1981*). Among early lepidosauromorphs, such cervical vertebrae occur in *Fraxinisaura rozynekae* (*Schoch & Sues, 2018a*), *Planocephalosaurus robinsonae* (*Fraser & Walkden, 1984*), *Clevosaurus hudsoni* (*Fraser, 1988*), and vertebrae referred to *Sophineta cracoviensis* (*Evans & Borsuk-Białynicka, 2009*). Based on present phylogenetic hypotheses, it is plausible that diapsids plesiomorphically had proportionally elongated cervical vertebrae before transitioning to the relatively shorter, taller vertebrae seen in younginiforms and lepidosauromorphs. The condition in early archosauromorphs would represent a reversal to the plesiomorphic diapsid state. The transitions between these suites of vertebral features and their apparent fixation within certain major clades of Diapsida warrant further study.

## Trunk Vertebrae

As in the cervical region, trunk vertebrae are either disarticulated or preserved in short segments (Figs. 5, 6). Two trunk vertebrae are preserved in articulation just posterior to the exposed pectoral girdle. One and one-half articulated trunk vertebrae are positioned medial to the ninth through twelfth patagial ossifications just dorsal to the gastral basket.

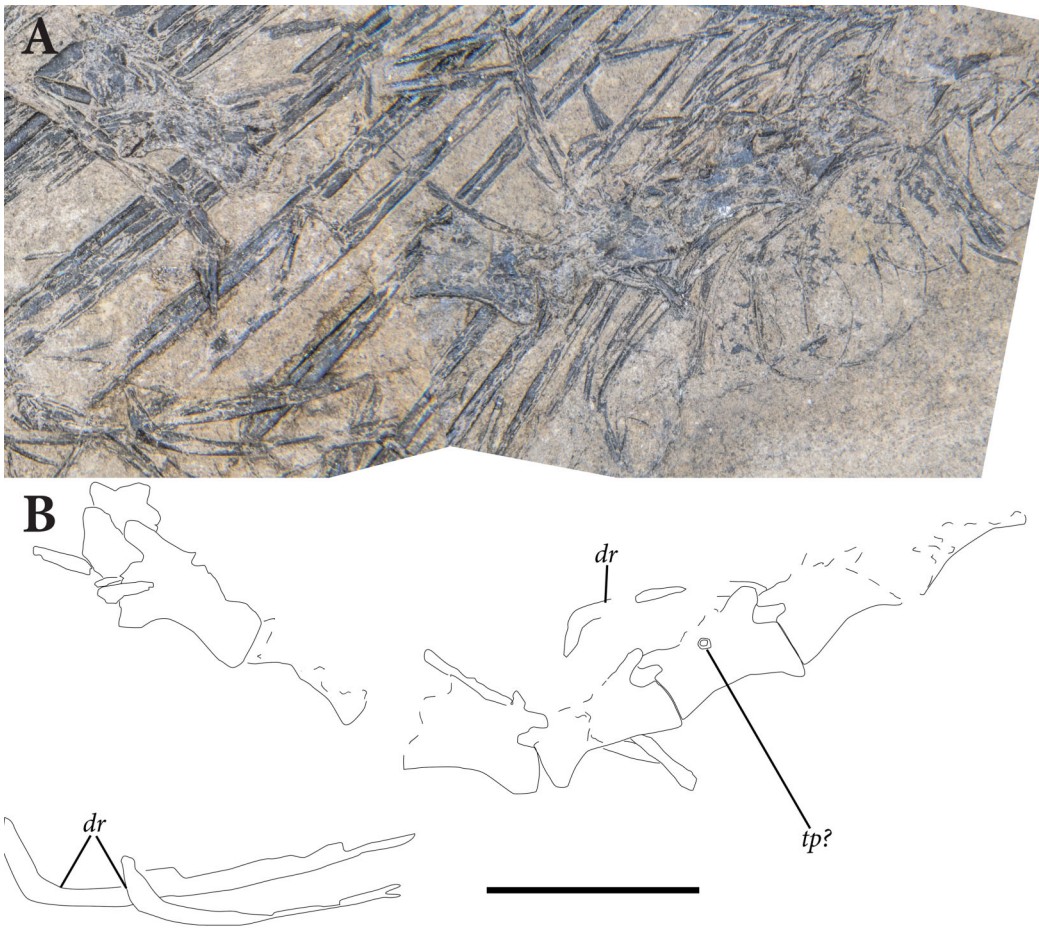

**Figure 6 The preserved mid- and posterior dorsal vertebrae and associated ribs of SMNK-PAL 2882 (*Weigeltisaurus jaekeli*).** (A) Photograph and (B) Interpretative drawing demarcating the margins of elements with interpretive callouts. Abbreviations: **dr**, dorsal ribs; **tp**, transverse process. Scale bar equals 1 cm.

A stretch of five mid-to-posterior trunk vertebrae sits posterior to the level of the fifteenth patagial ossification, curving along with the rest of the skeleton of the trunk.

The anterior trunk vertebrae preserved near the pectoral girdle only preserve small traces of the centra (Fig. 6). These are proportionally shorter than those in the posterior cervical region, a pattern evident in some diapsid reptiles such as *Crocodylus acutus* (*Mook, 1921*) and *Varanus* spp. (*Hoffstetter & Gasc, 1969*). The relative length of the vertebral centra transition from shorter to longer in many other lepidosaurian taxa, including *Iguana*, *Ophisaurus*, and *Tupinambis* (*Hoffstetter & Gasc, 1969*). Each centrum is concave ventrally and the articular surfaces—though poorly preserved—were flat or concave.

A single complete trunk vertebra is preserved medial to the eighth and ninth patagial ossifications (Fig. 6). It is heavily crushed and distorted. Its neural spine is not preserved. The prezygapophyses are elongate, anterodorsally rounded, and extend far anteriorly of the centrum. The centrum itself is proportionally longer than those in the anterior trunk region and bears a prominent ventral concavity.

The vertebrae in the articulated series represent the mid-to-posterior trunk region (Fig. 6). None of these preserves a clear neural spine. The preserved zygapophyses are smaller and shorter than those in the cervical or anterior trunk region. They extend only slightly anteriorly relative to the anterior margin of their respective centra. The postzygapophyses are similarly small and barely extend beyond the posterior margins of their respective centra. There are prominent anterior and posterior concavities ventral to the zygapophyses that clearly separate them from the centra. A small, subcircular facet is preserved on the third vertebra in this sequence.

It sits at the dorsal margin of the centrum at the anteroposterior midpoint of the vertebra. It is plausible that ribs in this portion of the trunk were holocephalous, as is common in the mid–to-posterior trunk vertebrae of diapsids (e.g., *Hoffstetter & Gasc, 1969*; *Gow, 1975*; *Reisz, 1981*; *Fraser & Walkden, 1984*; *Nesbitt et al., 2015*). The centra are proportionally more elongate than those in the anterior trunk region, similar to the increasing relative lengths in lepidosaurs such as *Iguana*, *Tupinambis*, and *Chioninia* (*Hoffstetter & Gasc, 1969*). The articular surfaces are not well exposed but were either flat or concave. Each centrum bears a prominent ventral concavity.

## Sacral vertebrae

Sacral vertebrae are not clearly exposed in the pelvic region of SMNK-PAL 2882. It is possible that some remain buried under the other pelvic elements.

## Caudal vertebrae

The caudal vertebrae are the best represented region of the axial skeleton, being relatively well preserved and representing a wide variety of regions in the column (Fig. 5). A series of articulated anterior caudal vertebrae sit several probable vertebral positions posterior to the pelvic girdle. The dorsal margins of most of these caudal neural arches are well preserved, but none bears a distinct dorsal projection or spine. The preserved caudal vertebrae also lack distinct transverse processes, suggesting that they may have been absent from all but the anteriormost caudals, comparing favorably with the anteriormost caudal neural arches of *Coelurosauravus elivensis* (MNHN.F.MAP327; *Carroll, 1978*). This stands in stark contrast to the anterior caudal neural spines projecting well dorsally of the zygapophyses in early diapsids such as *Petrolacosaurus kansensis* (*Reisz, 1981*), *Hovasaurus boulei* (*Currie, 1981a*), *Thadeosaurus colcanapi* (MNHN.F.MAP360; *Carroll, 1981*), and *Protorosaurus speneri* (*Gottmann-Quesada & Sander, 2009*).

Pre- and postzygapophyses are small and similar to those in the posterior trunk vertebrae (Fig. 5). They do not extend far beyond the anterior or posterior margins of the centra. They are separated from the centra by small anterior and posterior concavities, which are visible in lateral view. The centra themselves are cylindrical and substantially longer than they are dorsoventrally tall.

The articulated portion of the vertebral column preserved further posteriorly extends from deep to the right mandibular ramus (Fig. 5). The first two vertebrae in this sequence are well preserved, with much of the finished bone still intact. Further posteriorly, they are preserved as impressions with small vestiges of finished bone. None of these bears a

distinct neural spine. Where visible, the zygapophyses are small and do not extend far beyond their respective centra. Each centrum is greatly elongated and slender, with a substantially greater length/height ratio than in the anterior caudal vertebrae. This pattern is similar to the condition in many extant squamates (*Etheridge, 1967*) and *Trilophosaurus buettneri* (TMM 31025-140; *Nesbitt et al., 2015*). Each vertebra decreases in absolute size further posteriorly.

We consider it likely that the tail was even longer than is preserved in SMNK-PAL 2882. The posteriormost complete vertebrae still bear prominent pre- and postzygapophyses. A small portion of a caudal vertebra is preserved posterior to this, although this element is clearly incomplete.

## Ribs

Ribs are not well represented in SMNK-PAL 2882, but examples from the cervical and trunk regions are present (Figs. 5, 6). Two probable cervical ribs are present in the anterior-to-mid cervical region. Each has two distinct proximal heads that appear widely separated from one another. Dichocephalous ribs occur in *Captorhinus aguti* (*Dilkes & Reisz, 1986*) and *Petrolacosaurus kansensis* (*Reisz, 1981*), although the heads are connected by a web of bone in these taxa. Dichocephalous rib heads occur in the anterior cervical vertebrae of *Hovasaurus boulei*, transitioning to holocephalous ribs at the fifth cervical rib (*Currie, 1981a*). In SMNK-PAL 2882, each rib is positioned lateral to the anterior half of the vertebra. The ribs lack the distinct anterior process that characterizes cervical ribs in early archosauromorphs (e.g., *Nesbitt et al., 2015*; *Ezcurra, 2016*). The rib shafts are not well preserved, but they do not appear to be longer anteroposteriorly than their respective vertebrae.

A complete dorsal rib is exposed just posterior to the pectoral girdle (Fig. 5). It is dichocephalous, bearing two distinct articular eminences on its proximal end. The proximal expansion and double articular surfaces compare favorably to the anterior dorsal ribs of *Coelurosauravus elivensis* (MNHN.F.MAP327; *Carroll, 1978*), *Petrolacosaurus kansensis* (*Reisz, 1981*), and *Araeoscelis gracilis* (*Vaughn, 1955*). The shaft is relatively straight, tapering ventrally from this articulation. A dorsal rib of similar morphology is preserved alongside the gastral basket near the anteroposterior midpoint of the trunk. Splints of bone are preserved all around the trunk region that may represent partial ribs, but these cannot be differentiated from the many fragments of patagial ossifications scattered across SMNK-PAL 2882. No sacral or caudal ribs are exposed.

## Chevrons

Few definitive chevrons are exposed in SMNK-PAL 2882. A single, Y-shaped bone is preserved lateral to the gastral basket nearly two-thirds down the length of the trunk region (Fig. 5). This bone likely represents a chevron from the anterior portion of the caudal series. It is preserved in either anterior or posterior view, such that the shape in lateral view cannot be assessed. The dorsal tips of the 'Y' shape are connected by a crossbar, framing a dorsally flat and semicircular hemal canal. Similar Y-shaped chevrons with intercentrum-derived crossbars occur in *Petrolacosaurus kansensis* (*Reisz, 1981*) and

*Youngina capensis* (AMNH FARB 5561; *Currie, 1981b*). The chevron bears a distinct ventral process similar in extent to the height of the anterior caudal centra.

The anterior portion of the caudal series does not preserve any chevrons, so we cannot compare with the club-shaped anterior chevrons present in TWCMS B5937.1 and those noted by *Evans & Haubold (1987)* in the *Gracilisaurus ottoi* holotype. We identify a few chevrons still in articulation with the mid-to-posterior caudal vertebrae. Each is roughly triradiate (Fig. 5). A dorsoventrally short dorsal process articulates between two adjacent caudal centra. Further ventrally, the chevron bifurcates into distinct anterior and posterior processes that parallel the long axes of the caudal centra. The anterior and posterior processes are roughly equivalent in anteroposterior length, akin to the condition in *Trilophosaurus buettneri* (TMM 31025-140; *Gregory, 1945*). In the absence of prominent neural spines and elongated chevrons, the tail would have been extremely slender throughout much of its anteroposterior length. In extant lepidosaurs, these processes support the longitudinal muscles necessary for flexion of the tail (e.g., *Ali, 1948*). The lack of elongate processes throughout much of the caudal series in *Weigeltisaurus jaekeli* is strikingly similar to the condition in some species of *Draco* [e.g., *D. dussumieri* (FLMNH Herp 61535, ark:/87602/m4/M36111)]. This may indicate a limited degree of flexibility and powerful motions in the weigeltisaurid tail relative to those of most other Permo-Triassic diapsids.

## APPENDICULAR SKELETON

### Pectoral girdle

The pectoral girdle in SMNK-PAL 2882 is heavily crushed and fractured, such that determining the original sutural boundaries between elements—or even the relative extent of the right and left halves preserved—is difficult (Fig. 7). In our interpretation, the exposed portion of the girdle includes the lateral surfaces of the left **scapula**, **coracoid**, **clavicle**, and **cleithrum**. The **interclavicle**, if it was present, cannot be distinguished from the crushed pieces of the scapulocoracoid.

The scapula has a transversely thin blade that is dorsoventrally taller than it is anteroposteriorly long (Fig. 7). The posterior margin of the blade is angled posterodorsally, with a strong angulation near its dorsoventral midpoint. The dorsal margin of the blade is subtly convex and heavily weathered. The anterodorsal margin of the scapular blade is inclined anteroventrally, subparallel to the posterior margin of the bone. The preserved fragments of bone that make up the scapular blade are smoothly textured laterally. In its length/height ratio and orientation, the scapular blade in SMNK-PAL 2882 resembles the condition in *Coelurosauravus elivensis* (MNHN.F.MAP327; *Carroll, 1978*) and *Youngina capensis* (*Gow, 1975*).

Further ventrally, the anterior surface of the scapular blade is convex and subtly rounded. In this region, the blade is more heavily fragmented with little of the original bone still exposed. Due to the fracturing of the bone, it is not at all clear where the scapula and coracoid contacted one another.

At the posteroventral edge of the scapular blade, the smoothly textured bone abruptly transitions into pebbly, gray-colored bone (Fig. 7). This rougher-textured bone appears to

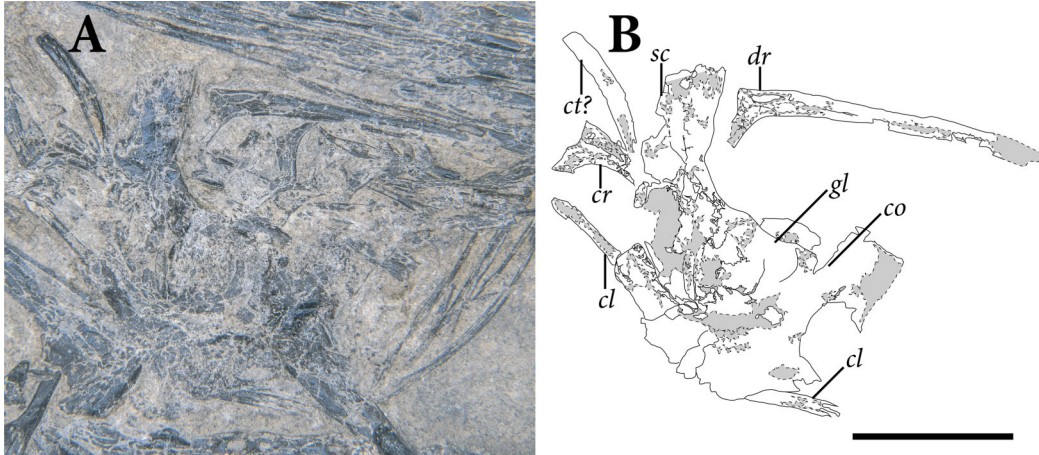

**Figure 7 The pectoral girdle and anterior trunk region of SMNK-PAL 2882 (*Weigeltisaurus jaekeli*) in left lateral view.** (A) photograph of specimen. (B) Interpretative drawing demarcating the margins of elements and the cracks and gaps within individual bones and identification callouts. Abbreviations: **co**, coracoid; **cr**, cervical rib; **ct**, cleithrum; **dr**, dorsal rib; **gl**, glenoid fossa of scapulocoracoid; **sc**, scapula. Scale bar equals 1 cm.

be restricted to the posterolateral surface of the scapula ventral to the blade, which we consider to represent the position of the glenoid fossa. The fossa likely faced posterodorsally, as in *Youngina capensis* (BP/1 3859) and *Prolacerta broomi* (BP/1 2575, NMQR 3763).

Ventral to the fossa, the scapulocoracoid is marked by finished bone surface. This surface continues posteroventral to the glenoid fossa as an anteroposteriorly elongate posterior process of the coracoid. Similar processes occur in *Araeoscelis gracilis* (MCZ 8828; *Reisz, Berman & Scott, 1984*), *Youngina capensis* (AMNH FARB 5561), and *Trilophosaurus buettneri* (TMM 31025-140; *Gregory, 1945*). The bone anteroventral to the glenoid fossa is fractured into small fragments, such that the boundaries between the scapulocoracoid and elements further anteriorly (clavicle, cleithrum) cannot be discerned.

Anterodorsal to the scapular blade, there is an anteroposteriorly slender and dorsoventrally tall blade of bone that we interpret as the left **cleithrum** based on morphology and position (Fig. 7). The cleithrum is displaced strongly anterodorsally relative to the anterodorsal margin of the blade. The bone curves strongly anterodorsally, with a prominently concave anterior surface. The lateral surface of the cleithrum is made up of finished bone marked by slender, dorsoventrally extending ridges. Ventrally, the lateral surface of the cleithrum is poorly preserved, with little intact lateral surface. The bone tapers slightly at its ventral tip.

The cleithrum is absent in nearly all known Permo-Triassic diapsid reptiles. It is present in *Coelurosauravus elivensis* (MNHN.F.MAP 327; *Carroll, 1978*), *Araeoscelis gracilis* (*Vaughn, 1955*; *Reisz, Berman & Scott, 1984*), *Petrolacosaurus kansensis* (*Reisz, 1981*), and possibly *Hovasaurus boulei* (*Currie, 1981a*) and *Acerosodontosaurus piveteaui* (*Bickelmann, Müller & Reisz, 2009*). A distinct cleithrum is absent in *Drepanosaurus* spp. (MCSNB 5728, GR 113; *Renesto, 1994a*; *Harris & Downs, 2002*) *Youngina capensis* (BP/1 3859; *Gow, 1975*), *Protorosaurus speneri* (*Gottmann-Quesada & Sander, 2009*),

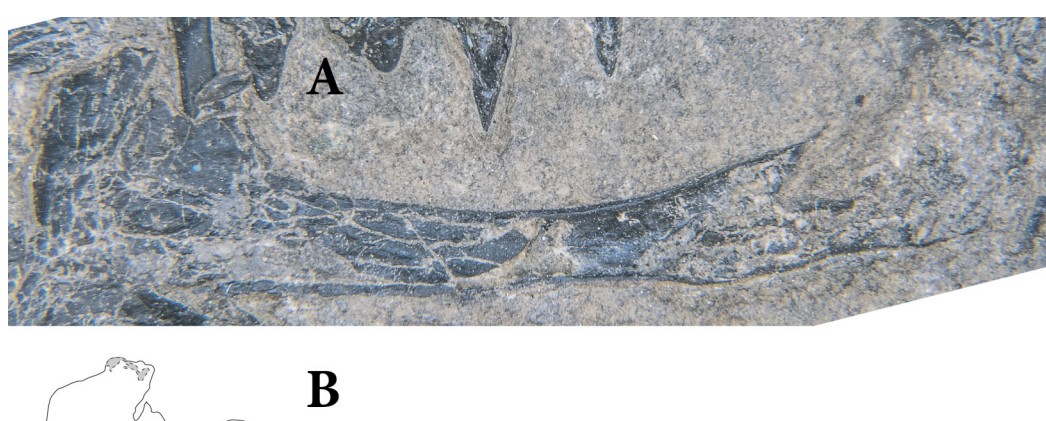

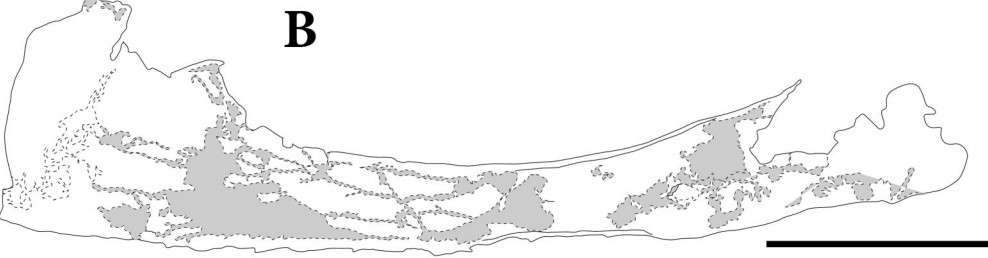

**Figure 8 The right humerus of SMNK-PAL 2882 (*Weigeltisaurus jaekeli*) in dorsal/extensor view.** (A) photograph of specimen. (B) Interpretative drawing demarcating the margins of elements and the cracks and gaps within individual bones and identification callouts. Scale bar equals 1 cm.

*Clevosaurus hudsoni* (NHMUK R36832; *O'Brien, Whiteside & Marshall, 2018*), and *Eusaurosphargis dalsassoi* (*Scheyer et al., 2017*).

A probable **clavicle** is preserved ventral to the cleithrum described above (Fig. 7). It sits anteriorly of the blade of the scapula, but it is partially articulated to the anterior margin of the coracoid further ventrally. The bone bears an anterodorsally curved shaft that is heavily weathered at its dorsal tip. Further ventrally, it passes deep to several unidentified fragments that conceal its complete shape. The slenderness and shape of the bone resembles those of the clavicle of *Coelurosauravus elivensis* (MNHN.F.MAP 327; *Carroll, 1978*), *Youngina capensis* (*Gow, 1975*), and *Prolacerta broomi* (NMQR 3763; *Gow, 1975*).

## Humerus

Both humeri are preserved largely intact in SMNK-PAL 2882, with their proximal ends exposed in extensor view (Figs. 8, 9). The proximal ends and humeral shafts are preserved, but the distal articular surfaces and epicondyles are weathered away. Both humeri are broken open distal to the midshaft. They are hollow inside, and the cortical bone is extremely thin (thickness < 1 mm).

As preserved, the proximal end of the humerus is expanded to approximately three times the breadth at midshaft. The proximal thirds of the left and right humeri are exposed in extensor view and heavily crushed. The humeral head is convex proximally. There is no clear internal tuberosity raised beyond the humeral head, similar to the condition in *Araeoscelis gracilis* (*Vaughn, 1955*), *Claudiosaurus germaini* (SAM-PK 8580), and

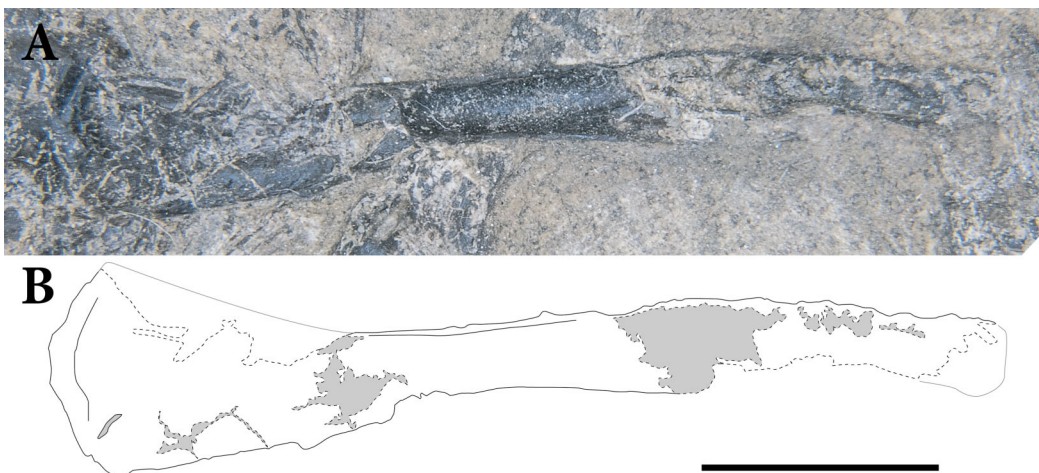

**Figure 9 The left humerus of SMNK-PAL 2882 (*Weigeltisaurus jaekeli*) in lateral view.** (A) photograph of specimen. (B) Interpretative drawing demarcating the margins of elements and the cracks and gaps within individual bones and identification callouts. Scale bar equals 1 cm.

*Youngina capensis* (BP/1 3859; *Gow, 1975*). The preserved bone surface on the extensor surface of the proximal humerus is smooth.

The humerus tapers distally, achieving a breadth one third that of the proximal end at approximately one-third the length of the humerus. It maintains this breadth for the central one-third of the length of the humerus, broadening again at the distal one-third the length of the bone. The elongate slender shaft and limited expansion of the proximal and distal ends compare favorably to *Coelurosauravus elivensis* (MNHN.F.MAP327; *Carroll, 1978*), *Rautiania* sp. (PIN 5130/54; *Bulanov & Sennikov, 2010*), and *Araeoscelis gracilis* (*Vaughn, 1955*). The humerus is relatively more robust, with more expanded ends and a proportionally shorter shaft, in *Thadeosaurus colcanapi* (MNHN.F.MAP360; *Carroll, 1981*), *Youngina capensis* (BP/1 3859; *Gow, 1975*), and *Protorosaurus speneri* (*Gottmann-Quesada & Sander, 2009*).

In SMNK-PAL 2882, the left humerus preserves its central one-third largely uncrushed (Fig. 9). This uncrushed portion does not bear any prominent crests or ridges. The central one-third of the right humerus is cracked open, exposing the medullary cavity and the submillimeter-thick cortical bone (Fig. 8). The exposed cavity indicates that the humeri were subcircular in cross-section at midshaft. The distal one-third of the humerus expands relative to midshaft. Very little morphology can be seen due to weathering. It is not clear if the humeri in SMNK-PAL 2882 bore the distally expanded entepicondyle characteristic of *Coelurosauravus elivensis* (MNHN.F.MAP317; *Carroll, 1978*) and *Rautiania* sp. (PIN 5130/54).

## Radius

The radius is a simple bone, consisting of an elongate shaft with an ovoid cross-section and small proximal and distal expansions. The right radius preserves only a faint outline of its proximal third (Fig. 10). The central third of the bone is uncrushed, but the distal end

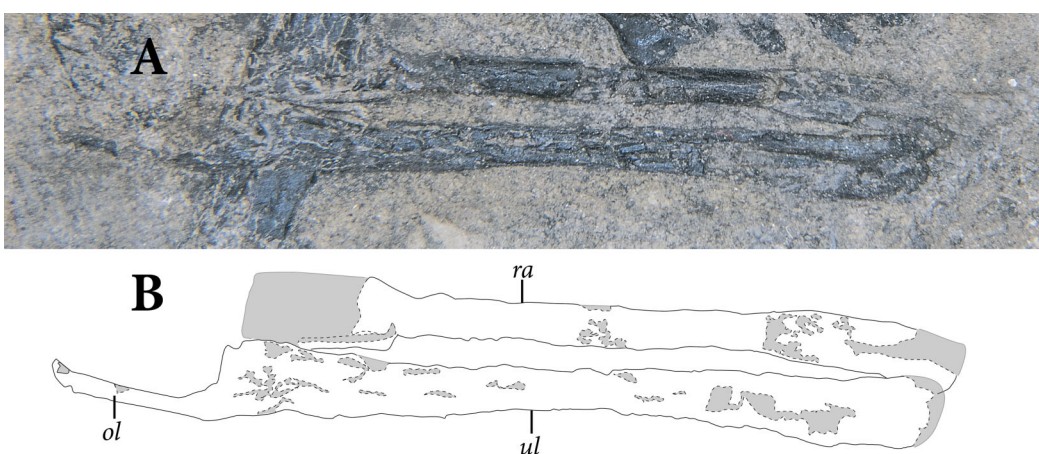

**Figure 10 The right radius and ulna of SMNK-PAL 2882 (*Weigeltisaurus jaekeli*) in lateral view.**
(A) photograph of specimen. (B) Interpretative drawing demarcating the margins of elements and the cracks and gaps within individual bones and identification callouts. Abbreviations: **ol**, olecranon process; **ra**, radius; **ul**, ulna. Scale bar equals 1 cm.

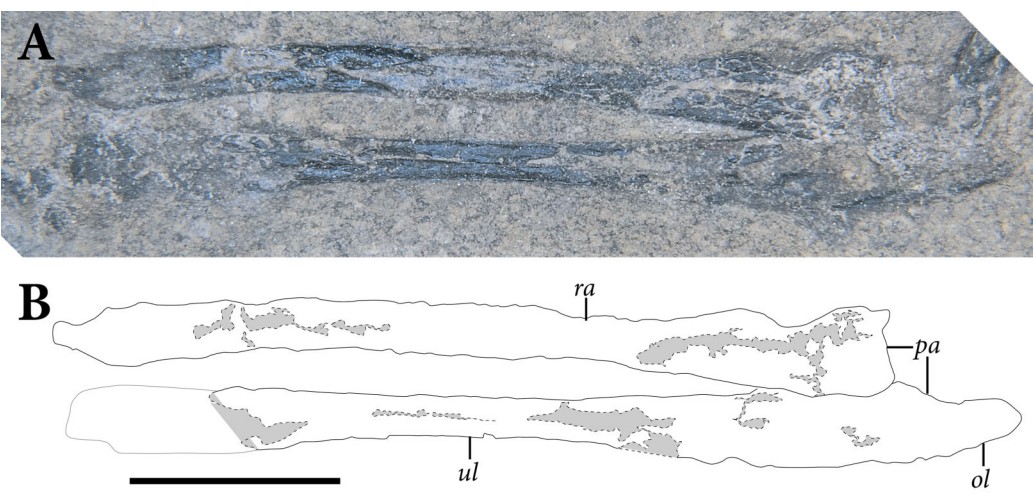

**Figure 11 The left radius and ulna of SMNK-PAL 2882 (*Weigeltisaurus jaekeli*) in lateral view.**
(A) photograph of specimen. (B) Interpretative drawing demarcating the margins of elements and the cracks and gaps within individual bones and identification callouts. Abbreviations: **ol**, olecranon process; **pa**, proximal articular surface; **ra**, radius; **ul**, ulna. Scale bar equals 1 cm.

is heavily crushed and weathered. The left radius is more complete, preserving nearly the full length of the bone, but it is heavily cracked and crushed throughout its length (Fig. 11). The midshaft region and the distal end are heavily weathered, obscuring their original morphology.

The shaft of the radius is straight, lacking the sinusoidal curvature present in some early diapsids (e.g., *Thadeosaurus colcanapi*, MNHN.F.MAP360 *Trilophosaurus buettneri*, TMM 31025-140). The straight shaft more closely resembles the radii of the holotype of *Weigeltisaurus jaekeli* (SSWG 113/7), *Coelurosauravus elivensis* (MNHN.F.MAP327;

*Carroll, 1978*), and *Megalancosaurus preonensis* (MFSN 1769; *Renesto, 1994b*). The proximal end of the radius is subtly expanded relative to the midshaft, being 1.5 times wider. The radial shaft is smoothly textured without any prominent crests or tubers. At its distal end, the bone expands subtly relative to midshaft. The distal articular surface appears to be slightly convex without a complex, screw-shaped articulation. Such a complex, screw-shaped articulation occurs in *Drepanosaurus* sp. (GR 737; *Pritchard et al., 2016*), and squamates such as *Iguana iguana* (*Gauthier, Estes & de Queiroz, 1988*; *Russell & Bauer, 2008*). In SMNK-PAL 2882, it is not clear which carpals the radius met at its distal end, as the carpi are not preserved.

## Ulna

Both **ulnae** are complete (Figs. 10, 11). However, the proximal and distal articular surfaces and the olecranon processes are heavily weathered. The shafts of both ulnae are crushed. They remain in articulation with both the radii and humeri, likely in their original articulations.

The olecranon process of the ulna is elongate and straight, extending well beyond the ulnar articulation with the humerus (Fig. 11). The apex of this process is not upturned. Similarly elongate olecranon processes extending beyond the margins of the humeral articulation occur in *Coelurosauravus elivensis* (MNHN.F.MAP317; *Carroll, 1978*), *Captorhinus aguti* (*Fox & Bowman, 1966*), *Petrolacosaurus kansensis* (*Reisz, 1981*), and *Megalancosaurus preonensis* (MPUM 6008; *Renesto et al., 2010*). In *Thadeosaurus colcanapi* (MNHN.F.MAP360; *Carroll, 1981*), *Trilophosaurus buettneri* (TMM 31025-140; *Gregory, 1945*), *Azendohsaurus madagaskarensis* (*Nesbitt et al., 2015*), and many early lepidosaurs (*Cocude-Michel, 1963*; *Simões et al., 2017*) the olecranon does not extend beyond the defined margins of the humeral articulation. The olecranon process is poorly developed in *Youngina capensis* (BP/1 3859; *Gow, 1975*), *Claudiosaurus germaini* (*Carroll, 1981*), and *Macrocnemus bassanii* (PIMUZ T/4355; *Peyer, 1937*).

The shaft of the ulna is expanded to its greatest diameter just distal to the articulation with the humerus. Just distal to the humeral articulation, the flexor surface of the right ulna contacts the proximal end of the radius for a short length. Distal to the radioulnar articulation, the ulna tapers slightly.

The ulnar shaft is straight, retaining roughly the same diameter throughout its length. In its distal fourth, the shaft expands slightly. The distal end of the ulna is heavily eroded on the left side, but the right retains some of its original shape (Fig. 11). The distal articular surface was slightly convex. On the right side, the radius and ulna are in contact at their distal tips (Fig. 10). However, this may be a preservational artifact. In the absence of carpals, we cannot determine which carpals contacted the ulna.

## Carpus

Both **carpi** are poorly preserved. None of the left carpus is visible, whereas four poorly preserved clumps of bone are present on the right (Fig. 12). These lack defined boundaries, such that their morphology cannot be discerned. The largest clump is positioned several millimeters distal to the radius and just proximal to the first, second, and third

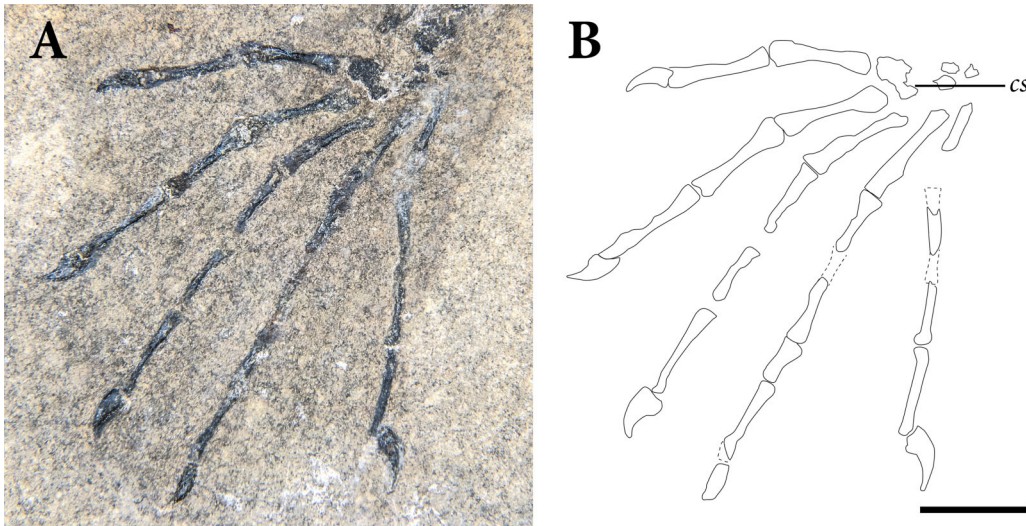

**Figure 12** **The left manus of SMNK-PAL 2882 (*Weigeltisaurus jaekeli*) in dorsal view.** (A) photograph of specimen. (B) Interpretative drawing demarcating the preserved margins of elements. Abbreviations: **cs**, preserved bones of carpus, **mc**, metacarpal. Scale bar equals 1 cm.

metacarpals. It may represent portions of distal carpals 1, 2, and 3. The holotype of *Weigeltisaurus jaekeli* (SSWG 113/7) has two or three proximal carpals and five distal carpals (*Evans & Haubold, 1987*).

## Manus

The left and right **manus** are well preserved (Figs. 12, 13). The left preserves all metacarpals and phalanges (Fig. 12), whereas the right preserves everything but the phalanges of the first digit (Fig. 13). Both hands are exposed in roughly dorsal view, although certain digits are twisted into lateral and medial views. Although many metacarpals and phalanges are weathered, the hands provide excellent detail concerning the structure and proportions in the manual digits.

The manual phalangeal formula of *Weigeltisaurus jaekeli* is 2-3-4-5-4 based on SMNK-PAL 2882 and SSWG 113/7. The same formula occurs in a probable manus of *Rautiania* sp. (PIN 5130/10; *Bulanov & Sennikov, 2010*). This is one more phalanx in the fifth digit than in the manus of most early eureptiles and diapsids; a formula of 2-3-4-5-3 occurs in *Labidosaurus hamatus* (*Sumida, 1989*), *Hovasaurus boulei* (*Currie, 1981a*), *Vallesaurus cenensis* (MCSNB 4751; *Renesto & Binelli, 2006*), *Protorosaurus speneri* (*Gottmann-Quesada & Sander, 2009*), and *Macrocnemus fuyuanensis* (GMPKU-P-3001; *Jiang et al., 2011*). It may represent a synapomorphy of Weigeltisauridae.

The metacarpals are best preserved in the left hand of SMNK-PAL 2882, where they radiate outwards from the distal end of the carpus (Fig. 12). The proximal articular surfaces of the metacarpals do not appear to overlap one another, although this may be due to post-mortem disarticulation. The proximal articular surfaces of the metacarpals do overlap in the holotype of of *Weigeltisaurus jaekeli* (SSWG 113/7; *Evans & Haubold, 1987*), *Petrolacosaurus kansensis* (*Reisz, 1981*), *Protorosaurus speneri* (*Gottmann-Quesada &*

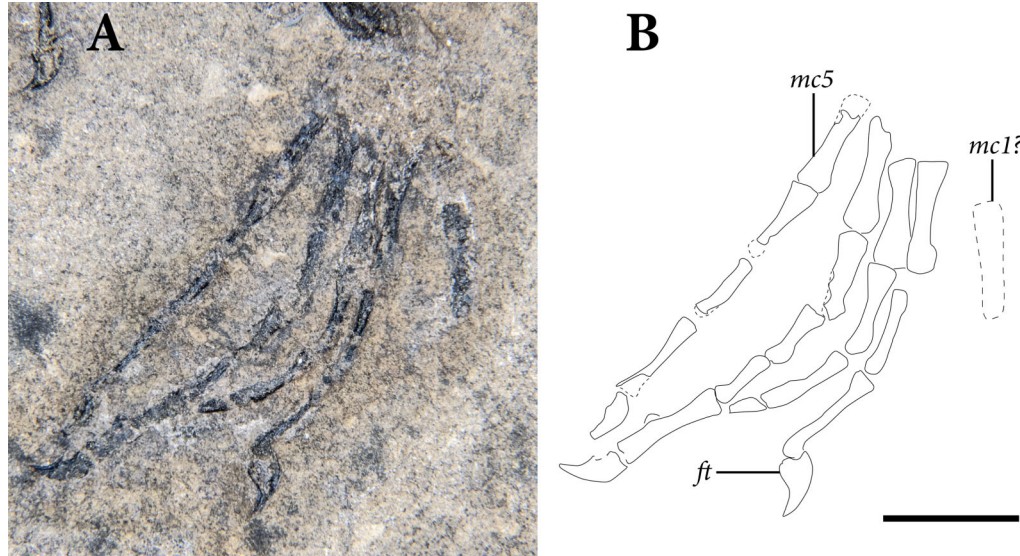

**Figure 13 The right manus of SMNK-PAL 2882 (*Weigeltisaurus jaekeli*) in dorsal view.** (A) photograph of specimen. (B) Interpretative drawing demarcating the preserved margins of elements. Abbreviations: **ft**, flexor tubercle of manual ungual, **mc**, metacarpal. Scale bar equals 1 cm.

*Sander, 2009*), *Mesosuchus browni* (SAM-PK 6046; *Dilkes, 1998*), and *Iguana iguana* (*Russell & Bauer, 2008*).

In SMNK-PAL 2882, each metacarpal is broad proximally, tapering to a slender shaft. The shafts expand slightly at their distal ends, terminating in a distally convex articulation for the proximal phalanges. The proximodistal lengths of metacarpals I through III increase sequentially, with the fourth metacarpal being roughly the same length as the third. The same relative lengths occur in the holotype of *Weigeltisaurus jaekeli* (SSWG 113/7) and the manus referred to *Rautiania* sp. (PIN 5130/10; *Bulanov & Sennikov, 2010*). Although its exact length cannot be determined, the fifth metacarpal is relatively shorter than the fourth. Similar relative metacarpal proportions also occur in *Thadeosaurus colcanapi* (MNHN.F.MAP360; *Carroll, 1981*), *Hovasaurus boulei* (*Currie, 1981a*), and *Protorosaurus speneri* (*Gottmann-Quesada & Sander, 2009*).

The non-ungual phalanges are relatively uniform in shape (Figs. 12, 13). Each has slightly expanded proximal and distal ends and an elongate, tapered shaft. The fifth digit of the left hand and the first through fourth digits of the right are exposed in lateral view, such that a slight bowing of the palmar surfaces of each phalanx may be seen. The relative lengths of the non-terminal phalanges closely resemble the condition in *Rautiania* sp. (PIN 5130/10; *Bulanov & Sennikov, 2010*). In each digit, the penultimate phalanx is the longest proximodistally. Similar relative elongation of the penultimate phalanges occurs in *Megalancosaurus preonensis* (MFSN 1769; *Renesto, 1994b*), *Trilophosaurus buettneri* (TMM 31025-140; *Gregory, 1945*), and the first manual digit (the only one completely preserved) of *Icarosaurus siefkeri* (AMNH FARB 2101; *Colbert, 1970*). Similar proportions occur in some extant arboreal squamates, such as *Draco dussumieri* (FLMNH Herp

19920, ark:/87602/m4/M36111) and *Plica plica* (CAS HERP 231777, ark:/87602/m4/M74709).

The unguals are relatively proximodistally shorter than any non-ungual phalanges and metacarpals (Figs. 12, 13). The manual unguals all are similar in proximodistal length. The shape is best preserved in the second and third digits of the left manus and the second digit of the right hand. Each ungual bears a strong curvature towards the palmar surface of the hand and a large ventral flexor tubercle.

The articular facet for the penultimate phalanx is positioned at the proximodorsal margin of the ungual. Its dorsal margin is framed by a proximodistally short, proximally projecting process. There is a dorsoventrally short, flattened surface ventral to the phalangeal articular surface. Proximoventrally, each well-preserved ungual bears a prominent, ventrally convex flexor tubercle. The flexor tubercle is similarly rounded, pendent, and proximally positioned in *Protorosaurus speneri* (*Gottmann-Quesada & Sander, 2009*), *Trilophosaurus buettneri* (TMM 31025; *Gregory, 1945*), and *Azendohsaurus madagaskarensis* (*Nesbitt et al., 2015*).

The well-preserved examples taper prominently throughout their proximodistal lengths, with the distal tip positioned ventrally relative to the flexor tubercle. A distinct lateral groove is present on the lateral surfaces of the best preserved unguals. It extends distoventrally along the ventrolateral margin of the bone, from the level of the flexor tubercle to the distal tip. The arcing of the distal tips of the unguals compares favorably with *Trilophosaurus buettneri* (TMM 31025-140; *Gregory, 1945*) and *Azendohsaurus madagaskarensis* (*Nesbitt et al., 2015*). It contrasts with the relatively straight unguals in *Thadeosaurus colcanapi* (MNHN.F.MAP360; *Carroll, 1981*), *Boreopricea funerea* (PIN 3708/1), and "*Chasmatosaurus*" *yuani* (IVPP 4067).

## Pelvis

As is the case with the pectoral girdle, the **pelvis** is heavily crushed and difficult to interpret (Fig. 14). Unfortunately, there are few other weigeltisaurid specimens that preserve pelves in articulation to use for comparisons. Our interpretation of this region is tentative, largely based on the position of the femora relative to the crushed pelvic bones.

The proximal articular surfaces of both femora sit lateral to trapezoidal bones with dorsally expanded margins that we interpret as ilia (Fig. 14). Both are broken ventrally and exposed in medial view. The medial surfaces are all heavily weathered, such that no articular surfaces for the sacral ribs nor other pelvic elements can be discerned. Both bones bear a distinct, dorsolateral concavity positioned dorsal to the femoral articulation. The anterior tips of the iliac blades and likely most of the pubes are covered by clusters of the left patagial ossifications.

Under our interpretation, both ischia are exposed in dorsal view (Fig. 14). The ischial symphysis is preserved as a somewhat sigmoid line, the anterior tip of which may be seen posterior to the left ilium. The symphysis curves slightly towards the right ilium along its length. The left ischium is displaced further posteriorly than the right.

A small, posterolateral process with a posteriorly convex margin is present posterior to the acetabulum on both sides. We interpret these as posterolateral processes comparable to

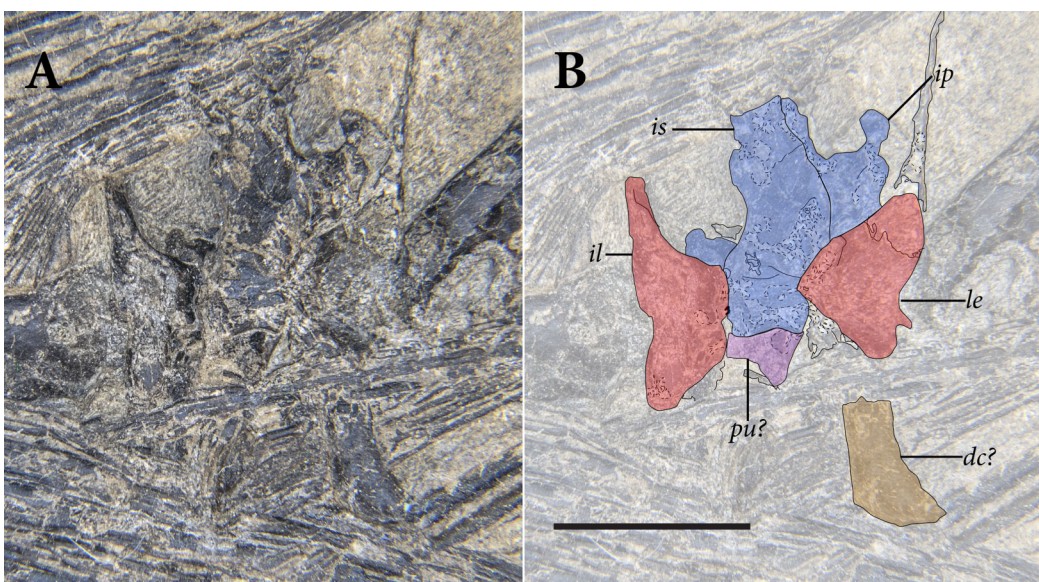

**Figure 14 The pelvic region of SMNK-PAL 2882 (*Weigeltisaurus jaekeli*) in dorsal view.** (A) photograph of specimen. (B) Reduced opacity photograph of specimen with interpretive lines demarcating the preserved margins of elements. Abbreviations: **dc**, dorsal vertebral centrum; **il**, ilium; **ip**, posterior process of ischium; **is**, ischium; **le**, dorsolateral embayment of iliac blade; **pu**, pubis. Scale bar equals 1 cm.

similar processes in sphenodontians (e.g., *Carroll, 1985*; *Fraser, 1988*) and drepanosauromorphs (e.g., *Renesto, 1994a*; *Colbert & Olsen, 2001*). Posteromedially, the ischia are drawn out into elongate posteromedial processes that form much of the anteroposterior lengths of the bones. This conformation of processes in SMNK-PAL 2882 is quite similar to the ischium of the stem-turtles *Pappochelys rosinae* (e.g., *Schoch & Sues, 2018b*) and *Eorhynchochelys sinensis* (e.g., *Li et al., 2018*).

Only one weigeltisaurid specimen preserves the pelvis in mostly lateral view: a skeleton of *Coelurosauravus elivensis* (MNHN.F.MAP327; *Carroll, 1978*; *Evans & Haubold, 1987*). In that specimen, only the right ilium and right pubis are clearly exposed. The right ischium is overlain by the proximal portion of the right femur, which remains in articulation with the acetabulum. The ilium is trapezoidal, broadening from the level of the acetabulum to a mostly flat dorsal margin. As a result, there are prominent anterior and posterior processes of the ilium that taper anterodorsally, a shape similar to the crushed ilium of a specimen of *Weigeltisaurus jaekeli* described by *Evans (1982)*. These processes compare favorably with the dorsally expanding blade of what we consider the ilium in SMNK-PAL 2882. The inverted trapezoid shape of the iliac blade resembles the condition in *Hyperodapedon gordoni* (*Benton, 1983*) and *Teraterpeton hyrnewichorum* (NSM 018GF010.002; *Pritchard & Sues, 2019*). It contrasts sharply with the posterodorsally tapered, anteriorly flattened shape of the iliac blades of *Araeoscelis gracilis* (*Vaughn, 1955*), *Thadeosaurus colcanapi* (MNHN.F.MAP360; *Carroll, 1981*), and *Youngina capensis* (BP/1 3859; *Gow, 1975*). The ilium of MNHN.F.MAP327 bears an apparent embayment along its dorsolateral surface that also compares favorably with the morphology of the proposed ilia from SMNK-PAL 2882.

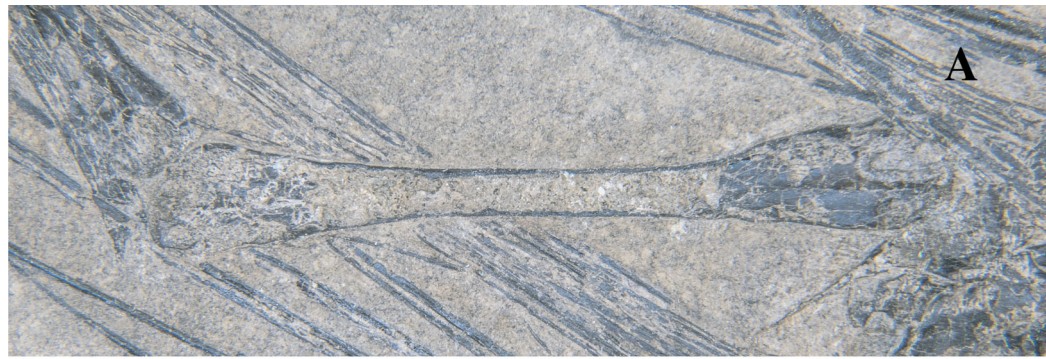

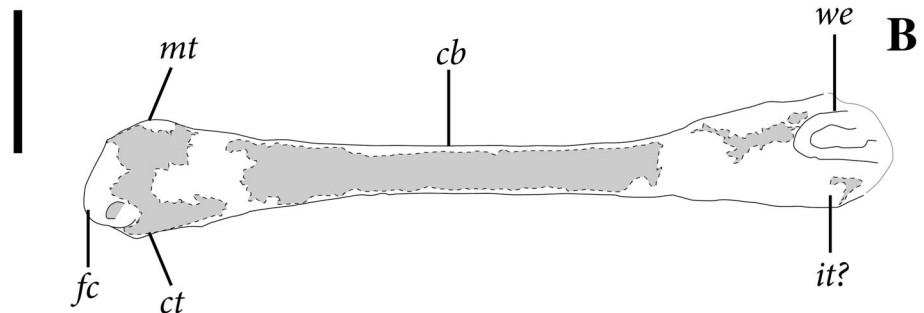

**Figure 15 The left femur of SMNK-PAL 2882 (*Weigeltisaurus jaekeli*) in extensor view.** (A) photograph of specimen. (B) Interpretative drawing showing the outline of the femur. Abbreviations: **cb**, cortical bone of the femoral shaft; **ct**, crista tibiofibularis/lateral tibial condyle; **fc**, fibular condyle; **it**, internal trochanter; **mt**, medial tibial condyle; **we**, weathered margin. Scale bar equals 1 cm.

## Femur

The left femur is nearly complete (Fig. 15). A portion of the cortical bone of the left shaft is missing, as is a section of the femoral head. The proximal and distal ends are somewhat crushed. The proximal end of the right femur is preserved in articulation with the pelvis (Fig. 2). It lies dorsal to the posteriormost seven patagial ossifications. The shaft and distal end of the right femur are completely missing.

The proximal end of the left femur is positioned in articulation with the crushed left pelvic girdle (Fig. 15). Its head is convex, and the posterior margin is heavily weathered. A proximally convex femoral head occurs in the holotype of *Weigeltisaurus jaekeli*, *Coelurosauravus elivensis* (MNHN.F.MAP327; *Carroll, 1978*), *Araeoscelis gracilis* (MCZ 4360; *Vaughn, 1955*), *Hypuronector limnaios* (AMNH FARB 7759), *Vallesaurus cenensis* (MCNSB 4751), and *Clevosaurus hudsoni* (NHMUK PLR 600a). Proximally flattened or slightly concave proximal articular surfaces occur in *Thadeosaurus colcanapi* (MNHN. F.MAP360, *Carroll, 1981*), *Youngina capensis* (BP/1 3859), *Prolacerta broomi* (BP/1 2676; *Gow, 1975*), and *Pamelaria dolichotrachela* (*Sen, 2003*).

On the left femoral head of SMNK-PAL 2882, weathering has formed a prominent depression in its posterior edge, framed by a ring of cortical bone. The internal trochanter forms a prominent crest that extends ventrally from the femoral head on the left side. At its proximal end of the femur in SMNK-PAL 2882, the crest is equivalent in dorsoventral

depth to the femoral head. It decreases in height further distally, becoming indistinguishable from the femoral shaft one quarter the length down the shaft. A similarly short internal trochanter occurs in *Coelurosauravus elivensis* (MNHN.F.MAP 17). The internal trochanter remains distinct for a greater portion of the shaft in *Araeoscelis gracilis* (*Vaughn, 1955*), *Thadeosaurus colcanapi* (MNHN.F.MAP360), and *Prolacerta broomi* (BP/1 2676; *Gow, 1975*).

The femoral shaft is elongate and slender, consistently maintaining a narrow diameter for approximately one-half its total length (Fig. 15). The posterodorsal surface of the femoral shaft is largely weathered away, exposing the medullary cavity. Similar to the humeral shaft, the cortical bone of the femur is extremely thin (<1 mm) throughout the exposed length of the shaft. Due to this weathering, any crests or muscle attachments on the femoral shaft are not preserved. Similarly thin cortical bone occurs in the forelimb bones of *Drepanosaurus* sp. (*Pritchard et al., 2016*), extant flighted birds (*Sullivan et al., 2017*), and pterosaurs (*De Ricqlès et al., 2000*).

The shaft of the femur expands in diameter three quarters the length down the shaft, expanding into the distal articular surface (Fig. 15). The distal end is exposed in posterodorsal view. Although the dorsal surface of the distal femur is heavily weathered, it seems to lack a proximodistally extending depression separating the lateral and medial condyles. A similar condition is present in *Thadeosaurus colcanapi* (MNHN.F.MAP360; *Carroll, 1981*), *Prolacerta broomi* (BP/1 2676; *Gow, 1975*), and *Clevosaurus hudsoni* (NHMUK PLR 600a). A prominent depression occurs between the condyles of *Captorhinus aguti* (*Fox & Bowman, 1966*), *Araeoscelis gracilis* (*Vaughn, 1955*), and *Petrolacosaurus kansensis* (*Reisz, 1981*).

The distal end of the left femur in SMNK-PAL 2882 may be divided into three distinct condyles: the medial tibial, lateral tibial, and fibular condyles. The fibular and lateral tibial condyles are closely associated, comprising much of the posterior portion of the distal end of the femur. These two condyles notably extend distally relative to the medial tibial condyle. Similar prominence of the lateral two condyles occurs in *Coelurosauravus elivensis* (MNHN.F.MAP325), *Petrolacosaurus kansensis* (*Reisz, 1981*), *Zarcasaurus tanyderus* (CM 41704; *Brinkman, Berman & Eberth, 1984*), and *Vallesaurus cenensis* (MCSNB 4751). In contrast, the femoral condyles are similarly distally expressed in *Claudiosaurus germaini* (SAM-PK 8266), *Prolacerta broomi* (BP/1 2676; *Gow, 1975*), and *Clevosaurus hudsoni* (NHMUK PLR 600a; *Fraser, 1988*).

The posterior condylar surface is marked by a prominent, distally rounded crest that we identify as the fibular condyle. The central portion of the fibular condyle is marked by a prominent depression. An additional condyle—the lateral tibial condyle—sits ventral to the fibular condyle. It appears to be similar in dorsoventral height to the fibular condyle, but little can be said of its morphology due to surficial weathering. The medial tibial condyle is also heavily weathered. Its distal margin is subtly convex.

## Tibia

The full proximodistal length of the left tibia is preserved, although the bone is heavily crushed and much of the cortical bone of the tibial shaft has weathered away

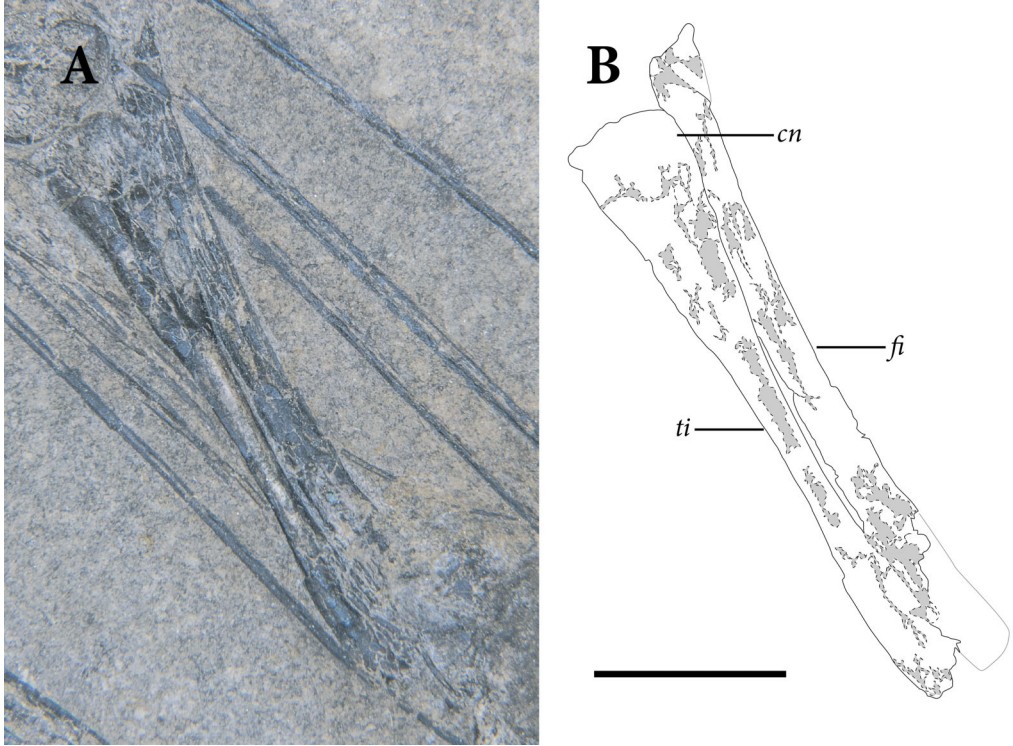

**Figure 16 The distal portion of the left hindlimb of SMNK-PAL 2882 (*Weigeltisaurus jaekeli*) in extensor view.** (A) photograph of specimen. (B) Interpretative drawing demarcating the preserved margins of elements. Abbreviations: **cn**, cnemial crest; **fi**, fibula; **ti**, tibia. Scale bar equals 1 cm.

(Fig. 16). The distal end of the right tibia is exposed and well preserved; the proximal end of the bone is buried beneath the bundle of patagial ossifications from the right side of the body.

The tibia is overall straight and slender, with expansions proximally and distally (Fig. 16). The proximal end of the bone is only visible on the left tibia, and it is both crushed and weathered. The exposed portion of the bone is not as transversely broad as the distal end of the femur, which could be a product of distortion or of some portion of the tibia remaining concealed in the rock. On the left hindlimb, the exposed proximal tibial articular surface is flat and positioned close to the medial tibial condyle of femur. There is a small cnemial eminence on the anterior surface of the proximal tibia, similar to the condition in *Thadeosaurus colcanapi* (MNHN.F.MAP360; *Carroll, 1981*), *Prolacerta broomi* (BP/1 2676; *Gow, 1975*), and *Clevosaurus hudsoni* (NHMUK PLR 600a; *Fraser, 1988*). This cnemial crest does not extend proximally as a prominent tuberosity, as occurs in *Captorhinus aguti* (*Fox & Bowman, 1966*) and *Zarcasaurus tanyderus* (CM 41704; *Brinkman, Berman & Eberth, 1984*). In SMNK-PAL 2882, the left tibia remains in articulation with the proximal end of the left fibula. The exposed bone of the extensor surface of the tibia is smooth, without any obvious crests or muscle scars.

Distal to the proximal articular surface, the tibia continuously decreases in diameter. One-third the length down the shaft, the diameter of the bone is roughly one third that of

its proximal articular surface. For most of the length of the tibia distal to this point, the cortical bone of the extensor surface is weathered away. The cortical bone of the tibial shaft is <1 mm in thickness, similar to that of the femur and the humerus.

The distal one fifth of the left tibia preserves the outer surface of the bone, although the anterior half of this surface is broken into small bone fragments (Fig. 16). The distal fifth of the bone is roughly two times the diameter of the thinnest portion of the tibial shaft. The preserved bone on the posterior surface of the tibia is smoothly textured. The distal articular surface of the right tibia for the tarsals is too weathered to describe adequately.

The right tibia is compressed but relatively well preserved, with much of the cortical bone intact (Fig. 2). As in the left tibia, the distal fifth expands to be approximately twice the diameter of the shaft. The distal articular surface preserves a distinct step, but it is not clear whether the distally projecting portion of the surface was positioned anteriorly or posteriorly. A similarly stepped distal articular surface occurs in *Araeoscelis gracilis* (MCZ 4173; *Vaughn, 1955*), *Zarcasaurus tanyderus* (CM 47104; *Brinkman, Berman & Eberth, 1984*), and some archosauriforms (*Nesbitt, 2011*). In SMNK-PAL 2882, the tarsus is not preserved, so the relationships of the tibia to more distal elements cannot be discerned. *Evans & Haubold (1987)* described the tarsus of the holotype of *Coelurosauravus elivensis* (MNHN.F.MAP325) as consisting of a distinct astragalus, calcaneum, lateral centrale, and five distal tarsals.

## Fibula

Only the left **fibula** is exposed in SMNK-PAL 2882 (Fig. 16). The right element is concealed under the patagial ossifications from the right side of the body. The left is largely complete, with only the distal tip of the bone being weathered away. It is heavily crushed and exposed mostly in anterior view.

The left fibula appears to retain its articular contacts with both the left femur and tibia. It extends further proximally than the tibia, fitting against the lateral surface of the lateral tibial condyle. The proximal end of the bone is largely flat with a small proximally projecting process on its extensor margin. Distal to the proximal articular surface, the shaft of the fibula does not constrict in diameter. The preserved surface of the bone is smoothly textured.

Near three quarters the length down the shaft, the fibula becomes heavily weathered with only a few splinters of bone remaining to reflect its original shape. The shaft appears to have expanded slightly in diameter at this point. The preserved portion of the fibula terminates at a jagged break well proximal to the distal end of the left tibia. It is not clear whether the fibula was sigmoidally curved or straight-shafted, but the fibulae in *Coelurosauravus elivensis* (MNHN.F.MAP325; *Carroll, 1978*) and the Eppleton weigeltisaurid specimen (TWCMS B5937.1) are straight along their proximodistal lengths.

## Pes

The pedes are preserved in a condition similar to the manus (Figs. 17, 18). As is the case for the carpi, the **tarsi** are not preserved in SMNK-PAL 2882. The space between the distal ends of the tibiae and fibulae and the proximal ends of the metacarpals is devoid of bone.

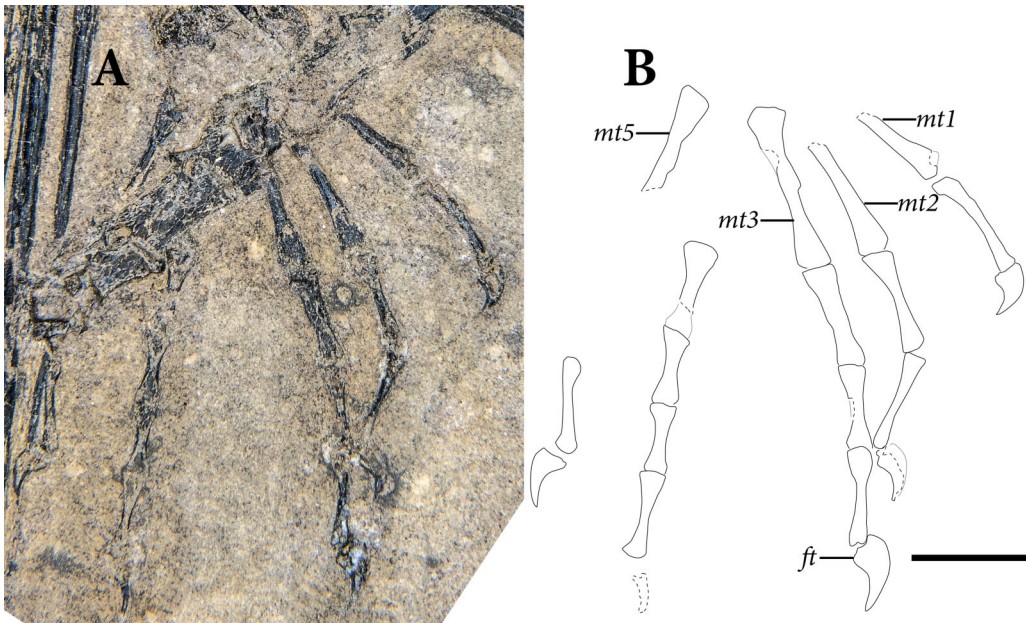

**Figure 17 The right pes of SMNK-PAL 2882 (*Weigeltisaurus jaekeli*) in dorsal view.** (A) photograph of specimen. (B) Interpretative drawing demarcating the preserved margins of elements. Abbreviations: **ft,** flexor tubercle; **mt**, metatarsal. Scale bar equals 1 cm.

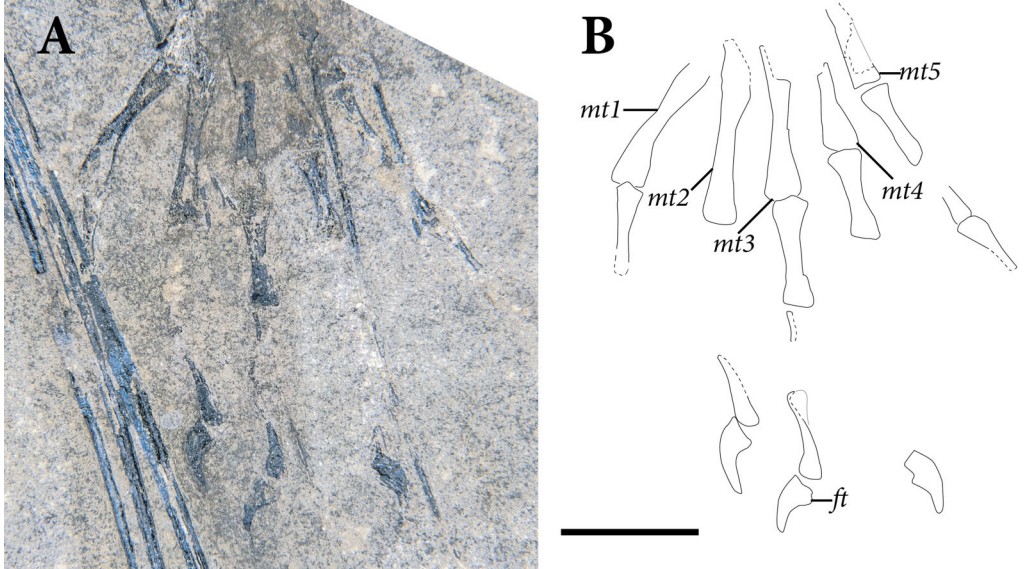

**Figure 18 The left pes of SMNK-PAL 2882 (*Weigeltisaurus jaekeli*) in dorsal view.** (A) photograph of specimen. (B) Interpretative drawing demarcating the preserved margins of elements. Abbreviations: **ft,** flexor tubercle; **mt**, metatarsal. Scale bar equals 1 cm.

There is a prominent, brown discoloration of the matrix at the proximal end of the left foot, which may relate to the original position of the tarsal bones. The tarsus of the holotype of *Weigeltisaurus jaekeli* has a large astragalus, a small calcaneum, a broad

centrale, and five distal tarsals (*Evans & Haubold, 1987*). By contrast, the metatarsals and phalanges are well preserved such that an account of the feet can be given.

The pedal digital formula indicated by SMNK-PAL 2882 is 2-3-4-5-?. An ungual and penultimate phalanx of the fourth digit is preserved in the right foot, and the space between these bones and the metatarsals indicates the presence of either three or four phalanges in this digit (Fig. 17). These observations are consistent with the pedal phalangeal formula of 2-3-4-5-4 observed in the Eppleton skeleton (TWCMS B5937.1; *Evans, 1982*).

The proximal ends of the metatarsals on the left pes are heavily weathered (Fig. 18), and only the third metatarsal is completely preserved on the right foot (Fig. 17). The metatarsals are more robust than the metacarpals, with proportionally larger expansions to the transverse breadth of the bone both proximally and distally. The shaft of the metatarsal makes up approximately two-thirds to the length of the bone and is half the transverse breadth of the proximal and distal expansions.

The proximal margin of the articular surface of a typical metatarsal is mostly flat and transversely oriented (Fig. 17). It is not clear if the proximal ends of the metatarsals overlapped one another, but the bones were close to articulation at their proximal ends in both feet. Distally, the metatarsals expand. The distal articular surfaces for the proximal pedal phalanges are flat. The fifth metatarsal also bears a straight shaft and lacks an "outer process" (sensu *Robinson, 1975*) extending from its proximolateral margin. Hooked metatarsals with outer processes occur broadly in saurian reptiles such as *Prolacerta broomi* (*Gow, 1975*), *Clevosaurus hudsoni* (*Fraser, 1988*), and *Trilophosaurus buettneri* (TMM 31025-140; *Gregory, 1945*).

The pedal phalanges are relatively uniform in shape (Fig. 17). Each is more robust than the manual phalanges, with a slightly more pronounced expansion of the proximal and distal ends. This may be seen best in the proximal phalanges of the third and fourth digits of the left pes and the second, third, and fourth digits of the right foot, all of which are exposed in dorsal view. The expansions of the distal ends of the phalanges become narrower further distally. As in the metacarpals, those metatarsals exposed in either lateral or medial views exhibit a prominent bowing of the ventral surface. This shape is particularly evident in the penultimate phalanges of the first and fifth digits of the right foot.

The pedal ungual phalanges are dorsoventrally deep and taper strongly at their distal tips. Based on the well-preserved unguals of the third pedal digits, most of the proximal surface of the bone formed a slightly concave articular surface for the penultimate phalanx. In all unguals that preserve a complete ventral surface, a prominent, ventrally convex flexor tubercle extends well ventral to the proximal articular surface. Based on the more complete unguals of the right foot, the second through fifth unguals were elongate, each longer than half the proximodistal length of the associated penultimate phalanx. The tapered tip of these unguals extends well ventral to the level of the flexor tubercle. Similar proximal articular surfaces, pendulous flexor tubercles, and ventrally tapering distal tips occur in the pedal unguals of the Eppleton weigeltisaurid specimen (TWCMS B5937.1).

The ungual of the first pedal digit is only preserved on the right foot, and its shape appears to differ from those on the other digits of the same foot. It is much shorter proximodistally, much less than half the length of its respective penultimate phalanx. Its tapering distal tip curves ventrally more abruptly than in the other unguals. A clear lateral groove is not preserved on any of the pedal unguals.

## Gastralia

The venter region of SMNK-PAL 2882 preserves a large number of disarticulated gastralia extending along the ventral margin of the trunk, indicating an extensive gastral basket along nearly the full length of the trunk posterior to the preserved pectoral girdle (Fig. 5). Each gastralium is rod-like and unbranched. Each is at least slightly curved, with some near the mid-length of the trunk being more strongly curved. None of the gastralia bears a midline apex that would suggest an anteriorly pointed median element.

Similarly shaped, disarticulated gastralia occur in specimens of *Coelurosauravus elivensis* (MNHN.F.MAP327) and the Eppleton skeleton (TWCMS B5937.1; *Evans, 1982*). Only the privately owned Wolfsberg specimen of *Weigeltisaurus jaekeli* (described by *Schaumberg, 1976*) preserves the gastral basket in its apparent original position. It suggests a very slender trunk region with a strongly convex venter. We consider it likely that the medial elements of the baskets were unbranched, overlapping rods as in *Proterosuchus alexanderi* (NMQR 1484; *Cruickshank, 1972*), *Postosuchus alisonae* (e.g., *Weinbaum, 2013*), and some dinosaurs (e.g., *Sternberg, 1933*; *Fechner & Gößling, 2014*). As preserved in SMNK-PAL 2882, the gastralia were internal to the ventral bases of the patagial ossifications. In *Coelurosauravus elivensis* (MNHN.F.MAP327) and the Wolfsberg specimen of *Weigeltisaurus jaekeli*, the patagial rods appear to overlap the lateral third of the gastral basket.

## Patagial ossifications

The longest and most prominent bones in the entire skeleton of SMNK-PAL 2882 are the rod-like supporting bones interpreted as supporting a gliding membrane or patagium (Fig. 19). Henceforth, we will refer to these ossifications as "patagials." The patagials on the right side of the body are mostly complete, extending from near the posterolateral margin of the pectoral girdle to nearly the anterior margin of the pelvis. The shafts and distal tips of the left patagials from the anteroposterior midsection of the trunk are preserved. No ventral bases from the left side are preserved. Most of the patagials are crushed transversely and some are weathered to the point where their internal cavities are exposed.

Based on SMNK-PAL 2882, there is no 'standard' patagial in *Weigeltisaurus jaekeli*. There is substantial variation in the patagials throughout the anteroposterior length of the trunk, with typically proximodistally longer, anteroposteriorly thicker rods in the anterior one-third of the trunk region. Further posteriorly, the rods decrease in both proximodistal length and anteroposterior thickness. We estimate a minimum of 24 patagials based on the count in SMNK-PAL 2882, slightly higher than the count of 22 for the same skeleton given by *Frey, Sues & Munk (1997)*. We estimate a count of 29 patagials in *Coelurosauravus*

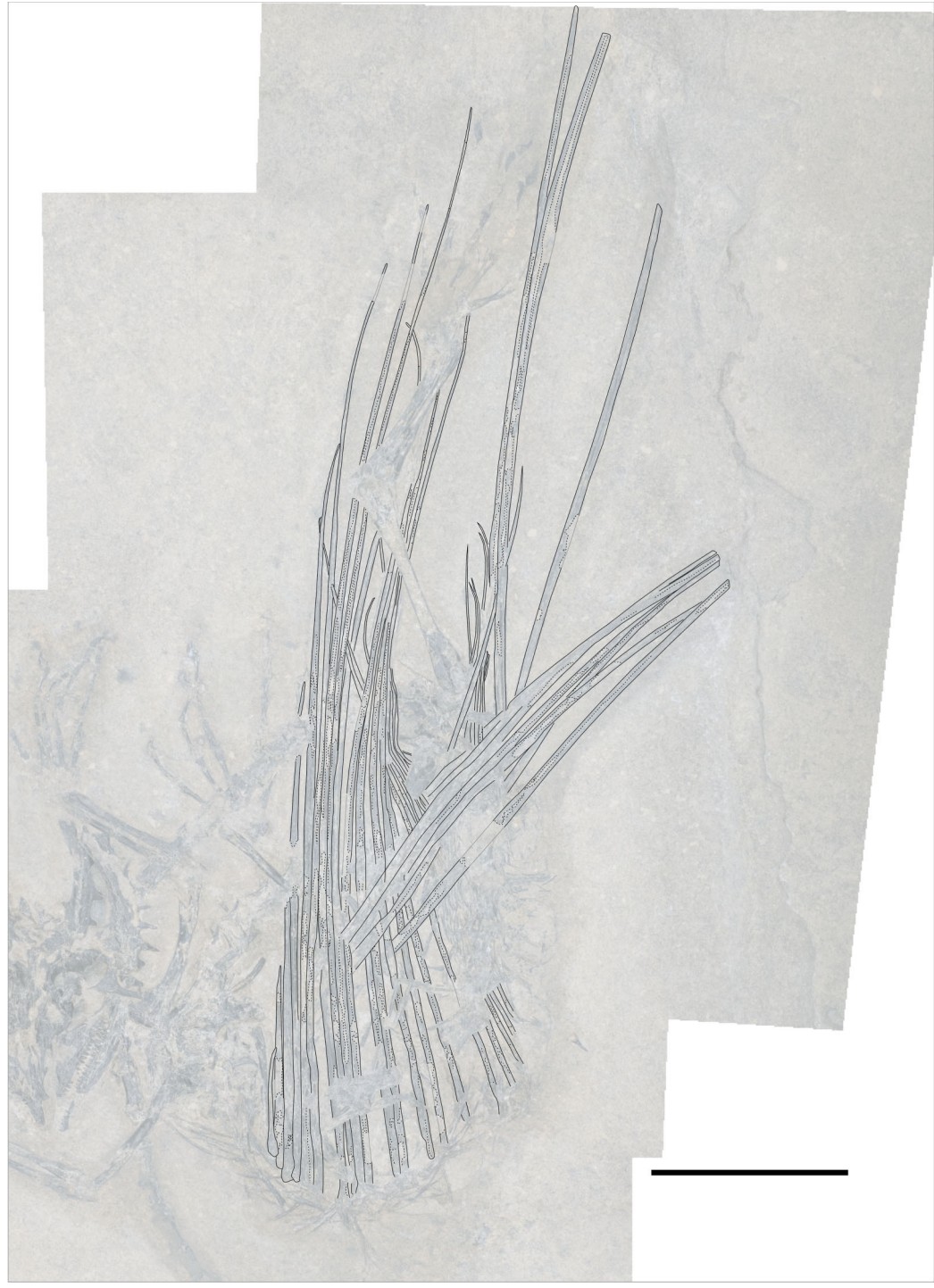

**Figure 19 Patagial spars of SMNK-PAL 2882 (*Weigeltisaurus jaekeli*) in primarily left lateral view.**
The patagial spars have been highlighted with interpretive lines on the reduced opacity image. Hashed lines indicate the positions of broken edges and gaps in the individual patagial elements. Readers may refer to Fig. 2 for an unmodified photograph of the SMNK-PAL 2882 slab. Scale bar equals 5 cm.

*elivensis* (based on MNHN.F.MAP327). Approximately 23 patagials are preserved in the Eppleton specimen, although the anteriormost preserved element is one of the absolute longest; as the longest include the sixth through tenth patagials in SMNK-PAL 2882, the Eppleton specimen would also have at least 28 patagials. Thus, a total patagial count of between 25 and 30 elements appears to be consistent across known Weigeltisauridae.

We will first describe the general morphology of the individual patagials based on the preserved structures on the right and left sides of the body. We will then describe the condition of the individual patagials on SMNK-PAL 2882.

The first patagial arises posterior to the coracoid, its base laying across the lateral surface of the anteriormost part of the gastral basket. It maintains a consistent anteroposterior width throughout the proximal four fifths of its proximodistal length, which is straight. Further distally, the bone curves slightly posteriorly and tapers to a rounded terminus. The first patagial in *Weigeltisaurus jaekeli* compares favorably with that in *Coelurosauravus elivensis* (MNHN.F.MAP327), although the distal tip of the first patagial in that specimen curves slightly anteriorly before curving posteriorly and tapering to its distal tip.

The second patagial is larger, with its proximal base and proximodistal lengths being roughly twice those of the first patagial. The bone tapers subtly from its proximal base to approximately one half the length up the shaft. Further distally, the bone tapers more abruptly to a sharper point than the first patagial. The anterior surface of the distal half of the second patagial is subtly anterodorsally convex, whereas the posterior surface of the distal half is flat.

The third patagial is roughly one quarter longer than the second, although the bone is incompletely preserved on both sides of SMNK-PAL 2882. It bears an anteroposteriorly broad base that tapers abruptly to the width it maintains throughout most of its proximodistal length. The bone tapers subtly from the level of the distal tip of the second patagial to the distal terminus.

The fourth patagial is roughly twice the proximodistal length of the third. The bone maintains its anteroposterior width for most of its proximodistal length to the level of the distal tip of the third patagial. Further distally, the fourth patagial begins to taper subtly. The morphology of the distal terminus is unclear. The shaft of the bone bears a distinct, longitudinal depression throughout much of its proximodistal length. A similar longitudinal depression also occurs in the fifth through 16th patagials.

The fifth patagial is roughly 40% proximodistally longer than the fourth. It maintains its anteroposterior breadth throughout the first half of its proximodistal length, after which it begins to taper gradually. The sixth patagial is roughly 25% proximodistally longer than the fifth. It begins to gradually taper some three fifths up the proximodistal length of the shaft. In the distal fifth of its proximodistal length, the sixth patagial maintains a consistent, anteroposteriorly narrow width (<1 millimeter) to its distal tip. This thin distal portion of the patagial arcs smoothly throughout its length.

The seventh patagial is only slightly longer proximodistally than the sixth. It begins to taper near its proximodistal midpoint. In contrast to the elongate, thin distal segment in

the sixth patagial, the seventh patagial only has a thin, curved segment for a very short length. The thin segment begins at the level of the distal tip of the sixth patagial.

In SMNK-PAL 2882, the eighth patagial appears to be the absolutely longest of all. It is roughly 15% longer than the seventh patagial. It is not clear at what point down the shaft the eight patagial begins to taper. Much as in the sixth patagial, the distal fifth of the eighth is consistently extremely thin throughout its length. This thin segment arcs anteriorly near its distal tip.

The ninth through 14th patagials are all similar in robusticity and anteroposterior breadth to the eighth. The relative lengths of the ninth and 10th patagials are not clear, as they are both buried deep to the left hindlimb. The tenth through fourteenth patagials begin to taper anteroposteriorly at roughly two thirds the length of the eighth patagial (the longest one that can be evaluated). The full lengths of the 10th through 14th patagials cannot be fully evaluated, as they are embedded below the left pes. However, one of these bones is broken near its midpoint, and the distal tip angles anteriorly. It is tapered to a thin splint that appears to terminate at roughly four fifths down the shaft of the eighth patagial.

The 15th through 18th patagials are anteroposteriorly narrower at their bases than the ninth through 14th, being roughly two-thirds as broad as those further anterior patagials. These two begin to taper rapidly near their anteroposterior midpoints, terminating as thin threads. These bones are roughly three fifths the length of the eighth and longest patagial.

Further posteriorly, the remaining patagials are tightly bundled together on the right side of the body, which is the only one that preserves them. We estimate that there are at least six patagials posterior to the 18th for a minimum total of 24. Each of these patagials is anteroposteriorly narrower than the one anterior to it. Their full proximodistal lengths are obscured by other bones on both sides, but they are all certainly shorter than those between the eighth and 18th positions. The bases of these posteriormost patagials are obscured by the proximal end of the right femur and the pelvis, both of which overlie the right wing. However, it is evident that the patagials nearly extended to the posterior edge of the trunk region just anterior to the pelvic girdle. A reconstruction of the skeleton with patagials extended is presented in Fig. 20.

## Phylogenetic analysis

The analysis produced 10 most-parsimonious trees of 1,228 steps recovered in 7,440 out of 10,000 replicates (CI = 0.306, RI = 0.644). All most-parsimonious trees recover Weigeltisauridae as the sister taxon of Drepanosauromorpha in a monophyletic Avicephala (sensu *Senter, 2004*). Avicephala forms a clade with all 'younginiform'-grade diapsids and Sauria (simplified topology presented in Fig. 21). This Avicephala + Sauria clade forms a polytomy with *Orovenator mayorum* and *Claudiosaurus germaini*. See 'Discussion' section for character supports for the major clades in question for this study and comparisons with previous analyses of diapsid phylogeny. Characters with a consistency index greater than 0.5 are marked with an asterisk (*).

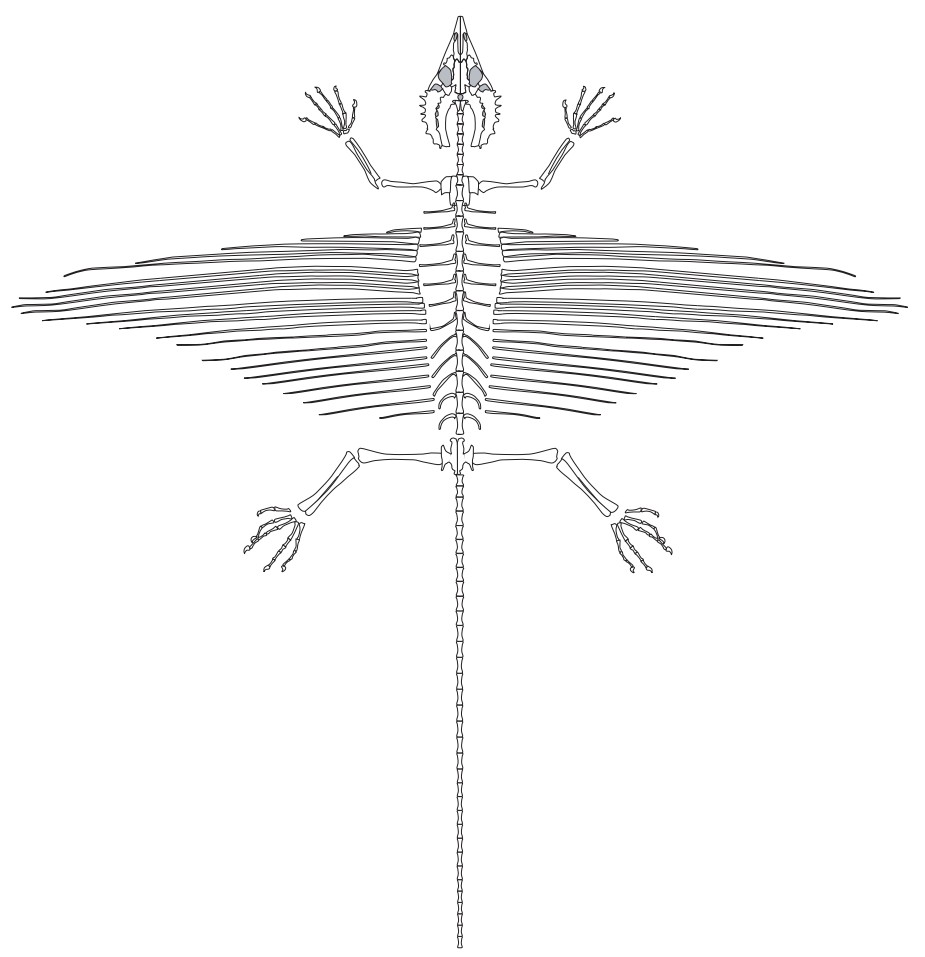

**Figure 20 Schematic reconstruction of the skeleton of *Weigeltisaurus jaekeli* in primarily dorsal view, based on the proportions of skeletal elements in SMNK-PAL 2882.** Note that this reconstruction is intended to illustrate the proportions of the skeleton rather than a natural posture. The proportions of the mid–posterior dorsal ribs and the posterior patagial elements are based in part on the Eppleton weigeltisaurid skeleton (TWCMS B5937.1). 

## DISCUSSION

### Homology and function of the patagial ossifications
#### *Historical context*

SMNK-PAL 2882 preserves a nearly complete patagial apparatus. Our reconstruction of the skeleton and complete patagium in SMNK-PAL 2882 is presented in Fig. 20. A similarly complete structure is otherwise only known from a skeleton of *Coelurosauravus elivensis* (MNHN.F.MAP327; *Carroll, 1978*), the holotype of "*Daedalosaurus madagascariensis*". Since the initial reports of weigeltisaurid skeletons early in the twentieth century, the homology and function of these structures have been a source of continued controversy. In his description of SSWG 113/7—the holotype of *Weigeltisaurus jaekeli—Weigelt (1930a*: 626) noted that he had first heard about the specimen as a rumored record of "Der Flugsaurier im Perm"—the pterosaur in the Permian—from the Kupferschiefer.

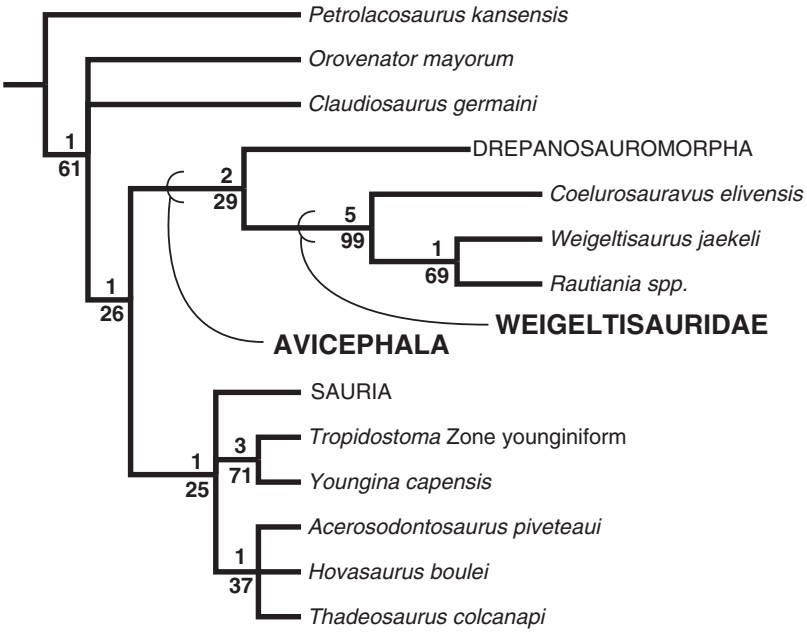

**Figure 21 Simplified strict consensus tree produced by this analysis.** The topologies of Drepano-sauromorpha and Sauria are congruent with those recovered by *Pritchard & Sues (2019)*. Numbers above branches represent Bremer Support values whereas numbers below branches represent frequency differences recovered from the Jackknife analysis.

*Weigelt (1930a)* dismissed the idea that SSWG 113/7 represented a flying reptile of any kind, despite the wing-like appearance of the patagials framing the partially preserved skeleton. Weigelt argued strongly that these elements were in fact the fin rays of a coelacanth, specifically *Coelacanthus haasiae* (now considered a subjective junior system of *C. granulatus*; *Schaumberg, 1978*), illustrating his taphonomic hypothesis with a photograph of a skeleton of a present-day pelican with a fish skeleton draped over it. Subsequent discussions of the holotype of *Weigeltisaurus jaekeli* in later decades would ignore the patagial ossifications based on this hypothesis (e.g., *Huene, 1930*, *1956*). Around the same time as Weigelt's work, *Piveteau (1926)* reported the first weigeltisaurid specimens from the Permian of Madagascar. The two skeletons he identified as *Coelurosauravus elivensis* clearly show the ribs of the trunk region, but neither preserved the patagial ossifications.

*Schaumberg (1976)* reported on two additional skeletons of *Weigeltisaurus jaekeli*. The so-called Wolfsberg specimen was part of the private collection of Wolfgang Munk (presently being transferred to the collections of the Naturkundemuseum im Ottoneum in Kassel) and consists of a partial skull and portions of the cervical, trunk, and caudal regions. The specimen preserves curved trunk ribs, part of the gastral basket ventrally, and two partial patagial-supported wings. The second specimen—dubbed the Bauhaus specimen—was discovered by the private collector W. Simon. It preserves only a partial caudal series, partial fore- and hindlimbs, and a series of patagials.

*Schaumberg (1976)* recognized the bony rods as belonging to the skeleton of *Weigeltisaurus* rather than fin rays from a coelacanth, noting both the improbability of the

association in three skeletons and the dissimilarity of the patagials from coelacanth fin rays. Although he offered no definitive hypothesis of their homology, Schaumberg speculated that the bones were related to the integument of the animal, rather than being part of the axial skeleton.

*Carroll (1978)* re-examined the skeletons of *Coelurosauravus elivensis* and described a third skeleton (MNHN.F.MAP327) that he designated as the holotype of a new, closely related taxon, *Daedalosaurus madagascariensis.* This specimen consists of a fragmentary skull and nearly complete, articulated postcranial skeleton complete with the right patagial ossifications. *Carroll (1978*:144) noted a few characters differentiating the species, one of which was the presence of "greatly elongate ribs, resembling those of the Upper Triassic gliding lizards *Kuehneosaurus* (*Robinson, 1962*) and *Icarosaurus* (*Colbert, 1970*)." Carroll noted both typical diapsid dorsal ribs and greatly elongated bony rods in the skeleton, but he considered these to represent the anteriormost and posterior dorsal ribs respectively. This was reflected in his reconstruction of the skeleton of *Daedalosaurus madagascariensis* (*Carroll, 1978*: fig. 9). *Evans (1982)* later argued that no major differences existed between *Coelurosauravus elivensis* and *Daedalosaurus madagascariensis*, and she synonymized the two under the former name.

*Pettigrew (1979)* reported on a beautifully articulated partial postcranial skeleton referred to *Weigeltisaurus* from the upper Permian Marl Slate of England (TWCMS B5937.1). The skeleton bears many complete, dichocephalous dorsal ribs of plesiomorphic diapsid morphology articulated throughout the preserved posterior dorsal vertebral column. The distal tips of these are closely associated with the ventral tips of a series of patagial ossifications. *Pettigrew (1979)* assumed a one-to-one correspondence between the ribs and the patagials, indicating that the latter were a distal segment of the ribs themselves that connected with the proximal rib at a mobile joint. *Evans (1982)* re-studied the Eppleton skeleton and agreed with Pettigrew's assessment of bipartite ribs.

*Schaumberg (1986)* returned to the anatomy of *Weigeltisaurus jaekeli* with studies of additional new specimens from the Kupferschiefer of Germany. He questioned the interpretations of *Pettigrew (1979)* and *Evans (1982)* regarding the homology of the patagials, noting that a one-to-one correspondence of patagial ossifications and dorsal ribs would require over 30 dorsal vertebrae in the Eppleton skeleton. He argued that the new skeletons he reported and the Wolfsberg specimen indicated a much lower dorsal vertebral count relative to the number of patagials. Homology of the patagial ossifications with any other part of the axial skeleton was considered unlikely; Schaumberg suggested both osteoderms or ossified tendons as potential homologs.

*Frey, Sues & Munk (1997)* briefly reported on SMNK-PAL 2882, the most complete weigeltisaurid skeleton to date. They also noted the discrepancy between the number of dorsal vertebrae and patagial bones, arguing that the bones were "neomorphic dermal ossifications that formed the internal support of a lateral gliding membrane" (*Frey, Sues & Munk, 1997*: 1451). In this contribution, we strongly support the hypothesis of *Frey, Sues & Munk (1997)* and provide additional observations on relevant structures in extant reptiles.
### Trunk anatomy and the homology of patagial ossifications

In considering the possible associations between the patagial ossifications and both hard- and soft-tissue structures in present-day reptiles, it is important to accurately assess how the bases of the patagial ossifications relate to the other bony elements of the trunk region. Among weigeltisaurid skeletons, the association between the patagials and the rest of the thoracolumbar region is best seen in medial view in SMNK-PAL 2882 and in lateral view in MNHN.F.MAP327 (*Coelurosauravus elivensis*). In both skeletons, the ventral tips of the patagials extend ventrally beyond the ventral tips of the preserved dorsal ribs, extending laterally over the gastral basket.

The gastral baskets in SMNK-PAL 2882 and MNHN.F.MAP327 are preserved but extensively disarticulated in both specimens. No gastralium clearly maintains its original articulation with another although the individual gastralia remain at or near the venter of the trunk of the animal, still arranged in a roughly anteroposterior row. In both cases, the patagials extend just ventral to the lateral edge of the lateralmost gastralia.

The position of the patagial ossifications external to both the body cavity as framed by the ribs and the gastralia is important for understanding their homology. The gastral basket is not part of the axial skeleton of the thoracolumbar region, but instead a structure of dermal origin. In the few extant diapsids that possess a gastral basket (e.g., *Sphenodon punctatus* and Crocodylia), the bones develop in the dermis of the venter of the thoracolumbar region (*Claessens, 2004*; *Fechner & Schwarz-Wings, 2013*). They extend from the posterior end of the pectoral girdle to the anterior edge of the pubis, embedded in the superficial layers of the *M. rectus abdominis* (*Reese, 1915*; *Byerly, 1926*; *Vickaryous & Hall, 2008*). They develop in loose mesenchyme externally superficially to the hypaxial musculature (*Vickaryous & Hall, 2008*) and become secondarily integrated into the muscles during development in *Crocodylus* (*Claessens, 2004*). Thus, the patagial ossifications would be positioned entirely external to the *M. rectus abdominis*.

The ventral edges of the patagials are adjacent to the lateral edge of the gastral basket, which may indicate a relationship between those soft tissue structures attached to the lateral edges of the gastralia. In *Sphenodon punctatus*, the *M. obliquus abdominis externus* originate on the dorsal ribs, extending ventrolaterally to the lateral edges of the gastralia (*Byerly, 1926*). In *Alligator mississippiensis*, the *M. obliquus abdominis externus* inserts superficially on the ventral fascia surrounding the *M. rectus abdominis* and the gastral basket, but deep to the specialized trunco-caudalis portion of the posterior part of the latter muscle (*Romer, 1923*; *Chiasson, 1962*; *Frey, 1988*; *Fechner & Schwarz-Wings, 2013*). Considering this positioning across extant diapsids, it is likely that at least the ventral edge of the patagials were dermal ossifications directly associated with the musculature attached to the lateral edge of the gastral basket, namely the *M. obliquus abdominis externus*.

Clarifying the patagials as dermal ossifications suggests various possible origins for the bones. One unlikely possibility is that a patagial represents an intermembranous ossification that formed within the myosepta of external trunk musculature. Such intermembranous bones are present in a wide variety of teleostean fishes, forming in the myosepta of the epaxial and hypaxial musculature of the body (*Patterson & Johnson,*

*1984*; *Gemballa & Britz, 1998*). However, such myosepta are eliminated early in the development of the trunk region of present-day amniotes such that the muscles form continuous sheets. Among tetrapods, the only reported example of intermembranous bones are ossifications in the caudal region of pachycephalosaurian dinosaurs (*Brown & Russell, 2012*).

An additional possibility is that the patagials represent lateral gastralia. In the few weigeltisaurid specimens that preserves extensive gastralia, the elements are often disarticulated such that it is not clear how many bones made up an individual segment within the basket. Those specimens preserve uniform, curved gastralia, which may represent the medial elements within each segment. The elongate, straight patagials would thus represent lateral gastralia. This hypothesis of homology would explain the higher number of patagials than dorsal vertebrae, as most Permo-Triassic diapsids (and *Sphenodon punctatus*) possess a higher number of gastral segments than trunk vertebrae (*Romer, 1956*; *Carroll, 1981*; *Benton, 1983*; *Muscio, 1996*). Alternatively, the patagial ossifications may represent true neomorphs without homology with any pre-existing soft or hard tissues. It is to be hoped that future discoveries and studies of articulated weigeltisaurid skeletons will clarify the relationships of the patagials to other skeletal elements.

### Comparative and functional morphology of the gliding membrane in Weigeltisauridae

Patagia that support a gliding membrane independently evolved several times in diapsid reptiles, with weigeltisaurids representing only the oldest known example of such a system (Table 1). However, in all other cases the homologies of the bones supporting the patagium are well understood. The most common patagium supports represented in extinct reptiles are elongated dorsal ribs (e.g., Kuehneosauridae, *Mecistotrachelos apreoros*, *Xianglong zhaoi*) that extend laterally or posterolaterally from their articulations with trunk vertebrae. A rib-supported patagium is also present in the extant gliding iguanians of the genus *Draco* of southeast Asia (e.g., *Colbert, 1967*; *John, 1970*; *Russell & Dijkstra, 2001*). The patagial supports would have prevented curling-up of the lateral edges of the patagial membrane during gliding, which would have decreased the surface area of the patagium and reduce its aerodynamical performance (*Frey, Sues & Munk, 1997*). *Dehling (2017)* showed that *Draco* attaches the forelimbs to the leading edge of the patagium while airborne. This attachment of the forelimbs to the patagium suggests that the airfoil is controlled through movements of the forelimbs. Dehling furthermore suggested that this was also probably the case in weigeltisaurids. As in *Draco*, the long tail may have aided in steering (*Frey, Sues & Munk, 1997*).

Weigeltisaurids likely had different gliding mechanisms from other patagium-borne gliders. In all other known gliders where the patagium is supported by a series of transversely elongate bony rods, the support structures are modified dorsal ribs (Table 1). These patagial ribs project laterally or ventrolaterally from the costal facets of the dorsal vertebrae, such that the patagium arises from the dorsolateral margin of the trunk. By contrast, the cross-section of the trunk in weigeltisaurids would be very different, with

**Table 1** Summary of extinct and extant diapsid reptiles that exhibit patagial membranes and/or associated bony support structures for gliding and flight.

| TAXON | Exemplar taxa | Primary patagium support | Stratigraphic range | General references |
|---|---|---|---|---|
| Weigeltisauridae (Neodiapsida) | *Weigeltisaurus jaekeli, Coelurosauravus elivensis, Glaurung schneideri, Rautiania* spp. | Dermal bone spars along trunk region | Upper Permian | *Schaumberg (1976, 1986), Schaumberg, Unwin & Brandt (2007), Evans (1982), Frey, Sues & Munk (1997), Bulanov & Sennikov (2006, 2010, 2015a).* |
| Kuehneosauridae (Neodiapsida, Sauria?) | *Kuehneosaurus latus, Kuehneosuchus latissimus, Icarosaurus siefkeri* | Dorsal ribs | Lower?–Upper Triassic | *Robinson (1962, 1967), Colbert (1966, 1970), Evans (2009), Pritchard & Nesbitt (2017)* |
| *Mecistotrachelos apeoros* (Archosauromorpha?) | Single species | Dorsal ribs | Upper Triassic | *Fraser et al. (2007)* |
| Sharovipterygidae (Archosauromorpha?) | *Sharovipteryx mirabilis, Ozimek volans?* | Hindlimb-supported uropatagium | Middle–Upper? Triassic | *Gans, Darevski & Tatarinov (1987), Unwin, Alifanov & Benton (2000), Dzik & Sulej (2016).* |
| *Xianglong zhaoi* (Squamata) | Single species | Dorsal ribs | Lower Cretaceous | *Li et al. (2007)* |
| *Draco* (genus) (Iguania) | *Draco dussumieri, Draco maculatus, Draco volans* | Dorsal ribs | Recent | *John (1970), McGuire & Heang (2001), Russell & Dijkstra (2001)* |
| *Chrysopelea* (genus) (Serpentes) | *Chrysopelea paradisi, Chrysopelea ornata* | Trunk ribs | Recent | *Heyer & Pongsapipatana (1970), Socha (2002, 2006, 2011), Holden et al. (2014)* |
| Gekkota (multiple lineages) | *Ptychozoon, Hemidactylus, Luperosaurus* | Webbing or dermal fringes on feet, hands, trunk, and tail | Recent | *Heinicke et al. (2012)* |
| Pterosauria (Archosauria) | *Rhamphorhynchus muensteri, Pterodactylus antiquus,* | Elongate forelimb with hypertrophied fourth digit | Upper Triassic–Upper Cretaceous | *Wellnhofer (1991), Unwin (2005), Witton (2013)* |
| Scansoriopterygidae (Theropoda) | *Ambopteryx longibrachium, Y. qi* | Elongate forelimb with manual digits and possible styliform bone | Upper Jurassic–Lower Cretaceous | *Xu et al. (2015), Wang et al. (2019)* |

the patagials arising from the ventrolateral surface of the trunk and projecting laterally or dorsolaterally. This low-wing configuration is unique among known vertebrate gliders and emphasizes the distinctiveness of the gliding mechanism of weigeltisaurids. *Frey, Buchy & Martill (2003)* presented evidence of a low-wing configuration in tapejarid pterosaurs. Drawing an analogy with aircraft wings, they suggested that low-wing configurations allowed greater maneuverability at the expense of lower stability than high-wing configurations. Stability in a low-wing configuration could be increased by angling the wings slightly dorsally (=dihedrally angled wings). Such dihedral angling of the wings would be present in weigeltisaurid patagia, considering the position of their bases within the ventrolateral margin of the trunk.

The gliding snake *Chrysopelea paradisi* presents a body cross-section most similar to that reconstructed for weigeltisaurids. In contrast to the gliding lizard *Draco*, the ribs in

*Chrysopelea paradisi* do not support a thin patagial membrane. To form its gliding surface, the snake abducts its ribs laterally such that the ventral surface of the trunk region is flat and the cross-section of the body resembles that of a cambered wing. However, this cross-section is not held more or less orthogonally to the direction of the glide as would be expected in a weigeltisaurid. Instead, the snake actively undulates its body so that parts of the cambered trunk cross-section are held in line with the direction of the glide (*Socha, 2011*; *Holden et al., 2014*).

The patagial ribs in extant *Draco* are highly flexible, capable of a great deal of bending. *Colbert (1967)* noted that the bending allows sufficient curvature to shift the shape of the wing from a flattened plane to a dorsally convex surface. *Russell & Dijkstra (2001)* noted strong posterior bending at the distal tips of the ribs when the patagium was fully extended. The tips of the patagials in some weigeltisaurid specimens show distinct posteromedial bending (e.g., TWCMS B5937.1, MNHN.F.MAP317) in otherwise undistorted skeletons, indicating flexibility comparable to that of the patagial ribs of *Draco*. Although the thickness of the anterior patagials in weigeltisaurids may have made them rigid, the flexible tips indicate the possibility of three-dimensional changes to the shape of the wing.

The patagium of weigeltisaurids was likely under muscular control in much the same way that rod-supported patagia are in extant reptiles. The wings of *Draco* have received the most attention, with published dissections by *Colbert (1967)*, *John (1970)*, and *Russell & Dijkstra (2001)*. All these authors noted prominent ligamentous connections between adjacent dorsal ribs and prominent intercostal musculature at the anterior and posterior margins of the intercostal spaces. They also reported the presence of a series of posterolaterally expanding muscles at the anterior margin of the patagium, although there exists disagreement concerning the homology of the muscles.

*Colbert (1967)* argued that this muscle was a component of the *M. iliocostalis*, whereas *John (1970)* considered it a component of the *M. obliquus abdominis externus*. *Colbert (1967)* did not note any muscular tissue framing the outer perimeter of the patagium, and *Russell & Dijkstra (2001)* described a collagen-rich band extending between the tips of the patagial ribs. However, *John (1970)* described a muscular expansion of the protracting muscles at the anterior margin of the patagium that extends along the perimeter of the wing. It is not clear if these discrepancies are a result of the contrasting interpretations of the authors or genuine differences between species of *Draco*. *Colbert (1967)* dissected *D. maculatus whiteheadi*, whereas *John (1970)* dissected *D. dussumieri*.

As the patagials of weigeltisaurids are definitely not derived from costal elements, it is highly unlikely that the patagium in weigeltisaurids was controlled by intercostal musculature. However, their close association with the gastral basket suggests that the *M. rectus abdominis* and *M. obliquus abdominis externus* may have been involved in the positioning of the patagials, at least at their proximal ends. In the lateral wall of the trunk in extant diapsid reptiles, the fibers of the *M. obliquus abdominis externus* pass ventrolaterally from their origins on the trunk ribs to insert on the fascia of the *M. rectus abdominis* and in close association with the gastralia where present. Such an orientation provides a mechanism by which the patagials could be protracted, abducted, and elevated from the lateral surface of the trunk. The scale of the patagials also suggests that the muscles of the

trunk wall were modified to accommodate the wings of weigeltisaurids. Examining the distribution of Sharpey's fibers within well-preserved individual patagials may provide further insight into the muscular mechanisms of patagial control in Weigeltisauridae.

*McGuire & Dudley (2011)* studied gliding performance in a range of species referable to *Draco* and compared their results with those for the Late Triassic gliding reptile *Kuehneosaurus* sp. and the weigeltisaurid *Coelurosauravus elivensis*. They argued that, due to their larger body size, both extinct taxa performed relatively poorly compared to both *Draco* and their estimates for the Late Triassic gliding reptile *Icarosaurus siefkeri*. *McGuire & Dudley (2011)* inferred that *Coelurosauravus* would have employed shallow-trajectory gliding requiring an extended ballistic dive and a substantial drop in height over the glide distance during each glide event. The Zechstein flora included a diversity of conifers and ginkgophytes but it is not known what heights the arborescent plants attained (*Bernardi et al., 2017*).

### Character support and discussion of relevant clades

Unnamed Clade Avicephala + ("younginiforms" + Sauria)

(Bremer = 1, GC [frequency difference] = 26)

**Unambiguous synapomorphies** - one row of teeth on transverse process of pterygoid (87.1), *dentition absent on cultriform process of parabasisphenoid (104.1), *retroarticular process present (137.1), vertebrae holocephalous in posterior dorsal region (192.1).

**Discussion** - We include five operational taxonomic units of early African diapsids that have been recovered within a monophyletic Younginiformes by some earlier analyses (e.g., *Benton, 1985*; *Gauthier, Estes & de Queiroz, 1988*) and as a paraphyletic or polyphyletic grouping of early diapsid taxa (e.g., *Bickelmann, Müller & Reisz, 2009*; *Pritchard & Nesbitt, 2017*; *Simões et al., 2018*). This excludes the Malagasy taxon *Claudiosaurus germaini*, which is recovered outside of the clade hypothesized here. Our analysis incorporates the *Tropidostoma* Zone younginiform identified as *Youngina capensis* by *Smith & Evans (1996)* as a separate operational taxonomic unit (OTU) because there are substantial differences in the skull and limb skeletons from the material of *Youngina capensis* from the *Dicynodon* Assemblage Zone (pers. obs. based on SAM-PK 7719, 8565 vs. BP/1 3859).

The early diapsid *Claudiosaurus germaini* is recovered in a polytomy with *Orovenator mayorum* and this clade. Most previous analyses find the former species more proximate to 'younginiform' diapsids (e.g., *Müller, 2004*; *Pritchard & Nesbitt, 2017*; *Simões et al., 2018*; *Ford & Benson (2019b)*). The constraint analysis of *Pritchard & Nesbitt (2017)* holding Weigeltisauridae and Drepanosauromorpha as sister taxa—as occurred in our study—did recover *Claudiosaurus* in a similarly basal position.

Avicephala

(Bremer = 2, GC = 29)

**Unambiguous synapomorphies** - absence of intercentra in cervical region (184.1); absence of intercentra in dorsal region (208.1); scapulocoracoid, ratio of anteroposterior length at base of scapular blade to dorsoventral height of scapular blade between 0.4 and 0.25 (233.1); outer process of fifth metatarsal absent (318.0).

**Discussion** - This analysis recovers a sister-group relationship between Drepanosauromorpha and Weigeltisauridae, the first iteration of the analysis by *Pritchard & Nesbitt (2017)* to do so. *Pritchard & Nesbitt (2017)*, *Pritchard et al. (2018)*, and *Pritchard & Sues (2019)* all found Drepanosauromorpha as the sister taxon to a clade including Weigeltisauridae and all other neodiapsids except *Orovenator mayorum*. Unambiguous synapomorphies supporting the clade have uniformly low consistency indices and include reductions in ossification of the postcranial skeleton (e.g., absence of intercentra throughout the vertebral column).

A monophyletic clade uniting weigeltisaurids and drepanosauromorphs was first proposed by *Merck (2003)* in an abstract for a presentation at the Annual Meeting of the Society of Vertebrate Paleontology in St. Paul, Minnesota. In his analysis, successive sister taxa to drepanosauromorphs included *Longisquama*, *Coelurosauravus*, and *Wapitisaurus*. *Senter (2004)* subsequently hypothesized a similar arrangement of early diapsids, recovering a clade *Coelurosauravus* + *Longisquama* as the sister taxon of drepanosauromorphs. He defined this as a stem-based taxon including "all taxa most closely related to *Coelurosauravus* and *Megalancosaurus* than to Neodiapsida" (*Senter, 2004*:261). However, *Renesto & Binelli (2006)* presented modified versions of the *Senter (2004)* matrix, incorporating the pterosaur *Eudimorphodon* and numerous additional codings of drepanosauromorph taxa. The analysis based on the recoded matrix placed Drepanosauromorpha + *Eudimorphodon* as the sister taxon of Archosauriformes, whereas *Coelurosauravus* + *Longisquama* remained outside of the diapsid crown-group.

Other previous analyses of early diapsid interrelationships have recovered Weigeltisauridae and Drepanosauromorpha in a wide variety of different positions. *Müller (2004)* recovered *Coelurosauravus* as the sister taxon of a clade including the poorly known South African diapsids *Palaeagama* and *Saurosternon*. By contrast, Drepanosauridae was recovered as sister to Kuehneosauridae in a clade one step closer to the reptilian crown group.

*Simões et al. (2018)* incorporated two weigeltisaurid species (*Coelurosauravus elivensis* and "*C.*" *jaekeli*) and only one drepanosauromorph, *Megalancosaurus preonensis*, into their character-taxon matrix. The two weigeltisaurid species were recovered in a monophyletic grouping outside of the reptilian crown group. *Megalancosaurus preonensis* was found as the sister taxon of the tanystropheid *Langobardisaurus pandolfii*, deeply nested in a clade outside of Sauria that also includes a polyphyletic grouping of tanystropheids and kuehneosaurids. This result differs radically from all other cladistic studies of early Diapsida and Archosauromorpha, in which tanystropheids, rhynchosaurs, azendohsaurids, and trilophosaurs are recovered as monophyletic groups nested closer to Archosauria than to Lepidosauria (e.g., *Gauthier, 1984*; *Benton, 1985*; *Rieppel, 1994*; *Dilkes, 1998*; *Müller, 2004*; *Gottmann-Quesada & Sander, 2009*; *Ezcurra, 2016*; *Pritchard & Nesbitt, 2017*; *Ezcurra & Butler, 2018*).

Weigeltisauridae
(Bremer = 5, GC = 99) -

**Unambiguous synapomorphies** - lacrimal contribution to lateral surface of rostrum restricted to orbital margin (25.2); jugal posterior process present and contacting

quadratojugal posteriorly to form complete lower temporal bar (58.2); *horn-like projections on lateral surface of squamosal (59.1); neural spines dorsoventrally lower than anteroposteriorly long (201.1); *patagial ossifications present in trunk region (204.1); chevrons T-shaped with prominent anterior process (219.1); *entepicondyle of humerus projecting distal to articular condyles (252.1).

**Discussion** - Our OTUs follow the taxonomic framework offered by *Bulanov & Sennikov (2006, 2015a, 2015b, 2015c)*, maintaining *Weigeltisaurus jaekeli* and *Coelurosauravus elivensis* in distinct genera. The monophyly of Weigeltisauridae has been accepted by previous authors studying this clade (e.g., *Evans, 1982*; *Bulanov & Sennikov, 2006, 2015b*). *Pritchard & Nesbitt (2017)* provided the first computational phylogenetic analysis to support the monophyly of this group with the same weigeltisaurid OTUs presented here. In their phylogenetic analysis, *Simões et al. (2018)* also recovered a monophyletic Weigeltisauridae.

*Evans (1982)* offered a general diagnosis for the clade, which included both apomorphies and plesiomorphies of the group. She recognized a small lacrimal, 'denticulated' squamosal, and low dorsal neural spines as diagnostic characters (*Evans, 1982*:297), all of which are also recovered here. In contrast to this study, *Evans (1982)* and *Evans & Haubold (1987)* reconstructed the temporal region of *Coelurosauravus* with an incomplete lower temporal bar and bipartite ribs rather than dermal patagial bones. Here we add T-shaped chevrons with a prominent anterior processes and a distally expanded entepicondyle of the humerus to the diagnostic characters of the clade.

*Weigeltisaurus jaekeli* + *Rautiania* spp.

(Bremer = 1, GC = 69)

**Unambiguous synapomorphies** - *horn-like projections on dorsolateral surface of parietal (43.1).

**Discussion** - Limited support for a clade including the European *Weigeltisaurus jaekeli* and *Rautiania* spp. is offered by this analysis. These taxa share the presence of horn-like projections on the parietal bones, in contrast to the Malagasy *Coelurosauravus elivensis*. The latter bears small ridges on the dorsolateral surfaces of the parietals, although these are not nearly as strongly developed as in the other weigeltisaurids. Although it is not coded into this analysis because the holotype is not curated in a publicly accessible museum collection, the German weigeltisaurid *Glaurung schneideri* also has a rugose dorsolateral margin of the parietal rather than horn-like projections. It is possible that *G. schneideri* is also basal to the *W. jaekeli*+*Rautiania* spp. clade or that it represents the sister taxon to *C. elivensis*.

Bony horn development and ontogeny can be closely associated in diapsid reptiles. The horns and epiossifications often grow at differential rates on a single animal's skull, as in marginocephalian dinosaurs (e.g., *Sampson, Ryan & Tanke, 1997*; *Horner & Goodwin, 2006*; *Goodwin et al., 2006*) and phrynosomatid lizards (e.g., *Powell, Russell & Ryan, 2002*; *Bergmann & Berk, 2012*). It is plausible that weigeltisaurid horns also developed at different rates such that the lateral arcade of horns developed prior to the parietal horns, such that the condition in *Coelurosauravus elivensis* represents an earlier ontogenetic state than is represented by known specimens of *Rautiania* spp. and *Weigeltisaurus jaekeli*.

Testing that hypothesis will require discovery of ontogenetic sequences of skulls from at least one weigeltisaurid species.

(*Youngina capensis* + *Tropidostoma* Zone 'younginiform') + (*Acerosodontosaurus piveteaui* + *Hovasaurus boulei* + *Thadeosaurus colcanapi*) + Sauria

(Bremer = 1, GC = 25)

**Unambiguous synapomorphies** - *squamosal posterior lamina absent, fully exposing quadrate in posterior view (61.1); *basal tubera of basioccipital project strongly ventral to occipital condyle (101.1).

**Discussion** - "Younginiformes" has been used to describe a monophyletic grouping of largely Permian diapsids including the eponymous *Youngina capensis* and several species from Permian deposits of Madagascar, such as *Acerosodontosaurus piveteaui*, *Hovasaurus boulei*, and *Thadeosaurus colcanapi* (*Currie, 1980*, *1981a*). 'Younginiforms' were long considered the earliest diverging clade of lepidosauromorphs, and numerous early cladistic studies explored this hypothesis (e.g. *Gauthier, 1984*; *Gauthier, Estes & de Queiroz, 1988*; *Evans, 1988*). However, more comprehensive analyses of early diapsid and saurian clades began to recover the traditional 'younginiform' taxa outside of Sauria (e.g., *Laurin, 1991*; *Merck, 1997*; *Dilkes, 1998*; *Müller, 2004*; *Senter, 2004*; *Ezcurra, Scheyer & Butler, 2014*; *Pritchard & Nesbitt, 2017*). *Bickelmann, Müller & Reisz (2009)* noted many anatomical differences among 'younginiform' species and raised the possibility that "Younginiformes" as traditionally conceived (e.g., *Currie, 1982*; *Benton, 1985*) represent a paraphyletic grade of non-saurian diapsids.

Although this node is not strongly supported in this phylogenetic analysis, the clade incorporating traditional 'younginiform' taxa and Sauria to the exclusion of all other early Diapsida has been recovered in prior iterations of the phylogenetic analysis of *Pritchard & Nesbitt (2017)*. A similar topology was recovered by *Laurin (1991)* and *Ezcurra, Scheyer & Butler (2014)*, in which a clade of *Youngina capensis* +*Acerosodontosaurus piveteaui* was recovered as the sister taxon of Sauria. Our analysis does not recover a monophyletic younginiform clade in the strict consensus. However, the Malagasy diapsids *Acerosodontosaurus piveteaui*, *Thadeosaurus colcanapi*, and *Hovasaurus boulei* are recovered as one clade. *Youngina capensis* and the *Tropidostoma* Zone younginiform are recovered as another. Both are part of a polytomy with Sauria.

By contrast, *Merck (1997)* recovered a clade *Acerosodontosaurus*+(*Youngina capensis* +*Hovasaurus boulei*) as the sister taxon of a clade *Claudiosaurus*+Sauria. *Müller (2004)* found his Younginiformes OTU well removed from Sauria, with *Claudiosaurus*, *Coelurosauravus*, Kuehneosauridae, and Drepanosauridae all positioned more proximate to the latter. *Simões et al. (2018)* similarly recovered a variety of diapsid clades as closer to Sauria than Younginiformes, including *Coelurosauravus*, a comprehensive marine reptile clade (unnamed), Choristodera, and a monophyletic Protorosauria. The exclusion of 'protorosaurs' from the stem leading to Archosauria is a highly divergent result, in contrast to all other published studies of early diapsid interrelationships.

The position of the traditional 'younginiform' reptiles as intermediate between archaic diapsid lineages and Sauria in this analysis is supported by changes in the posterior portion of the skull, including the reduction in the breadth of the ventral process of the

squamosal and consequent lateral exposure of the quadrate. *Youngina capensis* and the *Tropidostoma* Zone younginiform also share with early saurians the absence of a posterior lamina of the squamosal and the presence of a dorsal articular condyle on the quadrate for the squamosal. These all appear to be an incipient version of the prominent cephalic condyle of the quadrate in early Sauria and the narrow, laterally positioned ventral process of the squamosal. In many early saurians, these traits are accompanied by a posterior concavity of the quadrate and the presence of a lateral crest that likely supported a tympanic membrane (e.g., *Robinson, 1973*; *Müller, Bickelmann & Sobral, 2018*). In others, the ventral lamina of the squamosal is completely absent such that the quadrate may have been mobile, allowing the streptostyly of Squamata (e.g., *Robinson, 1973*; *Rieppel & Gronowski, 1981*). As such, the *Youngina*-type condition of the temporal region may be functionally intermediate in two ways: first, as the initial stage of reduction in the lateral and posterior bracing of the quadrate by the squamosal that would possibly permit cranial kinesis in some early saurian lineages (e.g., *Robinson, 1973*; *Rieppel & Gronowski, 1981*); and second, the lateral exposure of the portion of the quadrate that, in derived diapsids, closely correlates with the support of the tympanum in early Sauria (e.g., *Robinson, 1973*; *Montefeltro, Andrade & Larsson, 2016*; *Sobral & Müller, 2019*).

## CONCLUSIONS

SMNK-PAL 2882 substantially advances our understanding of skeletal structure in the Weigeltisauridae, the oldest-known example of a lineage of gliding tetrapods in the fossil record. The skeleton not only provides detailed information for the anatomy and proportions of the skeleton in *Weigeltisaurus jaekeli*, but also provides key insights into the topology of the patagial ossifications to the remainder of the skeleton. Our study supports the hypothesis that the patagials represent dermal ossifications embedded in the ventrolateral surface of the trunk, possibly associated with the *M. obliquus abdominis externus*. Our phylogenetic analysis corroborates the hypothesis that the Weigeltisauridae represent an early-diverging diapsid clade. In our analysis, we resolve the neodiapsids *Youngina capensis*, *Thadeosaurus colcanapi*, and other traditional 'younginiforms' as closer relatives of Sauria. Future studies of this skeleton and other weigeltisaurid specimens using advanced imaging techniques may increase our three-dimensional understanding of the skeletal structure in these reptiles.

### Institutional Abbreviations

The following institutional abbreviations are use throughout the text and figure captions: **AMNH**, American Museum of Natural History (New York, New York, USA); **BP/1**, Evolutionary Studies Institute, University of the Witwatersrand (Johannesburg, South Africa); **CAS**, California Academy of Sciences (San Francisco, California, USA) ; **CM**, Carnegie Museum of Natural History (Pittsburgh, Pennsylvania, USA); **FLMNH**, Florida Museum of Natural History (Gainesville, Florida, USA); **GR**, Ruth Hall Museum of Paleontology (Abiquiu, New Mexico, USA); **IVPP**, Institute of Vertebrate Paleontology and Paleoanthropology (Beijing, China); **MCSNB**, Museo Civico Scienze Naturali Enrico Caffi (Bergamo, Italy); **MCZ**, Museum of Comparative Zoology, Harvard University

(Cambridge, Massachusetts, USA); **MFSN**, Museo Friulano di Storia Naturale (Udine, Italy); **MNHN**, Muséum National d'Histoire Naturelle (Paris, France); **MPUM**, Museo di Paleontologia Università degli Studi di Milano (Milan, Italy); **NHMUK**, Natural History Museum of the United Kingdom (London, UK); **NMQR**, National Museum Bloemfontein (Bloemfontein, South Africa); **NSM**, Nova Scotia Museum (Halifax, Nova Scotia, Canada); **PIMUZ**, Paläontologisches Institut und Museum der Universität (Zürich, Switzerland) **PIN**, Borissiak Paleontological Institute of the Russian Academy of Sciences (Moscow, Russia); **SAM**, Iziko South African Museum (Cape Town, South Africa); **SMNK**, Staatliches Museum fur Naturkunde Karlsruhe (Karlsruhe, Germany); **SSWG**, Sektion Geologie, Universität Greifswald (Greifswald, Germany); **TM**, Ditsong National Museum of Natural History (Pretoria, South Africa); **TMM**, Texas Memorial Museum (Austin, Texas, USA); **TWCMS**, Sunderland Museum of Tyne & Wear County Museums (Sunderland, UK); **UCMP**, University of California Museum of Paleontology (Berkeley, California, USA); **VMNH**, Virginia Museum of Natural History (Martinsville, Virginia, USA).

## ACKNOWLEDGEMENTS

We thank Eberhard Frey and Dieter Schreiber for facilitating access to the collections of the Staatliches Museum für Naturkunde Karlsruhe and the extended loan of SMNK-PAL 2882. We also thank J. Sébastien Steyer and Valentin Buffa for valuable discussions about the specimen and Weigeltisauridae. A. Pritchard would like to thank V. Bulanov and A. Sennikov (PIN), R. Allain (MNHN), R. Schoch and E. Maxwell (SMNS), S. Chapman and L. Steel (NHMUK), R. and Z. Erasmus (SAM), B. Zipfel (BP/1), J. Botha-Brink (NMQR), A. Downs (GR), M. Norell and C. Mehling (AMNH), A. Dooley and R. Vodden (VMNH), J. Liu (IVPP), A. Paganoni (MCSNB), A. Tintori (MPUM), G. Muscio (MFSN), J. Cundiff (MCZ), T. Rowe and M. Brown (TMM), and H. Furrer (PIMUZ) for access to specimens that proved invaluable for our comparative description. We thank D. Blackburn for access to digital datasets of extant reptiles on the Morphosource digital data repository (www.morphosource.org). For *Plica plica*, the California Academy of Sciences provided access to these data, the collection of which was funded by oVert TCN; NSF DBI-1701714; NSF DBI-1701713; NSF DBI-1701870. For *Draco dussumieri*, the Florida Museum of Natural History provided access to these data the collection of which was funded by oVert Thematic Collections Network (TCN), NSF DBI-1701714. The TNT software is freely available courtesy of the Willi Hennig Society. We thank J. Dzik, S. Renesto, and V. Buffa for their detailed reviews, which improved the quality of this manuscript. We also thank the editorial staff of *PeerJ*, especially R. Langshaw and F. Knoll, for their technical assistance with this submission.

### Funding

Adam C. Pritchard's work on SMNK-PAL 2882 and the phylogeny of early Diapsida was supported by a Peter Buck Postdoctoral Fellowship and the Smithsonian Institution.

Funding for Hans-Dieter Sues was provided by the Smithsonian Institution. Robert R. Reisz and Diane Scott were funded by an NSERC Discovery Grant (to Robert R. Reisz). The funders had no role in study design, data collection and analysis, decision to publish, or preparation of the manuscript.

### Grant Disclosures
The following grant information was disclosed by the authors:
SMNK-PAL 2882.
Phylogeny of early Diapsida.
Smithsonian Institution.
NSERC Discovery.

### Competing Interests
The authors declare that they have no competing interests.

### Author Contributions
- Adam C. Pritchard conceived and designed the experiments, performed the experiments, analyzed the data, prepared figures and/or tables, authored or reviewed drafts of the paper, and approved the final draft.
- Hans-Dieter Sues conceived and designed the experiments, performed the experiments, analyzed the data, authored or reviewed drafts of the paper, and approved the final draft.
- Diane Scott conceived and designed the experiments, performed the experiments, analyzed the data, prepared figures and/or tables, authored or reviewed drafts of the paper, and approved the final draft.
- Robert R. Reisz conceived and designed the experiments, performed the experiments, analyzed the data, authored or reviewed drafts of the paper, and approved the final draft.

### Data Availability
The phylogenetic data matrix and the changes to the codings in the matrix from its prior iteration in *Pritchard & Sues (2019)* are available in the Supplemental Files.

The data matrix is also available at MorphoBank. Project DOI 10.7934/P3656.

The specimen primarily described in this contribution is SMNK-PAL 2882, housed in the paleontological collections of the Staatliches Museum für Naturkunde Karlsruhe, Karlsruhe, Germany.

### Supplemental Information
Supplemental information for this article can be found online at http://dx.doi.org/10.7717/peerj.11413#supplemental-information.

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
