# Peer review of "Osteology, relationships and functional morphology of Weigeltisaurus jaekeli (Diapsida, Weigeltisauridae) based on a complete skeleton from the Upper Permian Kupferschiefer of Germany"

_PeerJ, doi:10.7717/peerj.11413_

## Round 0.1 · original submission · Minor Revisions

Overall the reviewers are positive. Nevertheless, they raise a number of issues, which should be addressed. They also offer several suggestions for improvement, which should be taken into account.
Please, together with your unmarked revised manuscript, provide a marked-up copy as well as a document explaining how you have addressed each of the points raised by the reviewers.

·

Basic reporting

Correct

Experimental design

This is a descriptive palaeontological paper

Validity of the findings

The newwly described fossil is of a great scientific value

Additional comments

This is a very well executed piece of palaeontological work and it is hard to find any substantial deficiency in the descriptive part of the paper. However, illustrations require some modification before they are published. For instance, it is hardly useful for the reader to have political maps presented on Fig. 1. One may easily find the district of Mansfeld-Südharz in the state of Saxony-Anhalt in smartphone without consulting this scientific paper. I suggest that rather the extend of the Kupferschiefer and the Zechstein Basin is shown with location of the finding.
One may ask for what purpose contours of bones on photographs on Figs 2, 3, 5, 6 and 17 are retouched, the same outlines being repeated on Fig. 3A and B, 17A and !7 B and 19. This obliterates the actual information provided by the fossil and unnecessarily mixes facts with interpretations. I suggest that these outlines are removed.
The reconstruction shows an animal in an unrealistic pose. In result, the patagial spars are so close to each other in their presumably resting folded orientation that there is little space for the patagium, whereas the appendages protrude straight to sides, in a gliding(?) orientation.
Probably the most interesting aspect of the paper from functional point of view, is the convincing evidence that the patagia were located ventro-laterally, which required their upward elevation during flight. Such a non-optimal arrangement requires historical explanation and the authors convincingly propose that they are homologous to the external rows of gastralia.
The long thin tail in flying reptiles (line 1636) is designed rather for stabilizing flight (like a rod in rocket) than for steering, which requires flattening of the tail.
The methodology of phylogenetic analysis used by the authors allows to receive any expected results by assembling appropriate matrix of data. The resulting cladogram is thus hardly a surprise.
No doubt that the paper deserves publication after some modification of pictures.

Jerzy Dzik

·

Basic reporting

Adding a figure is suggested.

Experimental design

no comment

Validity of the findings

no comment

Additional comments

The article is well organized, informative and scientifically relevant, thus I have only very few remarks:
At line 253-256 the Authors wrote that an anterodorsal (internarial) process of the premaxilla much longer than the alveolar proces is known in Weigeltisaurids, while in nearly all other reptiles the alveolar process is longer, however the same condition occurs in the Drepanosauromorpha in which this element is known, i. e. Megalancosaurus (Renesto and Dalla Vecchia 2005) and Vallesaurus (Renesto and Binelli 2005), and in Pterosaurs (e.g. Eudimorphodon, Rhamphorhynchus...), maybe this could be reported.

I definitely agree with the new reconstruction of the pattern of the patagials proposed by the Authors, but would suggest to briefly discuss (and dismiss) Frey et al. (1997) reconstruction with most of the patagials inserted anteriorly, that could possibly have been folded.

Also, at line 1629-1632, when the authors state that Weigeltisaurids had a low wing configuration, suggesting they had a peculiar gliding mechanism, a reference colud be made to the fact that low-wing structure may decrease stability but increase manoeuvrability, as hypothesized by Frey et al. (2003) for active fliers.

In summarizing previous phylogenies the authors should include also Renesto and Binelli 2005 just for sake of completeness.

Adding a drawing of the skull reconstructed in lateral view could be very informative.

Frey E., Buchy M. C. and Martill D. M (2003) Middle- and bottom- decker Cretaceous pterosaurs: unique designs in active flying vertebrates. In Buffetaut E. and Mazin J.M. (eds) 2003 Evolution and Paleobiology of Pterosaurs Geological Society Special Oublications 217:267-274. The Geological Society of London

·

Basic reporting

I thank the editors (and possibly the authors who proposed my name) for their invitation to review this work. As I am currently working on weigeltisaurid anatomy in the context of my PhD, I aim here to provide a constructive and objective review, and not to prevent the publication of this paper.

As stated by the authors, the main goal of the study is to provide an accurate description of the weigeltisaurid specimen SMNK-PAL 2882, explore the systematic position of the weigeltisauridae, and discuss the homology of the enigmatic patagial spars. This work was long overdue and I commend the authors on their precise anatomical study of this specimen. Nevertheless, I think that some minor issues need to be addressed in a revised manuscript:

1. How was SMNK-PAL 2882 identified as Weigeltisaurus jaekeli? Bulanov and Sennikov (Pal. Jour., 2015:1110) explicitly refer this specimen to the genus Weigeltisaurus but not to the species W. jaekeli. Please address this in the manuscript.

2. I suggest adding a ‘Systematic Paleontology’ section to (a). reassess the attribution of the specimen; (b). provide an emended diagnosis for W. jaekeli based on your phylogeny; (c). summarize the previous studies of this specimen.

3. Thank you for providing your matrix. However, Scheyer et al. (Roy. Soc. Open. Sci., 2020) have provided a revision of its 2018 version, but this study is not mentioned here. Please reference this article and explain why their modifications were not included in your work.

4. There is no mention of recent hypotheses on early amniote evolution (e.g. Laurin and Piñeiro, Front. Earth. Sci., 2017; Ford and Benson, Nat. Ecol. Evol., 2020). This may impact your discussion on early diapsid relationships, considering you refer to Orovenator (a varanopid according to Ford and Benson, 2019, 2020). If you prefer to maintain a more consensual view pertaining replication of such results, please say so.

5. I see some problems with the figures: (a). I suggest modifying Fig. 1 to illustrate the geological setting (please indicate Kupferschiefer strata, Ellrich city and precise locality if known); (b). Some bone colors are too similar on Fig. 3B and a portion of the right hemimandible is not figured; (c). Additional captions are welcome on Fig. 4 (e.g. tooth regions; tooth numbers; resorption pits; replacement teeth); (d). The photographs on Figs. 7A and 14A are not very sharp and the individual bones are therefore hard to follow.

Experimental design

See Remarks 1 and 3 in ‘Basic reporting’ section.

Validity of the findings

No comment.

Additional comments

Some minor general comments on the manuscript:

1. Incorrect specimen numbers: SMNK-PAL 2882, not SMNK 2882. Updated numbers for the MNHN collections are available online (https://science.mnhn.fr/). Note that R.L. Carroll and P.J. Currie erroneously refer to ‘1908-11-13a’ instead of ‘1908-11-18a’ for Thadeosaurus. The updated number for this specimen is MNHN.F.MAP360a. I provided the updated numbers below. Please correct occurrences in the manuscript and check other specimen numbers for similar mistakes.

2. Your ‘History of Research’ section is very similar to the summaries provided by Schaumberg et al. (Pal. Z., 2007:160-161) and Bulanov and Sennikov (Pal. Jour., 2015:1101-1104). I suggest shortening the first section to focus on the history of the studied specimen.

3. The Dentition section is hard to follow, for the most part due to Fig. 4, which I highly suggest revising (see ‘Basic Reporting’, point 5).

4. As noted, the clade Avicephala was described relative to Neodiapsida. However, the definition of neodiapsida was changed by Reisz et al. (Proc. Roy. Soc., 2011:3733), and includes weigeltisaurids in your analysis. Thus, I believe you should not refer to “Avicephala (sensu Senter, 2004)” but ‘Avicephala (sensu nuovo)’. I suggest you provide a new definition for this clade.

5. Your conclusion should be improved to include all results in the manuscript such as your interesting hypothesis of the patagial spars as lateral gastralia, or the grouping weigeltisaurids and drepanosauromorphs under Avicaphala in your analysis.

6. Please include the age and provenance of SMNK-PAL 2882 in all relevant figure captions. e.g. ‘Figure 2: The skeleton of SMNK-PAL 2882 (Weigeltisaurus jaekeli, Lopingian, Germany)’. Use the age and locality you deem adequate.


Some minor comments to improve the manuscript:

line 69: “The most highly specialized” -> “One of the most…”
line 79: add ‘in’/’near’ between “Ellrich” and “the Mansfeld mining district”
line 190: “All elements of the palate and braincase are absent”. Some unidentified bones on Fig. 3A could be unidentifiable elements from the palate or braincase. Please be less affirmative.
lines 204-207: hard to follow, please reformulate. Line 205 should refer to the tarsus instead of the carpus?
line 363: “blunted point” -> this appears quite sharp to me on Fig. 3A
line 364 (nasal section): Bulanov and Sennikov (2015:1107) describe an anterolateral crest on the Weigeltisaurus holotype. Is it visible on this specimen?
line 377: ‘contribution to’ instead of “contribution for”
line 378: “This contact is sigmoid […] dorsal process of the maxilla” -> This is not clearly visible on Fig. 3.
line 386 (maxilla section): There is no mention of the preorbital fenestra described on Coelurosauravus and Weigeltisaurus by Bulanov and Sennikov. This may be a diagnostic character of weigeltisaurids. If it is unpreserved on the specimen, please say so.
line 386 (maxilla section): Please, indicate the tooth number and compare it to other weigletisaurids (this may be placed in the dentition section)
line 404: MNHN.F.MAP327a-b instead of MNHN 1908-5-2, correct all occurrences.
line 416 (prefrontal section): can you see the foramen for the nasolacrimal duct described in Coelurosauravus? If not please say so.
line 416: reference Fig. 3 here
line 419: “prefrontal boss” -> I cannot see it on Fig. 3.
lines 426-428: “transverse breadth” -> as preserved or once reconstructed? This is not the same, I believe you mean ‘dorsoventral height’ (once reconstructed).
line 441 (frontal section): can you see traces of the median ventral lamina described in Coelurosauravus? If not please say so.
lines 464-469: “U-shaped frontoparietal contact”. Possibly, but the ‘zigzaged’ suture described in Coelurosauravus (Bulanov and Sennikov, 2015:417) seems plausible based on Fig. 3. If not, please say so.
line 534: replace the first “and” by a ‘,’
line 540: This is incorrect for Coelurosauravus (Bulanov and Sennikov, 2015:419)
line 567 (postorbital & prefrontal section): you say both bones are hard to distinguish and possibly fused, yet you describe and figure them as distinct elements. Please rephrase.
line 568: “fused into a single element” -> you should reference extant squamates here.
line 593: I suggest you also number the parietal horns.
line 682: “surangular” -> please indicate on Fig. 3.
line 689 “splenial” -> please indicate on Fig. 3.
line 729: remove ‘.’
line 810: “distinct diapophysis” -> please indicate on Fig. 5 or 6.
line 813 “dichocephalous cervical rib” -> refer to a Figure here.
line 816: Please provide value for cervical centrum height-length ratio.
line 817: MNHN.F.MAP317b instead of MNHN 1908-11-22a. Correct all occurrences.
line 899: see minor general comment 1.
line 903 “anterior and posterior cavities” -> add ‘in lateral view’
line 942 (chevrons section): Please compare your observations to that of Evans (Zool. Journ. Lin. Soc., 1982).
line 1005 (cleithrum): a cleithrum is also reported in Acerosodontosaurus.
line 1012 (clavicle): please indicate the clavicle on Fig. 7B
line 1042: “It does not bear any prominent crests or ridges” -> Could they have been obliterated due to postmortem compression?
line 1047: MNHN.F.MAP325a instead of MNHN 1908-11-21a
line 1058: MNHN.F.MAP360b instead of MNHN 1908-11-19a
line 1250: “median tibial condyle” -> please indicate on Fig. 15
line 1273: “There is no clear, proximally projecting cnemial crest” -> Yet you figure and describe it. If you consider it to be present bur reduced compared to other taxa, please say so.
line 1333 (metatarsus): please describe metatarsal V in more detail, including key characters coded in your matrix (e.g. outer process, hooking, etc.)
line 1441: this should refer to the 7th patagial instead of the 6th?
line 1472: see minor general comment 4.
line 1492: please provide translation for German terms.
line 1671: add ‘of weigeltisaurids’ after “As the patagials”.
line 1707 ‘Smith and Evans (1996)’ instead of “Smith and Evans (1992)”
line 1709: are the differences between both ‘Youngina’ forms based on personal observations? If so, please add “pers. obs.”
line 1921: ‘S2’ instead of “S@”

Matrix: ‘Petrolacosaurus kansensis’ instead of “Petrolacosaurus kansasensis”

---

## Round 0.2 · Minor Revisions

Please address the points raised by Reviewer 3. I will make my final decision rapidly after your resubmission.

·

Basic reporting

The revisions are adequately addressed, minor items to take into account are indicated below.

Experimental design

No comment.

Validity of the findings

No comment.

Additional comments

The authors have adequately addressed all of the requested revisions, including my own. They have argued strongly in favor of keeping a detailed ‘History of Research’ section. This is fine and, as correctly stated, is relevant for readers not familiar with weigeltisaurids.
There are, however, a few very minor items that need to be addressed before acceptance. Otherwise the revisions are adequate and the modified figures are very helpful:

1. Please address the following items in the Systematic Paleontology section:
- Line 188: The taxonomic authority should be in brackets: “(Weigelt, 1930)”.
- Lines 190-205: A chresonymy of W. jaekeli would be more pertinent in a revision of all of the specimens previously referred to this taxon, which is not the case here. Thus, I suggest providing only a synonymy sensu stricto for W. jaekeli in this work.
Furthermore, I strongly suggest following Matthews (Paleontology, 1973) and others, summarized in Sachs et al. (PeerJ, 2016) for the format of chresonymies. This is necessary to avoid confusion between taxonomic authorities and bibliographic references. If the authors find it pertinent, such a chresonymy could be provided for SMNK-PAL 2882 only, in the “referred material” paragraph.
- A “Holotype” paragraph should be included in this section even if it is not the specimen described here.


2. Please address the following items on the figures and figure captions:
- Fig. 3A: the interpretive lines have been removed, please correct the caption.
- Fig. 4: 21 should refer to a “maxillary” tooth position instead? 18 and 32 refer to “mandibular” and “dentary” tooth positions respectively, please keep only one.
- Fig. 6: “A.” and “B.” are bolded in the caption, but not in other figs. Please keep the same format for all captions.
- Figs. 12-13: Please indicate individual metacarpals as done in the tarsus (Figs. 17-18).

3. Please address the following items in the text:
- Line 130: the correct reference is De Stefano (1903) and this item is missing from the reference list.
- Line 254: Weigeltisaurus jaekeli should be in italics.
- Lines 303-304: I believe the bracketed references in the quote should be removed.
- Line 557: outer margins “are” discernable (not “is”).
- Line 861: remove one “of”.
- Line 1362: this should refer to MNHN.F.MAP325 instead.
- Lines 315, 646, 1123-1124, 1209, 1497, 2074: remove additional space. Check for others I might have missed.

---

## Round 0.3 · Minor Revisions

The issue raised by Reviewer 3 must be addressed before I make my decision.

·

Basic reporting

Almost all revisions have been adequately addressed. I only have a few minor comments.

Experimental design

No comment.

Validity of the findings

No comment.

Additional comments

I am really sorry for nitpicking in an otherwise excellent manuscript, but there remain a few very minor items to correct in the synonymy.

Synonymies are subject to a highly standardized format, and must include taxonomic authorities, if nothing else. References of works simply using previously established names should thus be removed.
A “simple” example could thus be: [Weigeltisaurus jaekeli (Weigelt, 1930a) removed as it is the valid name]

Palaeochamaeleo jaekeli Weigelt, 1930a
Gracilisaurus ottoi Weigelt, 1930b
Coelurosauravus jaekeli (Weigelt, 1930a)

Additional information could be included if the authors or the editors find it pertinent. Most journals require references to the first page where a name is established and all figures in that work. Authorities of new combinations may also be included, in which case the valid name should be re-included as it is a new combination.
If the authors wish to add such information, I suggest following the guidelines of specialized paleontological journals for all entries (e.g. JVP).

Other minor items:
Line 195: ‘Schaumberg et al., 2007’ instead of ‘Schaumberg, 2007’ but I suggest removing this, see above
Line 199: ‘patagials’ not ‘patagial’

---

## Round 0.4 · Minor Revisions

A scale should be added to every figure and the captions modified accordingly.

---

## Round 0.5 · accepted · Accept

I accept your MS now but urge you to add a scale on figs 1A and 20.